# Learning a 1-layer conditional generative model in total variation

**Ajil Jalal**  **Justin Kang**
UC Berkeley
{ajiljalal, justin_kang}@berkeley.edu

**Ananya Uppal**
UT Austin
ananya.uppal09@gmail.com

**Kannan Ramchandran**
UC Berkeley
kannanr@eecs.berkeley.edu

**Eric Price**
UT Austin
ecprice@cs.utexas.edu

## Abstract

A conditional generative model is a method for sampling from a conditional distribution $p(y \mid x)$. For example, one may want to sample an image of a cat given the label "cat". A feed-forward conditional generative model is a function $g(x, z)$ that takes the input $x$ and a random seed $z$, and outputs a sample $y$ from $p(y \mid x)$. Ideally the distribution of outputs $(x, g(x, z))$ would be close in total variation to the ideal distribution $(x, y)$.

Generalization bounds for other learning models require assumptions on the distribution of $x$, even in simple settings like linear regression with Gaussian noise. We show these assumptions are unnecessary in our model, for both linear regression and single-layer ReLU networks. Given samples $(x, y)$, we show how to learn a 1-layer ReLU conditional generative model in total variation. As our result has no assumption on the distribution of inputs $x$, if we are given access to the internal activations of a deep generative model, we can compose our 1-layer guarantee to progressively learn the deep model using a near-linear number of samples.

## 1  Introduction

Generative models are in the midst of an explosion in accessibility, as models like DALL-E [31] or Stable Diffusion [32] capture the attention of millions. In many cases, these generative models can be succinctly represented by a fundamental mathematical object—the conditional distribution $p(y \mid x)$. In the example of text-to-image generative models, $x$ can represent a text prompt or its Word2Vec embedding [26], and the model can be seen as sampling an image $y$ from its conditional distribution $p(y \mid x)$. With large numbers of people accessing these models, a massive amount of sample pairs $(y_i, x_i)$, are becoming available online. A natural question to ask is: How many samples $(y_i, x_i)$ does it take to learn the conditional generative model $p(y \mid x)$?

Recent empirical studies such as Stanford Alpaca [34], which attempt to learn GPT-3.5 from limited samples, indicate that the number of samples needed may be within a practical range. In this paper, we attempt to address this problem from a fundamental perspective grounded in a concept from classical theoretical statistics: the Maximum Likelihood Estimator (MLE). Specifically, we focus on feed-forward generative models from a relatively simple family and ask: with *no* assumptions on $x$, how many samples are required to efficiently learn the conditional generative model $p(y \mid x)$?

**Linear Regression.**  Consider ordinary linear regression with Gaussian noise: you observe independent samples $(x_i, y_i) \in \mathbb{R}^k \times \mathbb{R}$ of the form

$$y = x \cdot w^* + \eta \qquad \text{for } \eta \sim N(0, 1).$$

37th Conference on Neural Information Processing Systems (NeurIPS 2023).

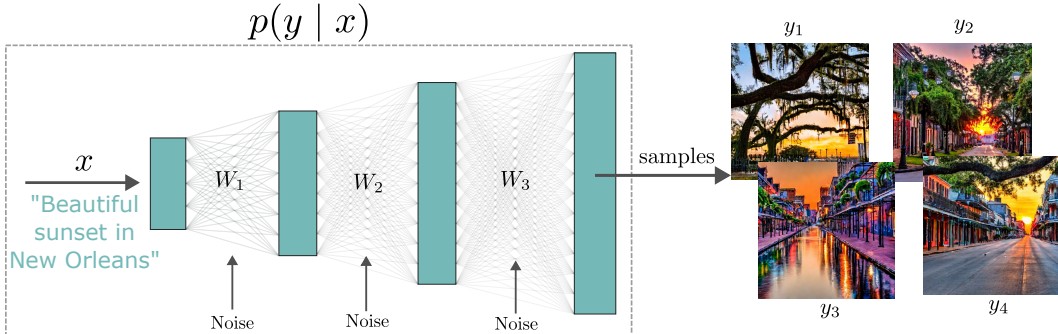

Figure 1: A conditional distribution defined by a conditional generative model. To sample from the conditional distribution $p(y \mid x)$, we perform inference. Due to the stochastic nature of the model, each output is different.

How many samples does it take to get a "good" solution? For standard metrics—such as parameter distance in $w^*$ or prediction error on $y$—the sample complexity of linear regression depends on properties of the distribution (such as the conditioning of $x$, or the variance of $y$) that could be unbounded and cannot be tested. For example, in the classic analysis (see [7], Chapter 3), sample complexity depends on the design matrix, which in turn results in a dependence on the expectation of $\|x\|_2$. The typical approach to deal with this would require bounded moments for $x$. For learning the conditional distribution, the more natural metric is total variation (TV) distance: the parameters $w$ induce a distribution $p_w(x, y) = p(x)p_w(y \mid x)$ where $p(x)$ is the *true* distribution of $x$, and we would like to find $\widehat{w}$ such that

$$d_{TV}(p_{w^*}(x, y), p_{\widehat{w}}(x, y)) \leq \varepsilon. \tag{1}$$

This ensures that when the input $x$ come from the user in the true unknown distribution $p(x)$, the model generates a conditional sample that is close in TV to the true model. It turns out that this goal, unlike parameter distance, *can* be solved with no assumption on the distribution.

**Theorem 1.1** (Informal version of Theorem 4.1). *The MLE (i.e., least squares regression) achieves* (1) *with $O(\frac{1}{\varepsilon^2} k \log \frac{1}{\varepsilon})$ samples, regardless of the distribution of $x$.*

To our knowledge, all previous guarantees for linear regression either require some assumptions on, or give guarantees in terms of, the distribution of $x$ or $y$. We avoid this dependence on the $x$ distribution by adopting a similar analysis to Theorem 11.2 in [20].

**Main Result: 1-Layer Networks.** Modern feed-forward conditional generative models (e.g., StyleGAN2) are more complicated than linear regression. They are stochastic neural networks with layers of the form:

$$y = \phi(W^*x + \eta) \qquad \text{for } \eta \sim \mathcal{N}(0, \Sigma^*) \tag{2}$$

for $x \in \mathbb{R}^k$, $y \in \mathbb{R}^d$, $\phi(x) = \max(x, 0)$ is the ReLU activation, and some weights $W^* \in \mathbb{R}^{d \times k}$, $\Sigma^* \in \mathbb{R}^{d \times d}$.

We show that 1-layer generative models of the form (2) can *also* be estimated using the MLE (which is concave), with small TV error and without any assumption on $x$.

**Theorem 1.2** (Informal version of Theorem 4.5). *Suppose $y$ is drawn according to* (2) *where $\Sigma^*$ has condition number at most $\kappa$. Using $O\left(\frac{kd+d^2}{\varepsilon^2} \log \frac{kd\kappa}{\varepsilon}\right)$ samples, the distribution generated by the MLE has TV error $\varepsilon$, regardless of the distribution of $x$.*

Like Theorem 1.1, Theorem 1.2 shows that a conditional generative model can be learned in TV regardless of the distribution of the input $x$. It generalizes Theorem 1.1 in three ways: (a) the output $y$ is $d$-dimensional rather than 1-dimensional; (b) the covariance of the noise $\Sigma^*$ is learned rather than specified; and (c) the ReLU nonlinearity $\phi$ imparts additional complex structure.

Indeed, the point (c) means that parameter distances are poor metrics for this problem. For example, when an element of $W^*x$ has a large negative bias, the corresponding coordinate in $y$ will almost

always be 0. This means sample pairs $(x_i, y_i)$ provide little information about $W^*$, but still provide useful information about the conditional distribution. As we will later see, exploiting this valuable information in the truncated samples allows us to significantly outperform prior works such as [35], which do not exploit these 0-valued samples.

**Multilayer Networks Given Activations.** Given our distribution-free results on 1-layer networks, it is possible to extend our results to deep multi-layer networks. Given access to the internal activations of a neural network (but not the weights), our results can be applied layer-wise. Intermediate layers may have poorly conditioned input distributions, but since our result does not depend on the input distribution, we achieve strong guarantees for layer-wise learning in Theorem 4.6.

## 1.1 Proof Approach

In this outline we focus on the case of 1-dimensional $y$, and and standard Gaussian noise $\eta \sim \mathcal{N}(0, 1)$. Our proof approach is inspired by learning bounds that exploit finite VC dimension. We would like to show (1), or equivalently,

$$d(w^*, \widehat{w}) := \mathbb{E}_x[d_{TV}(p_{w^*}(y \mid x), p_{\widehat{w}}(y \mid x))] \leq \varepsilon, \tag{3}$$

when we see $n$ samples $x_i$ of $x$, and one sample $y_i$ for each $x_i$. We do this in two stages. First, we show that the empirical distance between $w$ and $\widehat{w}$ is small i.e.,

$$\widetilde{d}(w^*, \widehat{w}) := \widetilde{\mathbb{E}}_x[d_{TV}(p_{w^*}(y \mid x), p_{\widehat{w}}(y \mid x))] \leq 0.5\varepsilon, \tag{4}$$

where $\widetilde{\mathbb{E}}_x[f(x)] = \frac{1}{n} \sum_{i=1}^n f(x_i)$ denotes the empirical expectation over $x$.

Second, we show that the empirical distance is a good proxy for the true distance, i.e.,

$$d(w^*, \widehat{w}) \leq \widetilde{d}(w^*, \widehat{w}) + 0.5\varepsilon \leq \varepsilon, \tag{5}$$

which gives (3).

**Linear Case.** In the linear case, both stages are straightforward. The linear regression solution has an explicit form, and it is well known and easy to show that

$$\widetilde{\mathbb{E}}(x^T(w^* - \widehat{w}))^2 \propto k/n.$$

Since $d_{TV}(p_{w^*}(y \mid x), p_{\widehat{w}}(y \mid x)) = \Theta(\min(1, |x \cdot (w^* - \widehat{w})|))$, Jensen's inequality implies (4) for $n > k/\varepsilon^2$.

Secondly, $f_w(x) := d_{TV}(p_{w^*}(y \mid x), p_w(y \mid x))$ is bounded and unimodal in $w$. Thus, it suffices to bound the deviation of the empirical average from the true $f_w(x)$ with Chernoff's inequality.

**ReLU Case.** In the ReLU case, we have $y = \phi(w^* \cdot x + \eta)$, and both stages of the previous analysis are more difficult.

The most interesting part of our proof is showing the first stage for the ReLU case, which states that the $\widehat{w}$ maximizing

$$L(w) := \frac{1}{n} \sum_{i=1}^n \log p_w(y_i \mid x_i)$$

satisfies (4). Now, for any $w$ not satisfying (4),

$$\mathbb{E}_y[L(w) - L(w^*)] = -\widetilde{\mathbb{E}}_x d_{KL}(p_{w^*}(y \mid x) \| p_w(y \mid x))$$

$$\leq -2\widetilde{\mathbb{E}}_x[d_{TV}(p_{w^*}(y \mid x), p_w(y \mid x))^2] \leq -2\varepsilon^2,$$

where the first inequality follows from Pinsker's inequality. Unfortunately, $L(w) - L(w^*)$ does not concentrate well, by virtue of the KL-divergence being unbounded. However, we can upper bound it via the Bernstein inequality, such that for a fixed $w$ not satisfying (4), and given $n = \frac{1}{\varepsilon^2} \log(\frac{1}{\delta})$ samples, we have

$$L(w) - L(w^*) \leq -\varepsilon^2,$$

with probability $1 - \delta$. Using a careful covering argument and $n = (k/\varepsilon^2)\log(1/\delta)$ samples, we can uniformly extend this to *all* $w$ not satisfying Eq (4). By definition, the MLE has $L(\widehat{w}) \geq L(w^*)$, and by our uniform bound, it must satisfy (4).

The second stage changes because $f_w(x)$ depends on $x \cdot w$ and $x \cdot w^*$ in a more complicated way than through $x \cdot (w - w^*)$. This makes showing a bounded VC dimension more difficult; however, unpacking the proof that VC implies generalization, we can still show that the net (normally given by Sauer's lemma) is bounded. This generalization holds as long as $f_w(x)$ is unimodal in $x \cdot w$.

## 2  Contributions

1. We show that MLE can perform distribution learning in the setting of linear regression and multi-layer ReLU networks. Our bounds do not make assumptions on the distribution of $x$ or the condition number of $W^*$, and achieve a sample complexity polynomial in the system parameters.

2. We improve the sample complexity bound in [35], which estimates the parameters of a one-layer ReLU network but suffers an exponential dependence on the $W^*$ term. In contrast, as we seek to estimate the *distribution* of $(x, y)$, rather than the parameter $W^*$, we are able to avoid this. See Section 4.2 for more details.

3. Our algorithm for learning multi-layer ReLU networks is considerably simpler than [1], who learn discriminators that are engineered to perform moment-matching on the output of each layer of the network. Furthermore, [1] impose a strong requirement on the sparsity and independence of the activations at each layer, which essentially allows standard techniques in sparse coding to recover these activations.

## 3  Related Work

Generative adversarial networks (GANs) [19, 2, 30] are a popular family of generative models that train a generator and discriminator in an adversarial training framework. The seminal result by [22] proposed *progressive* growing of GANs (PGGANs) as a way to stabilize and accelerate the training phase of these models. Future results, such as StyleGAN [23] introduce more complicated architectures and "style" variables. Additionally, these models add noise at each layer of the generator in order to introduce greater stochasticity in the generated images, which is important for textures such as hair and skin.

**Distributional Learning**   Most theoretical results have focused on the min-max optimality of GANs [15, 28, 29], characterizing their stationary points [21, 17], or characterizing their generalization once they have reached a global minimum [3, 5]. The closest result to ours is [35]. Setting $x$ to a deterministic scalar in our problem statement reduces it to [35], who consider $y = \phi(b + \eta)$ s.t. $b, \eta \in \mathbb{R}^d$, and they seek to learn the covariance of $\eta$ along with the bias vector $b$. However, their sample complexity bound suffers an exponential dependence on $\|b\|_\infty^2$.

Single layer networks have attracted recent attention, as they provide a tractable formulation for studying the dynamics and generalization of adversarial learning [24, 18, 25, 11]. The recent results of [1] show that multi-layered models that satisfy a property known as *forward super-resolution* (such as PGGANs) can be learned in polynomial time and sample complexity using stochastic gradient descent-ascent. In this case, the discriminator is designed to detect differences between higher order moments of the generated and training distribution. Deep models have also been considered in [12, 10].

## 4  Main Results

In this section we first show that the MLE of the parameters learns the input-output joint distribution for linear regression. Then, we extend this guarantee to the case where the ReLU activation function $\phi$ is applied to the multi-dimensional output $y$. Finally, we show that we can compose the 1-layer ReLU guarantee to learn the distribution generated by a multi-layer model.

## 4.1 Linear Regression

We begin with the classic linear regression problem of learning the parameter $w^*$ from a linear model

$$y = x \cdot w^* + \eta, \tag{6}$$

where $\eta \sim \mathcal{N}(0, \sigma^2)$, $w \in \mathbb{R}^k$, $\sigma$ is known and $x \in \mathbb{R}^k$ with some distribution. This problem has been studied for centuries. Our novelty is that we view (6) as a conditional generative process, and instead of studying error in Euclidean distance in parameter space, i.e., minimizing $\|\widehat{w} \cdot x - w^* \cdot x\|_2$, we focus on error in $d(w^*, \widehat{w})$ as defined in (3), which only captures the error in $\widehat{w}$ insofar as it impacts our distribution estimate. Given data $\{(x_i, y_i)\}_{i=1}^n$ generated by (6), the MLE is:

$$\widehat{w} := \arg\max_{w \in \mathbb{R}^k} \sum_{i \in [n]} \log p_w(y_i | x_i) \equiv \arg\min_{w \in \mathbb{R}^k} \sum_{i \in [n]} \frac{(y_i - w \cdot x_i)^2}{\sigma^2}.$$

The following theorem establishes that the MLE is close in TV distance. The proofs for all results in this section are in Appendix A.

**Theorem 4.1.** *Let $\{(x_i, y_i)\}_{i=1}^n$ be i.i.d. random variables generated from the linear model (6), and assume that $\sigma$ is known. Then, for a sufficiently large constant $C > 0$,*

$$n = C \frac{k}{\varepsilon^2} \log \frac{1}{\varepsilon}$$

*samples suffice to ensure that with probability $1 - e^{-\Omega(\varepsilon^2 n)}$ over the data,*

$$d(\widehat{w}, w^*) \leq \varepsilon.$$

Note that one cannot hope to get such a guarantee in the classical setting where error is measured in $\|\widehat{w} \cdot x - w^* \cdot x\|_2$ without additional assumptions on the distribution of $x$ because if $x$ is badly conditioned, the error may be dominated by very rare directions of $x$ that we never sample. For example, the bounds in [7], Chapter 3, require the second moment of $x$ to be bounded.

Since we only wish to learn the distribution of $y$ in total variation, no single $x$ can contribute much to our loss and we get a distribution-free result. This is possible as the total variation distance is bounded, and we can invoke Theorem 11.2 in Györfi et al [20].

We now state two lemmas needed to prove Thereom 4.1, assuming without loss of generality that $\sigma^2 = 1$. We split the proof into two stages. In the first stage, we bound $\widetilde{d}(\widehat{w}, w^*)$, which denotes the empirical TV distance (4) over the training set $\{x_i\}_{i=1}^n$.

**Lemma 4.2.** *Let $\{(x_i, y_i)\}_{i=1}^n$ be i.i.d. random variables such that $y_i = x_i \cdot w^* + \mathcal{N}(0, 1)$. Then, for $n \geq \frac{k}{2}$, with probability $1 - e^{-\Omega(n)}$, the MLE $\widehat{w}$ satisfies*

$$\widetilde{d}(\widehat{w}, w^*) \leq \sqrt{\frac{k}{2n}}.$$

The proof relies on the fact that $p_{w^*}(y | x_i)$ and $p_{\widehat{w}}(y | x_i)$ are Gaussian distributions, so

$$d(p_{\widehat{w}}(y | x_i), p_{w^*}(y | x_i)) = \Theta(\min\{1, |x_i^T(\widehat{w} - w^*)|\}).$$

Using the explicit form of the MLE, we can show that, with high probability,

$$\frac{1}{n} \sum_i \left( x_i^T(\widehat{w} - w^*) \right)^2 \leq \frac{k}{2n},$$

and Lemma 4.2 follows from Jensen's inequality.

The second stage shows that the empirical average of the TV distance $\widetilde{d}(\widehat{w}, w^*)$ is close to the population average $d(\widehat{w}, w^*)$.

**Lemma 4.3.** *Let $\{x_i\}_{i=1}^n$ be i.i.d. random variables such that $x_i \sim \mathcal{D}_x$. For a sufficiently large constant $C > 0$, and for $n = C \frac{k}{\varepsilon^2} \log \frac{1}{\varepsilon}$ with $n \geq \frac{k}{2}$, we have:*

$$\Pr_{x_i \sim \mathcal{D}_x} \left[ \sup_{w \in \mathbb{R}^k} \left| \widetilde{d}(w, w^*) - d(w, w^*) \right| > \varepsilon \right] \leq e^{-\Omega(n\varepsilon^2)}.$$

Note the probability in the above statement is with respect to the distribution of $x$, and does not depend on $y$. The proof follows Theorem 11.2 in Györfi et al [20]: it relies on the fact that the TV distance is bounded and a unimodal function of $w \cdot x$. This implies that for each $x_i$, we are able to partition the space of $w$ with $O(1/\varepsilon)$ hyperplanes such that within each cell the TV distance varies by at most $\varepsilon$. As we have $n$ samples and $w \in \mathbb{R}^k$, the number of cells induced in $\mathbb{R}^k$ is $\propto (n/\varepsilon)^k$, and it is sufficient to provide concentration bounds for one representative in each cell. This approach is similar to bounding the VC dimension of a set of binary functions. Setting $n = \Theta(\frac{k}{\varepsilon^2} \log \frac{1}{\varepsilon})$ and combining Lemma 4.2 with Lemma 4.3 gives $d(\widehat{w}, w^*) \leq \varepsilon$.

## 4.2 ReLU Case

Now consider the single-layer ReLU. We observe $(x, y) \in \mathbb{R}^k \times \mathbb{R}^d$ such that:
$$y = \phi(W^*x + \eta), \quad \eta \sim \mathcal{N}(0, \Sigma^*), \tag{7}$$
where $\eta \in \mathbb{R}^d$, $W^*$ and $\phi(\cdot) = \max(\cdot, 0)$ is applied coordinate-wise. The matrices $W^* \in \mathbb{R}^{d \times k}$ and $\Sigma^* \in \mathbb{R}^{d \times d}$ are *unknown*, and we do not observe $\eta$. The variable $x$ is drawn from an arbitrary probability distribution $\mathcal{D}_x$, and we make *no additional assumptions* on $\mathcal{D}_x$: this is important, as we will progressively cascade layers, and one should think of $\mathcal{D}_x$ being the distribution of activations at each layer.

Given a sample $(x, y) \in \mathbb{R}^k \times \mathbb{R}^d$, let $S$ denote the co-ordinates of $y$ that are zero-valued, and let $S^c$ denote the compliment of $S$. Then, the log-likelihood of $W, \Sigma$, on this sample is given by

$$\log p_{W,\Sigma}(y|x) = c - \frac{\log|\Sigma|}{2} + \log \int_{\substack{t_S \leq 0 \\ t_{S^c} = y_{S^c}}} \exp\left\{-\frac{(t - Wx)^T \Sigma^{-1} (t - Wx)}{2}\right\} dt_S. \tag{8}$$

where $c$ is a normalization constant which does not depend on $W$ or $\Sigma$. This function is a mixed density: in the coordinates of $y$ that are 0, i.e., in the set $S$, we integrate the Gaussian density over the negative orthant, as $W^*x + \eta$ could have been any negative value in those coordinates. In Lemma F.1 in the Appendix, we show that Eqn (8) is concave after an invertible reparameterization of $W, \Sigma$.

In this setting, proving an analogue of Theorem 4.1 poses multiple challenges:

- The output $y$ is $d$-dimensional rather than a scalar, and $\eta$ in Eqn (7) introduces correlations between the coordinates of $W^*x$, such that we cannot decompose the log-likelihood in Eqn (8) per coordinate.

- We do not know the covariance matrix $\Sigma^*$, and it must be estimated.

- Lemma 4.2 requires the explicit form of the MLE in linear regression. In the absence of such a closed-form solution, we need to directly analyze the log-likelihood, which is a mixed density and involves integrating the Gaussian likelihood over the zero-valued coordinates of $y$. In order to handle this, we use the Györfi approach again on the log-likelihood. This is challenging because the variables we concentrate are KL-divergences, which are unbounded and require Bernstein type inequalities.

- Recovering the true parameters $W^*, \Sigma^*$ is difficult: if we see a zero in $y$, we do not know its magnitude in $W^*x + \eta$ before the ReLU. This manifests in the results in [35], where it is assumed that each entry in $W^*$ is positive – otherwise, their sample complexity scales as $e^{\|W\|_\infty^2}$.

Nonetheless, we can handle most of these difficulties, and the only assumption we make is that the condition number of $\Sigma^*$ is bounded and known to our estimator.

**Assumption 4.4** (Condition number bound). *Let $\lambda_{\max}^*, \lambda_{\min}^*$ denote the largest and smallest singular values of $\Sigma^*$. We assume there exists $\kappa < \infty$ such that $\frac{\lambda_{\max}^*}{\lambda_{\min}^*} \leq \kappa$. We further assume that the value of $\kappa$ is known to our estimator.*

Note that the condition number only allows us to control the correlation between the coordinates of $W^*x + \eta$. The other challenges introduce by the ReLU, such as the lack of a closed form MLE, the need to estimate $\Sigma^*$ and a mixed density log-likelihood remain.

Under Assumption 4.4, the following theorem shows that the MLE $\widehat{W}, \widehat{\Sigma}$ achieves a small total variation distance. The proof of this theorem is in Appendix C.

**Theorem 4.5.** *Let $\mathbb{R}_\kappa^{d \times d}$ denote the set of positive definite matrices with condition number $\kappa$. Given $n$ samples $\{(x_i, y_i)\}_{i=1}^n$ satisfying Assumption 4.4, where $x_i \sim \mathcal{D}_x$ i.i.d., and $y_i$ is generated according to (7), let $\widehat{W}, \widehat{\Sigma} := \arg\max_{W \in \mathbb{R}^{d \times k}, \Sigma \in \mathbb{R}_\kappa^{d \times d}} \frac{1}{n} \sum_i \log p_{W,\Sigma}(y_i \mid x_i)$. Then, for a sufficiently large constant $C > 0$,*

$$n = C \cdot \left( \frac{kd + d^2}{\varepsilon^2} \right) \log \left( \frac{\kappa k d}{\varepsilon \delta} \right)$$

*samples suffice to ensure that with probability $1 - \delta$, we have*

$$d_{TV}\left( (\widehat{W}, \widehat{\Sigma}), (W^*, \Sigma^*) \right) \leq \varepsilon.$$

**Comparison to [35].** Our result is closely related to [35]. Our ReLU model reduces to their model by setting $W^* \in \mathbb{R}^d$ as a vector, and $x = 1$. In order to learn the distribution of $y$, they first estimate the parameters $W^*, \Sigma^*$, in $\ell_2$ norm and then convert the $\ell_2$ error to a TV error. The parameter recovery is done per coordinate of $W^*, \Sigma^*$, by performing MAP estimation on the positive samples in $y$. This crucially assumes that each coordinate has enough positive samples, and to that end, they assume that each entry in $W^*$ is positive—otherwise, their sample complexity scales as $e^{\|W^*\|_\infty^2}$. Our results *do not make any assumptions* on $W^*$ and handles a wider class of matrix valued $W^*$. Additionally, the objective function (8) does not discard the zero valued samples in $y$, making it more sample efficient.

We assumed that the covariance matrix $\Sigma^*$ has condition number $\kappa$, and our sample complexity scales as $\log \kappa$. Hence, even if $\kappa = e^{\mathrm{poly}(d,k)}$, we only pay a $\mathrm{poly}(d, k)$ penalty. This improves on the result in [35], where the sample complexity scales as $\kappa^2$. While our statistical guarantees are strictly better, [35] gives poly-time and poly-space algorithmic guarantees for their estimator. We discuss the empirical limitations of our algorithm in Section 6.

**Lower Bounds** Ignoring log factors, the complexity factor of $kd$ is obviously required. Furthermore, learning a Gaussian with unknown covariance matrix in total variation takes $\tilde{\Omega}(d^2)$ samples; see [6]. Our Theorem 4.5 would solve their lower bound instance, the same lower bound applies to our problem.

**Extension to Multi-Layer Generative Models.** Consider the following $(L+1)$-layered generative model.

$$x_{L+1} = W_L^* x_L + \eta_L, \quad \text{where} \quad x_\ell = \phi(W_{\ell-1}^* x_{\ell-1} + \eta_{\ell-1}) \in \mathbb{R}^{d_\ell} \quad \forall \quad \ell \in [1, L], \quad (9)$$

$$x_0 \sim \mathcal{D}_0 \quad \text{and} \quad \eta_\ell \sim \mathcal{N}(0, \Sigma_\ell^*) \quad \forall \quad \ell \in [0, L]. \quad (10)$$

We can compose the guarantees provided by Theorem 4.1 and 4.5 to show that we can learn this model.

**Theorem 4.6.** *Given $n$ i.i.d. samples of $(x_0, \ldots, x_{L+1})$, such that each matrix $\Sigma_\ell^*$ satisfies Assumption 4.4, let $\widehat{W}_\ell, \widehat{\Sigma}_\ell$, be the MLE estimates of $W_\ell^*, \Sigma_\ell^*$ learned from samples of $(x_\ell, x_{\ell+1})$. Define $m := \max_\ell d_\ell^2$. Then,*

$$n = O\left( \frac{l^2 m}{\varepsilon^2} \log\left( \frac{lm\kappa}{\varepsilon\delta} \right) \right),$$

*samples suffice to ensure that with probability $1 - \delta$,*

$$d_{TV}\left( \{\widehat{W}_\ell, \widehat{\Sigma}_\ell\}_{\ell=0}^L, \{(W_\ell^*, \Sigma_\ell^*)\}_{\ell=0}^L \right) \leq \varepsilon.$$

**Comparison to [1].** The modelling assumptions in [1] are similar to ours – the authors learn a generative model per layer, using images produced per layer. The key differences of their model are: (i) each layer is deterministic (there is no $\eta_\ell$), (ii) their learning algorithm does not require access to the activations of each layer, (iii) their algorithm performs moment-matching by crafting the discriminator strategically.

In order to avoid requiring activations at each layer, [1] imposes a sparsity assumption on the activations: this allows them to leverage tools from existing results in sparse coding [4], such that

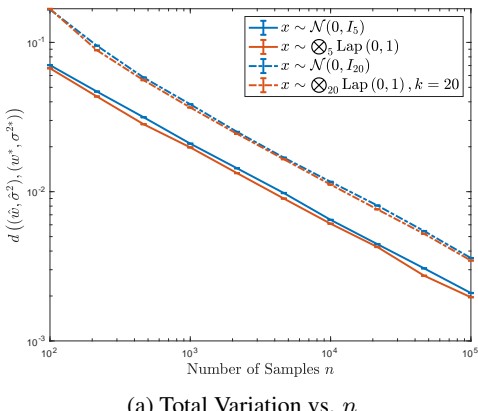 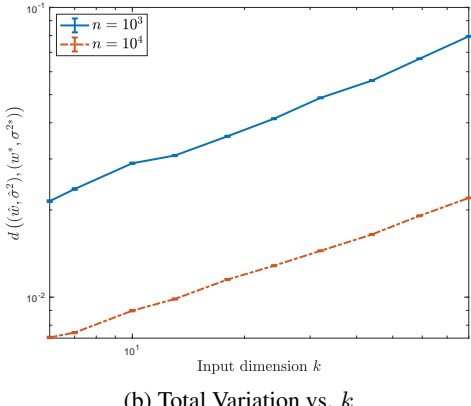

(a) Total Variation vs. $n$          (b) Total Variation vs. $k$

Figure 2: (a) Plot of TV distance vs. $n$. $\sigma^2 = 1$, $w^* = \mathbb{1}_{k \times 1}$, for two different values of $k = 5$ and $k = 20$. Plot includes data for two different distributions of $x$. Note that distribution has little impact on TV distance, and in both cases, we see the error decreasing with a slope of $-1/2$ in alignment with our theory. (b) Plot of TV distance vs. input dimension $k$. For both $n = 10^3$ and $n = 10^4$, the error grows with a slope of roughly $1/2$, in alignment with our theory. In both plots 2000 runs are used to compute the mean. Error bars represent 95% confidence intervals.

sparse activations at each layer can be recovered using images produced by the layer. This assumption can be somewhat strong, as it implies that the activations are roughly independent of one another, and the sparsity remains constant over layers, despite the layers themselves expanding by a factor of 4, i.e., $d_\ell \geq 4d_{\ell-1} \; \forall \; \ell \leq L - 1$.

## 5    Simulations

We now numerically verify our theoretical claims and compare against other approaches. A detailed description of simulation methods are included in the appendix. Our code is available at https://github.com/basics-lab/learningGenerativeModels.git.

### 5.1    Scaling in $n$ and $k$

Figure 2 numerically investigates how TV distance of the MLE scales with the number of samples $n$ and the input dimension $k$. We consider a model with 1-dimensional output and a $k$-dimensional input: $y = \phi(x \cdot w^* + \eta)$, for $w^* \in \mathbb{R}^k$ and $\eta \sim \mathcal{N}(0, \sigma^2)$. We set $\sigma^2 = 1$ and $w^* = \mathbb{1}_{k \times 1}$, both unknown to the optimizer, which has samples $(y_i, x_i)_{i=1}^n$. Figure 2a, which plots the error in TV distance against $n$ on a log-log plot, has a slope of roughly $-1/2$ as predicted by our theory. Similarly, Figure 2b has a slope of $1/2$, which is in line with our theory on scaling with respect to input dimension $k$. We defer simulations involving scaling in $d$ to the appendix because computing TV distance becomes increasingly difficult as $d$ becomes large, and we must resort to using upper bounds.

### 5.2    Distribution Independence

The fact that our guarantee does not depend on the distribution of $x$ suggests that the expected TV error of the distribution learned from the MLE may be similar for all distributions over $x$. To test this we consider both $x \sim \mathcal{N}(0, I_k)$ and $x \sim \bigotimes_k \mathrm{Lap}(0, 1)$, i.e., each element of $x$ is drawn independently standard Laplace. Figure 2a verifies our hypothesis, showing only very slight differences in our observed empirical average TV error between the two distributions over a wide range of $n$, and for $k = 5$ and $k = 20$.

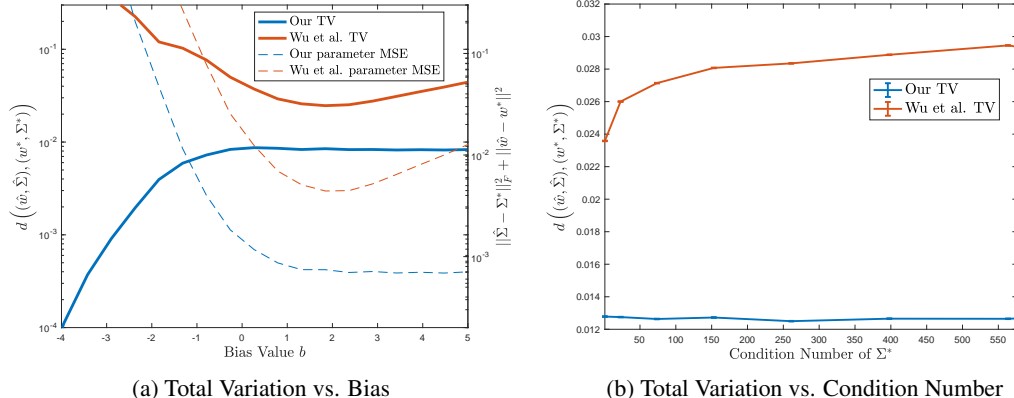

| (a) Total Variation vs. Bias | (b) Total Variation vs. Condition Number |

Figure 3: (a) Left hand axis shows TV distance vs. bias vector $b$ with $y = \phi(\eta + b\mathbb{1}_{d\times 1})$, $d = 3$, $\eta = \mathcal{N}(0, \Sigma)$, and $\Sigma = I_d$. Note that MLE (blue) has error going to zero as bias becomes negative, while the opposite is true for the baseline (red). Right hand axis shows the mean-squared error of the parameters $\Sigma$ and the mean $\mu$, each point was run a total of 2000 times. (b) TV distance vs. condition number, $d = 3$. MLE does not exhibit trend with condition number, but baseline does. Error bars are 95% confidence intervals, over 20000 runs.

### 5.3 Scaling with Bias and Condition Number of Covariance Matrix

A key feature of the MLE is that it makes use of truncated samples. This is in contrast to [35], which leverages results on learning truncated normal distributions [14] where truncated samples are not observed. This leads to a stark difference in performance as the number of truncated samples becomes large. To show this, we consider a model with a $d$-dimensional output and 1-dimensional input. We let $x = 1$ almost surely, and then take $w^* = b\mathbb{1}_{d\times 1}$ for some *bias* $b \in \mathbb{R}$ and $\eta = \mathcal{N}(0, \Sigma)$ with $\Sigma = I_d$, thus $y = \phi(\eta + b\mathbb{1}_{d\times 1})$. As $b$ becomes more negative, the number of truncated samples increases. Figure 3a shows the differing behavior of MLE and that of [35] as $b$ becomes negative. For ease of computing the TV distance we set $d = 3$, and restrict optimization over diagonal $\Sigma$. The solid blue line depicts the performance of the MLE. We observe that the TV error is constant for $b > 0$ and begins to decrease rapidly for $b < 0$. This happens because as $b$ becomes more negative, the truncation places more probability mass at $y = 0$. Indeed, the dashed blue lines indicate that even as the TV error is decreasing rapidly, the mean square estimation error of the covariance and mean increase, however, since most of the probability mass is at zero, this does not significantly impact the TV. In contrast, the method of [35] rapidly deteriorates as $b < 1$ as the number of untruncated samples decreases. We also point out that even when the bias is large, [35] is still significantly worse. We attribute this to the fact that even when there is no truncated samples, [35] is still minimizing a different MAP objective. More discussion of this is provided in the appendix.

**Robustness to Condition Number of $\Sigma$.** Another concern is how TV error scales as a function of the condition number of $\Sigma^*$. Poorly conditioned $\Sigma^*$ can put significant probability masses on small sets, and potentially cause large error. We consider a similar environment to the one described above, but fix $b = 1$ and alter diagonal entries of $\Sigma^*$ such that one entry is $\sqrt{\kappa}$, another is $\sqrt{\kappa^{-1}}$ and the rest are 1, making the condition number $\kappa$. Figure 3b shows that the MLE is not measurably impacted by the changing condition number over the range plotted. This is not true of [35], where we observe that TV does grow with condition number.

## 6   Limitations

This work is only a first step to understanding fundamental limits of learning generative models. We showed that our theorems for single-layer networks can be composed to get sample complexity bounds on deep networks, with a critical caveat: we require access to not just the input and output pairs, but also the intermediate activations. This is not practical in many scenarios, and removing this restriction will be an important direction for future research. Beyond this, we assume that the learner has an understanding of the model architecture. In many cases, however, a learner may not be aware

of the number of layers a network has or a vast number of other architectural details. Additionally, we inherit known problems with the MLE, such as exacerbation of biases that exist in the training data (Chapter 24.1.3 in [33]).

Our results place emphasis on sample complexity over *computational complexity*. Though we show that the MLE problem is concave, this work does not provide a thorough analysis of the optimization problem. It is possible that similar results to [14, 35] can be derived for the MLE problem. Indeed, empirically, we find that a similar projected stochastic gradient ascent performs well in our problem. A careful analysis must consider factors like the distribution over $x$, the condition number of $\Sigma^*$ and the truncation probability, all of which are likely to impact the optimization.

# 7 Conclusions

We have studied the problem of learning conditional generative models from a limited number of samples. We have shown that it is possible to learn a 1-layer ReLU conditional generative model in total variation, with no assumption on the distribution of the conditioning variable $x$ using the MLE. We have also shown that this result can be extended to multi-layer ReLU networks, given access to the internal activations. Our results suggest that MLE is a promising approach for learning feed-forward generative models from limited samples.

# 8 Acknowledgements

Ajil Jalal and Kannan Ramchandran are supported by ARO 051242-002. Justin Kang and Kannan Ramchandran are supported by NSF EAGER Award 2232146. Eric Price is supported by NSF awards CCF-2008868, CCF-1751040 (CAREER), and the NSF AI Institute for Foundations of Machine Learning (IFML).

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

# A  Proofs of Linear Case

Throughout the appendix, for ease of notation, we overload the definition of the function $d_{TV}(\cdot, \cdot)$. When inputs are random variables, it represent the TV distance between the distributions of those random variables.

**Lemma 4.2.** *Let $\{(x_i, y_i)\}_{i=1}^n$ be i.i.d. random variables such that $y_i = x_i \cdot w^* + \mathcal{N}(0, 1)$. Then, for $n \geq \frac{k}{2}$, with probability $1 - e^{-\Omega(n)}$, the MLE $\widehat{w}$ satisfies*

$$\widetilde{d}(\widehat{w}, w^*) \leq \sqrt{\frac{k}{2n}}.$$

The proof of this lemma requires Lemma A.1, which characterizes the distribution of the residual error of the MLE.

**Lemma A.1.** *Given $y \in \mathbb{R}^n, X \in \mathbb{R}^{n \times k}$ satisfying $y = Xw^* + \eta$, where $\eta \sim \mathcal{N}(0, \sigma^2 I_n)$, the least square solution $\widehat{w}$ satisfies*

$$Xw^* - X\widehat{w} \sim \mathcal{N}(0, \sigma^2 X(X^TX)^{-1}X^T) \Rightarrow \mathbb{E}[\|X\widehat{w} - Xw^*\|^2] = \sigma^2 k.$$

*Proof.* The least squares solution is given by

$$\begin{aligned}
\hat{w} &= (X^TX)^{-1}X^Ty, \\
&= (X^TX)^{-1}X^T(Xw^* + \eta), \\
&= w^* + (X^TX)^{-1}X^T\eta.
\end{aligned}$$

Multiplying on the left by $X$, we have

$$X\hat{w} = Xw^* + X(X^TX)^{-1}X^T\eta.$$

Since $\eta$ is i.i.d. Gaussian with variance $\sigma^2$, we have,

$$\begin{aligned}
X(X^TX)^{-1}X^T\eta &\sim \mathcal{N}(0, \sigma^2 X(X^TX)^{-1}X^TX(X^TX)^{-1}X^T) \\
&\sim \mathcal{N}(0, \sigma^2 X(X^TX)^{-1}X^T)
\end{aligned}$$

This implies

$$\begin{aligned}
\mathbb{E}[\|X\hat{w} - Xw^*\|^2] &= \sigma^2 \mathrm{Tr}[X(X^TX)^{-1}X^T], \\
&= \sigma^2 \mathrm{Tr}[(X^TX)^{-1}X^TX], \\
&= \sigma^2 k.
\end{aligned}$$

$\square$

*Proof of Lemma 4.2.* The KL divergence between two Gaussians $P = \mathcal{N}(\mu_1, \Sigma)$ and $Q = \mathcal{N}(\mu_2, \Sigma)$ is:

$$d_{KL}(P\|Q) = \frac{1}{2}(\mu_1 - \mu_2)\Sigma^{-1}(\mu_1 - \mu_2).$$

By Pinsker's inequality, this implies

$$d_{TV}(P\|Q) \leq \min\left\{1, \frac{1}{2}\sqrt{(\mu_1 - \mu_2)\Sigma^{-1}(\mu_1 - \mu_2)}\right\}.$$

Hence, the empirical TV on the dataset can be bounded by

$$\frac{1}{n}\sum_i d_{TV}(p_{\widehat{w}}(y|x_i), p_{w^*}(y|x_i)) \le \frac{1}{n}\sum_i \min\left\{1, \frac{1}{2}\frac{|x_i^T(\widehat{w}-w^*)|}{\sigma}\right\},$$

$$\le \sqrt{\frac{1}{n}\sum_i \min\left\{1, \frac{1}{2}\frac{|x_i^T(\widehat{w}-w^*)|}{\sigma}\right\}^2},$$

$$\le \sqrt{\min\left\{1, \frac{1}{4n}\sum_i \frac{(x_i^T(\widehat{w}-w^*))^2}{\sigma^2}\right\}},$$

$$= \sqrt{\min\left\{1, \frac{1}{4n}\frac{1}{\sigma^2}\|X(\widehat{w}-w^*)\|^2\right\}}.$$

where the second line follows from Jensen's inequality.

By Lemma A.1, we have

$$\mathbb{E}[\|X(\widehat{w}-w^*)\|^2] = \sigma^2 k.$$

which implies that with probability $1 - e^{-\Omega(n)}$, we have

$$\|X(\widehat{w}-w^*)\|^2 \le 2\sigma^2 k.$$

Substituting in the earlier inequality, we get

$$\frac{1}{n}\sum_i d_{TV}(p_{\widehat{w}}(y|x_i), p_{w^*}(y|x_i)) \le \sqrt{\min\left\{1, \frac{k}{2n}\right\}} = \sqrt{\frac{k}{2n}} \text{ for } n \ge \frac{k}{2}.$$

$\square$

**Lemma 4.3.** *Let* $\{x_i\}_{i=1}^n$ *be i.i.d. random variables such that* $x_i \sim \mathcal{D}_x$. *For a sufficiently large constant* $C > 0$, *and for* $n = C\frac{k}{\varepsilon^2}\log\frac{1}{\varepsilon}$ *with* $n \ge \frac{k}{2}$, *we have:*

$$\Pr_{x_i \sim \mathcal{D}_x}\left[\sup_{w \in \mathbb{R}^k}\left|\widetilde{d}(w, w^*) - d(w, w^*)\right| > \varepsilon\right] \le e^{-\Omega(n\varepsilon^2)}.$$

*Proof.* The proof is inspired by Theorem 11.2 in [20], with modifications to our setting.

Let Since $f_w(x)$ is bounded, for any fixed $w$, the Chernoff bound gives

$$\Pr\left[\left|\widetilde{d}(w, w^*) - d(w, w^*)\right| > \alpha\right] \le e^{-2n\alpha^2}. \tag{11}$$

for any $\alpha > 0$. The challenge lies in constructing a "net" to be able to union bound over $\mathbb{R}^k$ without assuming any bound on $w$ or the covariate $x$. A net is a partitioning of an space, where within each part, points are close together in some way. In this case, we construct a net using what we will refer to as "ghost" samples.

**Ghost samples.** First, we construct a "ghost" dataset $D'_x$ consisting of $n$ new samples, drawn i.i.d. $\{x'_i\}_{i\in[n]}$ of $\mathcal{D}_x$. This gives another metric $\widetilde{d}'(\cdot,\cdot)$. Instead of directly considering the distance between $\widetilde{d}(w, w^*)$ and $d(w, w^*)$, it is sufficient to consider the difference between $\widetilde{d}(w, w^*)$ and $\widetilde{d}'(w, w^*)$ i.e.,

$$\Pr\left[\sup_w\left|d(w, w^*) - \widetilde{d}(w, w^*)\right| > \varepsilon\right] \le 2\Pr\left[\sup_w\left|\widetilde{d}(w, w^*) - \widetilde{d}'(w, w^*)\right| > \varepsilon/2\right]. \tag{12}$$

To see this, let $\bar{w}$ maximize $\widetilde{d}(w, w^*) - \widetilde{d}'(w, w^*)$. Since $\bar{w}$ and $\{x'_i\}_{i\in[n]}$ are independent, by the Chernoff bound,

$$\Pr\left[\left|\widetilde{d}'(\bar{w}, w^*) - d(\bar{w}, w^*)\right| > \varepsilon/2 | D_x\right] \le e^{-n\varepsilon^2/2} \le 1/2.$$

for any $(D_x, \bar{w})$ and large enough $n$. Thus,

$$\Pr\left[\left|\widetilde{d}'(\bar{w}, w^*) - \widetilde{d}(\bar{w}, w^*)\right| > \varepsilon/2\right] \geq \Pr\left[\left|d(\bar{w}, w^*) - \widetilde{d}(\bar{w}, w^*)\right| > \varepsilon \cap \left|d(\bar{w}, w^*) - \widetilde{d}'(\bar{w}, w^*)\right| < \varepsilon/2\right]$$

$$= \mathop{\mathbb{E}}_{D_x}\left[1_{\{|d(\bar{w}, w^*) - \widetilde{d}(\bar{w}, w^*)| > \varepsilon\}} \Pr\left[\left|d(\bar{w}, w^*) - \widetilde{d}'(\bar{w}, w^*)\right| < \varepsilon/2 | D_x\right]\right]$$

$$\geq (1 - 1/2) \Pr\left[\left|d(w, w^*) - \widetilde{d}(w, w^*)\right| > \varepsilon\right],$$

which implies (12).

**Symmetrization.** Since $D_x$ and $D'_x$ each have $n$ independent samples, we could instead draw the datasets by first sampling $2n$ elements $x_1, \ldots, x_{2n}$ from $\mathcal{D}_x$, then randomly partition this sample into two equal datasets. Let $s_i \in \{\pm 1\}$ so $s_i = 1$ if $z_i$ lies in $D'_x$ and $-1$ if it lies in $D_x$. Then

$$\widetilde{d}'(\bar{w}, w^*) - \widetilde{d}(\bar{w}, w^*) = \frac{1}{n} \sum_{i=1}^{2n} s_i \cdot d_{TV}(p_w(y|x_i), p_{w^*}(y|x_i)).$$

For a fixed $w$ and $x_1, \ldots, x_{2n}$, the random variables $(s_1, \ldots, s_{2n})$ are a permutation distribution, so negatively associated. Then the variables $s_i \cdot d_{TV}(p_w(y|x_i), p_{w^*}(y|x_i))$ are monotone functions of $s_i$, so also negatively associated. They are also bounded in $[-1, 1]$. Hence we can apply a Chernoff bound:

$$\Pr\left[\left|\widetilde{d}'(\bar{w}, w^*) - \widetilde{d}(\bar{w}, w^*)\right| > \varepsilon\right] < e^{-n\varepsilon^2/2} \tag{13}$$

for any fixed $w$.

**Constructing a net.** We partition $\mathbb{R}^k$ the space of $w$ s.t. if $w, w'$ are in the same partition then,

$$\left|d_{TV}(p_w(y|x), p_{w^*}(y|x)) - d_{TV}(p_{w'}(y|x), p_{w^*}(y|x))\right| < \alpha.$$

for each $x$ in the dataset $x_1, \ldots, x_{2n}$. Then take the intersection of all $2n$ partitions to construct a net over $\mathbb{R}^k$.

As the total variation distance is a unimodal function of $x_i \cdot w - x_i \cdot w^*$, we partition $w$ the sets

$$\{w : d_{TV}(p_w(y|x_i), p_{w^*}(y|x_i)) \in [j\alpha, (j+1)\alpha]$$

where $j$ goes from 0 to $1/\alpha - 1$. So the space of $w$, $\mathbb{R}^k$ is partitioned by $2n$ sets of $1/\alpha$ parallel hyper-planes. Then the total number of cells is at most

$$\sum_{i=0}^{k} \binom{2n}{i} (2/\alpha)^i \leq 2\left(\frac{4en}{\alpha k}\right)^k$$

We define a net $N$ by choosing one representative of each cell in the partition, so $|N| \leq e^{2k \log \frac{n}{\alpha k}}$. By (13),

$$\Pr\left[\max_{w \in N}\left|\widetilde{d}'(\bar{w}, w^*) - \widetilde{d}(\bar{w}, w^*)\right| > \varepsilon\right] < |N| e^{-n\varepsilon^2/2} \leq e^{2k \log \frac{n}{\alpha k} - \varepsilon^2 n/2}.$$

Finally, for any $w \in \mathbb{R}^d$ let $\bar{w} \in N$ be the representative of its cell. By definition of the cells,

$$|d_{TV}(p_w(y|x_i), p_{w^*}(y|x_i)) - d_{TV}(p_{\bar{w}}(y|x_i), p_{w^*}(y|x_i))| < \alpha$$

for all $i \in [2n]$. Thus

$$\left|\left(\widetilde{d}'(w, w^*) - \widetilde{d}(w, w^*)\right) - \left(\widetilde{d}'(\bar{w}, w^*) - \widetilde{d}(\bar{w}, w^*)\right)\right| \leq \left|\widetilde{d}(w, w^*) - \widetilde{d}(\bar{w}, w^*)\right| + \left|\widetilde{d}'(w, w^*) - \widetilde{d}'(\bar{w}, w^*)\right| \leq 2\alpha$$

and so

$$\Pr\left[\sup_{w \in \mathbb{R}^d}\left|\widetilde{d}'(w, w^*) - \widetilde{d}(w, w^*)\right| > \varepsilon\right] \leq \Pr\left[\max_{w \in N}\left|\widetilde{d}'(w, w^*) - \widetilde{d}(w, w^*)\right| > \varepsilon - 2\alpha\right] \leq e^{2k \log \frac{n}{\alpha k} - (\varepsilon - 2\alpha)^2 n/2}$$

Setting $\alpha = \varepsilon/4$, we have that

$$n \lesssim \frac{1}{\varepsilon^2} k \log \frac{1}{\varepsilon}$$

suffices for

$$\Pr\left[\max_{w \in \mathbb{R}^k} \widetilde{d}'(w, w^*) - \widetilde{d}(w, w^*) > \varepsilon\right] < e^{-\Omega(\varepsilon^2 n)}.$$

$\square$

# B  ReLU Activation with Scalar $y$

In this section, we consider the model of

$$y = \phi(w^* \cdot x + \eta), \quad \eta \sim \mathcal{N}(0,1),$$

where $w^*, x \in \mathbb{R}^k$, $y, \eta \in \mathbb{R}$. We are given samples $(x, y) \in \mathbb{R}^k \times \mathbb{R}$, and want to estimate a $\widehat{w}$ that estimates the distribution of $y$ in TV.

The most challenging aspect of the ReLU setting is that we do not have an expression for the TV suffered by the MLE, such as Lemma 4.2 in the linear case. This forces us to directly analyze the log-likelihood.

For a fixed $x, w,$, the expectation of the log-likelihood ratio over $y$ is

$$\mathbb{E}_y\left[\log \frac{p_w(y \mid x)}{p_{w^*}(y \mid x)}\right] = -d_{KL}(w^*\|w) \le -2d_{TV}^2(w^*, w),$$

where the last inequality is via Pinsker's inequality. This equation implies that if $w$ is $\varepsilon$-far from $w^*$, then the expected log-likelihood ratio(LLR) is $< -2\varepsilon^2$. By definition, the MLE has a non-negative LLR. Hence, if the empirical LLR is close to the expectation, this would imply that the MLE has small TV.

However, we only receive a single sample of $y$ per $x$. For a fixed $w$, we can prove a Bernstein inequality, showing that given $1/\varepsilon^2 \log(1/\delta)$ samples, the empirical LLR is $< -\varepsilon^2$ for $w$ that are $\varepsilon$-far.

**Lemma B.1.** *Let $p_1, \ldots, p_n$ and $q_1, \ldots, q_n$ be distributions with $\mathbb{E}_i[d_{TV}(p_i, q_i)] \ge \varepsilon$, where we use the uniform measure on $i \in [n]$. Let $x_i \sim p_i$ for $i \in [n]$. Then w.p. $1 - \delta$, $\mathbb{E}_i[\log \frac{q_i(x_i)}{p_i(x_i)}] \le -\frac{\varepsilon^2}{4}$ for $n \ge O\left(\frac{1}{\varepsilon^2} \log \frac{1}{\delta}\right)$.*

The proof of this Lemma, as well as other Lemmas in this section, can be found in Appendix B.1.

In order to extend this to *all* $w \in \mathbb{R}^k$ that are $\varepsilon$-far, we will construct a cover over $\mathbb{R}^k$ depending on the values the log-likelihood ratio can take, and then apply the Bernstein inequality to each element in the cover.

In order to construct the cover, we first show that the log-likelihood ratio is bounded above by the magnitude of noise in $y$. For ease of notation, for a fixed $x \in \mathbb{R}^k$, and each $w \in \mathbb{R}^k$, define

$$\theta = \langle x, w \rangle \in \mathbb{R},$$

and let $\theta^* = \langle x, w^* \rangle$. Similar to the notation for $w$, for each $\theta \in \mathbb{R}$, define $p_\theta$ as the distribution of $\phi(\theta + \eta)$ for $\eta \sim N(0,1)$. Define the log likelihood ratio

$$\gamma_\theta(y) := \log \frac{p_\theta(y|x)}{p_{\theta^*}(y|x)}.$$

The following Lemma states that for a fixed datapoint $(x, y)$, the log-likelihood ratio is bounded by the noise in $y$:

**Lemma B.2.** *For any $y = \phi(\theta + \eta)$,*

$$\gamma_\theta(y) = \begin{cases} \log \Phi(-\theta) - \log \Phi(-\theta^*) & \text{if } y = 0 \\ \eta(\theta - \theta^*) - \frac{(\theta - \theta^*)^2}{2} & \text{if } y > 0 \end{cases}$$

*and therefore, for all $y$,*

$$\gamma_\theta(y) \le |\eta|^2/2.$$

Now, as $\gamma$ is bounded above by $\frac{|\eta|^2}{2}$, and it is concave wrt $\theta$, the following Lemma shows that we can partition $\theta$ into $O\left(\frac{A}{\varepsilon}\right)$ intervals, such that in each interval, $\gamma$ changes by atmost $\varepsilon$, or is very negative, i.e., $\gamma < -A$.

**Lemma B.3** (One-dimensional net)**.** *Let $A > B^2 > 1$. There exists a partition of $\mathbb{R}$ into $O(A/\varepsilon)$ intervals such that, for each interval $I$ in the partition and every $y = \phi(\theta^* + \eta)$ with $|\eta| \le B$, one of the following holds:*

- *For all $\theta \in I$, $\gamma_\theta(y) \leq -A$*
- *For all $\theta, \theta' \in I$, $|\gamma_\theta(y) - \gamma_{\theta'}(y)| \leq \varepsilon$*

Using Lemma B.2 and Lemma B.3, we can form a uniform bound, such that all $w$ that are $\varepsilon$-far from $w^*$ in distribution will have log-likelihood ratio smaller than $-\frac{\varepsilon^2}{4}$ on the training set. With some additional arguments, we can now show that as the MLE has positive log-likelihood ratio, it has small empirical TV.

**Lemma B.4.** *Let $x_1, \ldots, x_n$ be fixed, and $y_i \sim \phi(x_i^T w^* + \eta_i)$ for $\eta_i \sim \mathcal{N}(0,1)$. For $n \geq \frac{1}{\varepsilon^2} k \log \frac{1}{\varepsilon}$, the MLE $\widehat{w}$ satisfies*

$$\widetilde{d}(\widehat{w}, w^*) \leq \varepsilon.$$

This sample complexity guarantees that the MLE is good for the set of empirical $x_i \sim \mathcal{D}_x$, and we need to extend this to the expectation over $x \sim \mathcal{D}_x$, for which we use a Lemma similar to Lemma 4.3 in the linear case, and this completes the proof.

A straight forward combination of Lemma 4.3 and Lemma B.4 gives the following Theorem.

**Theorem B.5.** *Let $y = \phi(x^T w^* + \eta)$, for $w^* \in \mathbb{R}^k$, $x \sim \mathcal{D}_x$, and $\eta \sim \mathcal{N}(0,1)$. Then for a sufficiently large constant $C > 0$,*

$$n = C \cdot \frac{k}{\varepsilon^2} \log \frac{1}{\varepsilon}$$

*samples of $\{(y_i, x_i)\}_{i=1}^n$ suffices to guarantee that the MLE $\widehat{w}$ satisfies*

$$d(\widehat{w}, w^*) \leq \varepsilon.$$

## B.1 Proofs

**Lemma B.1.** *Let $p_1, \ldots, p_n$ and $q_1, \ldots, q_n$ be distributions with $\mathbb{E}_i[d_{TV}(p_i, q_i)] \geq \varepsilon$, where we use the uniform measure on $i \in [n]$. Let $x_i \sim p_i$ for $i \in [n]$. Then w.p. $1 - \delta$, $\mathbb{E}_i[\log \frac{q_i(x_i)}{p_i(x_i)}] \leq -\frac{\varepsilon^2}{4}$ for $n \geq O\left(\frac{1}{\varepsilon^2} \log \frac{1}{\delta}\right)$.*

*Proof.* Define $\gamma_i(x) = \log \frac{q_i(x)}{p_i(x)}$ and $a_i(x) := \max(\gamma_i(x), -2)$. We have that

$$\mathbb{E}_{i,x}[\gamma_i(x)] = -\mathbb{E}_i[d_{KL}(p_i, q_i)] \leq -\mathbb{E}_i[2d_{TV}(p_i, q_i)^2] \leq -2\varepsilon^2$$

and want to show that $\mathbb{E}_i[\gamma_i(x)] \leq -\varepsilon^2/4$ with high probability. Note that $a_i(x) \geq \gamma_i(x)$, so it suffices to show $\mathbb{E}_i[a_i(x)] \leq -\varepsilon^2/4$. We will do this with Bernstein's inequality, for which we need bounds on the moments of $a_i(x)$.

To simplify notation, fix a particular $i$ and consider $p = p_i, q = q_i, a = a_i$, and $x \sim p$.

For a random variable $v$, define $v_+, v_-$ to be the positive/negative parts of $v$, respectively, so $v = v_- + v_+$. Define $\Delta(x) = \frac{q(x)}{p(x)} - 1$. We have that $\mathbb{E}_{x \sim p}[\Delta(x)] = 0$, and

$$\mathbb{E}_{x \sim p}[\Delta_+(x)] = \mathbb{E}_{x \sim p}[-\Delta_-(x)] = d_{TV}(p, q). \tag{14}$$

Now, consider the function $b(z) := \max(\log(1 + z), -2) - z$. This function is nonpositive over $z \geq -1$, and $b(z) \leq -z^2/2$ for $z \leq 0$. Since

$$a(x) = b(\Delta(x)) + \Delta(x)$$

and $\mathbb{E}_{x \sim p}[\Delta(x)] = 0$, $\mathbb{E}_{x \sim p}[-a(x)] = \mathbb{E}_{x \sim p}[-b(\Delta(x))]$. This means

$$\mathbb{E}_{x \sim p}[-a(x)] = \mathbb{E}_{x \sim p}[-b(\Delta(x))] \geq \mathbb{E}_{x \sim p}[-b(\Delta(x))1_{\Delta(x) < 0}]$$

$$\geq \mathbb{E}_{x \sim p}[\Delta_-^2(x)/2]$$

or by (14),

$$\mathbb{E}_x[-a(x)] \geq \mathbb{E}_x[\Delta_-^2(x)/2] \geq \frac{1}{2} d_{TV}(p, q)^2. \tag{15}$$

**Bounding the positive higher moments.** We have that $p(x)e^{a(x)} = \max(q(x), e^{-2}p(x))$ so

$$\mathbb{E}[e^{a(x)}] = \int \max(q(x), e^{-2}p(x))dx$$

$$\leq 1 + e^{-2}\Pr[a(x) = -2].$$

In the following, we use that $e^t \geq 1 + t$ for all $t$, as well as $e^t = 1 + t + \sum_{k=2}^{\infty} \frac{1}{k!}t^k$. Therefore

$$1 + e^{-2}\Pr[a(x) = -2]$$

$$\geq \mathbb{E}[e^{a(x)}] = \mathbb{E}[e^{a_-(x)}1_{a(x)\leq 0} + e^{a_+(x)}1_{a(x)>0}]$$

$$\geq \mathbb{E}[1 + (a_-(x) + a_+(x)) + \sum_{k=2}^{\infty}\frac{1}{k!}a_+^k(x)]$$

$$= 1 + \mathbb{E}[a(x)] + \sum_{k=2}^{\infty}\frac{1}{k!}\mathbb{E}[a_+^k(x)]$$

so

$$\sum_{k=2}^{\infty}\frac{1}{k!}\mathbb{E}[a_+^k(x)] \leq \mathbb{E}[-a(x)] + e^{-2}\Pr[a(x) = -2].$$

We now show that the $\Pr[a(x) = -2]$ is smaller than the $\mathbb{E}[-a(x)]$ term, by relating to $-b$. When $a(x) = -2$, $\Delta(x) \leq -1 + 1/e^2$, and $b(\Delta(x)) = -2 - \Delta(x) \leq -1$. Since $-b(\Delta(x))$ is non-negative, and at least 1 whenever $a(x) = -2$,

$$\mathbb{E}[-a(x)] = \mathbb{E}[-b(\Delta(x))] \geq \Pr[a(x) = -2] \cdot 1$$

and hence

$$\sum_{k=2}^{\infty}\frac{1}{k!}\mathbb{E}[a_+^k(x)] \leq (1 + \frac{1}{e^2})\mathbb{E}[-a(x)]. \tag{16}$$

In particular, $\mathbb{E}[a_+^k(x)] \leq 2k!\,\mathbb{E}[-a(x)]$ for all $k \geq 2$.

**Bounding the second moment of $a$.** We have that

$$\mathbb{E}[a(x)^2] = \mathbb{E}[a_+^2(x) + a_-^2(x)]$$

and $\mathbb{E}[a_+^2(x)] \leq 4\,\mathbb{E}[-a(x)]$ by (16). We now bound $\mathbb{E}[a_-^2(x)]$. Note that $|a_-(x)| \leq \frac{2}{1-1/e^2}|\Delta_-(x)|$ by the construction of $a$. Therefore

$$a_-^2(x) \leq 6\Delta_-^2(x)$$

and so by (15),

$$\mathbb{E}[a_-^2(x)] \leq 6\,\mathbb{E}[\Delta_-^2(x)] \leq 12\,\mathbb{E}[-a(x)].$$

Thus

$$\mathbb{E}[a^2(x)] \leq 16\,\mathbb{E}[-a(x)]. \tag{17}$$

**Bernstein Concentration.** Now we can apply Bernstein's inequality (Theorem 2.10 of [8]). We apply the theorem to $X_i := a_i(x_i)$, which are independent. The theorem uses that

$$\sum_{i=1}^{n}\mathbb{E}[X_i^2] = n\,\mathbb{E}_{i,x}[a_i(x)^2] \leq 16n\,\mathbb{E}_{i,x}[-a_i(x)] =: v$$

by (17), and since

$$\sum_{i=1}^{n}\mathbb{E}[(X_i)_+^k] = n\,\mathbb{E}_{i,x}[a_{i,+}(x)^k] \leq 2k!\,\mathbb{E}_{i,x}[-a_i(x)] \leq \frac{1}{2}vk!$$

so we can set $c = 1$. Applying the theorem, we have that $S = \sum a_i(x_i) - \mathbb{E}[a_i(x_i)]$ satisfies

$$S \leq \sqrt{2v\log\frac{1}{\delta}} + \log\frac{1}{\delta}$$

with probability $1 - \delta$. Plugging in $v$ and rescaling by $n$, with probability $1 - \delta$ we have:

$$\mathbb{E}_i[a_i(x_i)] \leq \mathbb{E}_{i,x}[a_i(x)] + O(1) \cdot \sqrt{\mathbb{E}[-a(x)]\frac{1}{n}\log\frac{1}{\delta}} + \frac{1}{n}\log\frac{1}{\delta}$$

By our assumption on $n$ for a sufficiently large constant in the big $O$, this implies

$$\mathbb{E}_i[a_i(x_i)] \leq \mathbb{E}_{i,x}[a_i(x)] + \frac{1}{6}\varepsilon\sqrt{\mathbb{E}_{i,x}[-a_i(x)]} + \varepsilon^2/8$$

Since by (15), $\varepsilon \leq \sqrt{\mathbb{E}_i[d_{TV}(p_i, q_i)^2]} \leq \sqrt{\mathbb{E}_{i,x}[-2a_i(x)]}$, this means

$$\mathbb{E}_i[\gamma_i(x_i)] \leq \mathbb{E}_i[a_i(x_i)] \leq (-1 + \frac{\sqrt{2}}{6})\mathbb{E}_{i,x}[-a_i(x)] + \varepsilon^2/8$$

$$\leq (-1 + \frac{\sqrt{2}}{6})\frac{1}{2}\varepsilon^2 + \frac{1}{8}\varepsilon^2$$

$$\leq -\frac{1}{4}\varepsilon^2$$

as desired. $\qquad\square$

**Lemma B.2.** *For any $y = \phi(\theta + \eta)$,*

$$\gamma_\theta(y) = \begin{cases} \log\Phi(-\theta) - \log\Phi(-\theta^*) & \text{if } y = 0 \\ \eta(\theta - \theta^*) - \frac{(\theta-\theta^*)^2}{2} & \text{if } y > 0 \end{cases}$$

*and therefore, for all $y$,*
$$\gamma_\theta(y) \leq |\eta|^2/2.$$

*Proof.* Let $\Phi(x)$ be the cdf of a standard Gaussian. For $y > 0$,

$$\gamma_\theta(y) = \frac{1}{2}((y - \theta^*)^2 - (y - \theta)^2)$$

$$= \frac{1}{2}(\eta^2 - (\eta + \theta^* - \theta)^2)$$

$$= \eta(\theta - \theta^*) - \frac{(\theta - \theta^*)^2}{2}$$

Thus:

$$\gamma_\theta(y) = \begin{cases} \log\Phi(-\theta) - \log\Phi(-\theta^*) & \text{if } y = 0 \\ \eta(\theta - \theta^*) - \frac{(\theta-\theta^*)^2}{2} & \text{if } y > 0 \end{cases}$$

Now suppose $|\eta| \leq B$. We can upper bound $\gamma_\theta(y)$ for all $\theta$:

- If $y = 0$, then $-\theta^* \geq -B$, so
$$\gamma_\theta(0) \leq -\log\Phi(-\theta^*) \leq -\log e^{-B^2/2} = B^2/2.$$

- If $y > 0$, then
$$\gamma_\theta(y) = (\theta - \theta^*)\eta - \frac{(\theta - \theta^*)^2}{2} \leq \eta^2/2 \leq B^2/2.$$

as desired. $\qquad\square$

**Lemma B.3** (One-dimensional net). *Let $A > B^2 > 1$. There exists a partition of $\mathbb{R}$ into $O(A/\varepsilon)$ intervals such that, for each interval $I$ in the partition and every $y = \phi(\theta^* + \eta)$ with $|\eta| \leq B$, one of the following holds:*

- *For all $\theta \in I$, $\gamma_\theta(y) \leq -A$*
- *For all $\theta, \theta' \in I$, $|\gamma_\theta(y) - \gamma_{\theta'}(y)| \leq \varepsilon$*

*Proof.* To define our partition, we actually define two partitions, depending on whether $y = 0$, then intersect them for our final partition.

First, consider $y = 0$. By Lemma B.2, $\gamma_\theta(0)$ is monotonically decreasing in $\theta$, from its maximum of at most $B^2/2$. We can thus define a partition $P_1$ consisting of intervals of the form $I_i := \{\theta \mid \gamma_\theta(0) \in (B^2/2 - (i+1)\varepsilon, B^2/2 - i\varepsilon)\}$, for $i \in \{0, 1, \ldots, (A + B^2/2)/\varepsilon\}$, plus a special interval $I'$ of $\{\theta \mid \gamma_\theta(0) < -A\}$. When $y = 0$, this partition satisfies the desired conclusion to the lemma: $|\gamma_\theta(0) - \gamma_{\theta'}(0)| \le \varepsilon$ for all $\theta, \theta' \in I_i$, while $\gamma_\theta(0) < -A$ for $\theta \in I'$. Call this partition $P_0$, which has size $O(A/\varepsilon)$.

Second, consider $y > 0$. Define $R = \sqrt{2A} + B$. Note that $R^2 \lesssim A$ and $(R - B)^2 \ge 2A$. Therefore for $|\theta - \theta^*| \ge R$,

$$\gamma_\theta(y) \le -\frac{1}{2}\max(0, |\theta - \theta^*| - \eta)^2 \le -A.$$

Consider any $\theta, \theta' \in [\theta^* - R, \theta^* + R]$ with $\alpha := |\theta - \theta'|$. We have

$$|\gamma_\theta(y) - \gamma_{\theta'}(y)| \le |\eta(\theta - \theta')| + \frac{1}{2}\left|(\theta' - \theta^*)^2 - (\theta - \theta^*)^2\right|$$

$$\le B\alpha + \frac{1}{2}|(\theta' - \theta)(-2\theta^* + (\theta' + \theta))|$$

$$\le B\alpha + \frac{1}{2}\alpha(2R) = \alpha(B + R).$$

Thus, for $\alpha = \frac{\varepsilon}{2R}$, this is at most $\varepsilon$. If we partition $[\theta^* - R, \theta^* + R]$ into length-$\alpha$ intervals, we get a size $O(R^2/\varepsilon) = O(A/\varepsilon)$ partition $P_1$ of $\mathbb{R}$ that has the desired property for all $y > 0$.

Our final partition is defined by all endpoints in either $P_0$ and $P_1$. This has size $O(A/\varepsilon)$, and within each interval the conclusion holds for both $y = 0$ and $y > 0$, as needed.

$\square$

**Lemma B.4.** *Let* $x_1, \ldots, x_n$ *be fixed, and* $y_i \sim \phi(x_i^T w^* + \eta_i)$ *for* $\eta_i \sim \mathcal{N}(0, 1)$. *For* $n \ge \frac{1}{\varepsilon^2}k\log\frac{1}{\varepsilon}$, *the MLE* $\widehat{w}$ *satisfies*

$$\widetilde{d}(\widehat{w}, w^*) \le \varepsilon.$$

*Proof.* For any $w \in \mathbb{R}^k$, and a sample $(x_i, y_i)$, let $p_w(y|x_i)$ be the conditional distribution of $y = \phi(\langle x_i, w\rangle + \eta)$, and let $\gamma_{i,w}$ be the log-likelihood ratio between $w$ and $w^*$ on this sample:

$$\gamma_{i,w}(y) := \log\frac{p_w(y|x_i)}{p_{w^*}(y|x_i)}.$$

Then

$$\mathbb{E}_y[\gamma_{i,w}(y)] = -d_{KL}(p_{i,w^*}(y|x_i)||p_{i,w}(y|x_i)).$$

Define

$$d_{KL}(w^*, w) := \frac{1}{n}\sum_{i=1}^{n} d_{KL}(p_{i,w^*}(y|x_i)||p_{i,w}(y|x_i)).$$

**Concentration.** From Lemma B.1, we see that if $\widetilde{d}(w^*, w) \ge \varepsilon$, then for $n \ge O(\frac{1}{\varepsilon^2}\log\frac{1}{\delta})$,

$$\overline{\gamma}_w := \frac{1}{n}\sum_{i=1}^{n}\gamma_{i,w}(y_i) < -\frac{\varepsilon^2}{4}, \tag{18}$$

with probability $1 - \delta$.

Of course, whenever $\overline{\gamma}_w < 0$, the likelihood under $w^*$ is larger than the likelihood under $w$. Thus, for each *fixed* $w$ with $\widetilde{d}(w^*, w) \ge \varepsilon$, maximizing likelihood would prefer $w^*$ to $w$ with probability $1 - \delta$ if $n \ge O(\frac{1}{\varepsilon^2}\log\frac{1}{\delta})$.

Nothing above is specific to our ReLU-based distribution. But to extend to the MLE over all $w$, we need to build a net using properties of our distribution.

**Building a net.**   First, with high probability, $|\eta_i| \leq B = O(\sqrt{\log n})$ for all $i$. Suppose this happens. For each $i$, by an abuse of notation, let $\gamma_{i,w}(y) = \gamma_{\langle x_i, w\rangle}(y)$ where the value of $\theta^*$ when considering $i$ is $\langle x_i, w^*\rangle$. By Lemma B.2,

$$\gamma_{i,w}(y_i) \leq B^2/2$$

for all $i$. Let $A = O(n \log n) > nB^2$. By Lemma B.3, for each $i \in [n]$, there exists a partition $P_i$ of $\mathbb{R}$ into $O(A/\varepsilon^2)$ intervals, such that for interval $I \in P_i$, and any $w, w'$ with $x_i^T w, x_i^T w' \in I$, either

$$|\gamma_{i,w}(y_i) - \gamma_{i,w'}(y_i)| \leq \varepsilon^2/2 \tag{19}$$

or $\gamma_{i,w}(y_i) < -A$.

These individual partitions $P_i$ on $\langle x_i, w\rangle$ induce a partition $P$ on $\mathbb{R}^k$, where $w, w'$ lie in the same cell of $P$ if $\langle x_i, w\rangle$ and $\langle x_i, w^*\rangle$ are in the same cell of $P_i$ for all $i \in [n]$. Since $P$ is defined by $n$ sets of $O\left(\frac{A}{\varepsilon^2}\right)$ parallel hyperplanes in $\mathbb{R}^k$, the number of cells in $P$ is:

$$2\left(\frac{2Aen}{\varepsilon^2 k}\right)^k.$$

We choose a net $\mathcal{N}$ to contain, for each cell in $P$, the $w$ in the cell maximizing $\widetilde{d}(w^*, w)$. This has size

$$\log|\mathcal{N}| \lesssim k \log \frac{n}{\varepsilon}.$$

By (18), for our $n \geq O\left(\frac{1}{\varepsilon^2} k \log \frac{k}{\varepsilon}\right)$, we have with high probability that $\overline{\gamma}_w \leq -\frac{\varepsilon^2}{4}$, for all $w \in \mathcal{N}$ with $\widetilde{d}(w^*, w) \geq \varepsilon$. Suppose that both this happens, and $|\eta_i| \leq B$ for all $i$. We claim that the MLE $\widehat{w}$ must have $\widetilde{d}(w^*, \widehat{w}) < \varepsilon$.

Consider any $w \in \mathbb{R}^d$ with $\widetilde{d}_{TV}(w^*, w) \geq \varepsilon$. Let $w' \in \mathcal{N}$ lie in the same cell of $P$. By our choice of $\mathcal{N}$, we know $\widetilde{d}_{TV}(w^*, w') \geq \widetilde{d}_{TV}(w^*, w) \geq \varepsilon$, so $\overline{\gamma}_{w'} \leq -\varepsilon^2$. Now we consider two cases. In the first case, there exists $i$ with $\gamma_{i,w}(y_i) < -A$. Then

$$\overline{\gamma}_w = \frac{1}{n} \sum_i \gamma_{i,w}(y_i) \leq -\frac{A}{n} + B^2/2 < 0.$$

Otherwise, by (19),

$$\overline{\gamma}_w \leq \overline{\gamma}_{w'} + |\overline{\gamma}_w - \overline{\gamma}_{w'}| \leq -\varepsilon^2 + \max_i |\gamma_{i,w}(y_i) - \gamma_{i,w'}(y_i)| \leq -\varepsilon^2/2.$$

In either case, $\overline{\gamma}_w < 0$ and the likelihood under $w^*$ exceeds that under $w$. Hence the MLE $\widehat{w}$ must have $\widetilde{d}(w^*, \widehat{w}) \leq \varepsilon$. $\qquad\square$

**Theorem B.5.** *Let $y = \phi(x^T w^* + \eta)$, for $w^* \in \mathbb{R}^k$, $x \sim \mathcal{D}_x$, and $\eta \sim \mathcal{N}(0,1)$. Then for a sufficiently large constant $C > 0$,*

$$n = C \cdot \frac{k}{\varepsilon^2} \log \frac{1}{\varepsilon}$$

*samples of $\{(y_i, x_i)\}_{i=1}^n$ suffices to guarantee that the MLE $\widehat{w}$ satisfies*

$$d(\widehat{w}, w^*) \leq \varepsilon.$$

*Proof.* Let $D_x$ denote the dataset $\{x_i\}_{i \in [n]}$ that is used to find the MLE. Notice that the MLE is found using this finite subset, but we would like to make a claim about $\mathcal{D}_x$ without making any parametric or simplifying assumptions on the distribution $\mathcal{D}_x$.

An application of Lemma 4.3 tells us that with probability $1 - e^{-\Omega(n\varepsilon^2)}$, the expectation over the distribution $\mathcal{D}_x$ and the dataset $D_x$ are within $\varepsilon/2$ of one another:

$$d(\widehat{w}, w^*) \leq \widetilde{d}(\widehat{w}, w^*) + \varepsilon/2.$$

Now, all we need to show is that the MLE has a small TV distance on the finite dataset, and Lemma B.4 tells us that with probability $1 - e^{-\Omega(n\varepsilon^2)}$,

$$\widetilde{d}(\widehat{w}, w^*) \leq \varepsilon/2.$$

Substituting in the above inequality, we get $d(\widehat{w}, w^*) \leq \varepsilon$. $\qquad\square$

# C  ReLU Activations with $d > 1$, Unknown Covariance

We recommend the reader review Appendix B, which contains the proof recipe for the case of scalar $y$. The proofs in this section generalize those of Appendix B.

Consider a sample $(x, y) \in \mathbb{R}^{k \times d}$, with

$$y = \phi(W^* x + \eta), \tag{20}$$

where $W^* \in \mathbb{R}^d \times k$, and noise $\eta \sim \mathcal{N}(0, \Sigma^*)$. The matrices $W^*$ and $\Sigma^*$ are unknown. For each matrix $W \in \mathbb{R}^{d \times k}$, let $\theta = Wx \in \mathbb{R}^d$, denote a reparametrization of $W$, and let $\theta^*$ denote $\theta^* = W^* x$. Let $S$ denote the co-ordinates of $y$ that are zero-valued. Then the log-likelihood for each $\theta, \Sigma$ is given by

$$f_{\theta, \Sigma}(y) := \log p_{W, \Sigma}(y \mid x) = c - \frac{1}{2} \log|\Sigma| + \log \int_{t : t_S \leq 0, t_{S^c} = y_{S^c}} \exp\Big\{ -(t - \theta)^T \Sigma^{-1} (t - \theta)/2 \Big\}.$$

where $c$ is a normalization constant which does not depend on $\theta$ or $\Sigma$. Let

$$P := \Sigma^{-1}$$

and let $P^*$ be the precision matrix of the noise $\eta$, and $P_S, P_{SS^c}, P_{S^c S}, P_{S^c}$ be the block matrices of $P$ corresponding to the index sets $S$ and its complement $S^c$.

By some arithmetic involving completion of squares, we can decompose the integral in $f$ into the sum of two functions $g, h$, such that

$$f_{\theta, \Sigma}(y) = c - \frac{1}{2} \log|\Sigma| + g_{\theta, \Sigma}(y) + h_{\theta, \Sigma}(y).$$

The first term $g$ corresponds to the quadratic term involving the observed positive-valued coordinates $y_{S^c}$:

$$g_{\theta, \Sigma}(y) = -(y_{S^c} - \theta_{S^c})^T (P_{S^c} - P_{S^c S}(P_S)^{-1} P_{SS^c})(y_{S^c} - \theta_{S^c})/2.$$

As the matrix $P_{S^c} - P_{S^c S}(P_S)^{-1} P_{SS^c} = ((P^{-1})_{S^c})^{-1} = \Sigma_{S^c}^{-1}$ is the precision matrix of $\eta_S$, if $\Sigma$ were the covariance of $\eta$, we can simplify the above equation as

$$g_{\theta, \Sigma}(y) = -(y_{S^c} - \theta_{S^c})^T (\Sigma_{S^c})^{-1}(y_{S^c} - \theta_{S^c})/2. \tag{21}$$

The second term corresponds to the probability under $\theta, P$ of observing zero-valued coordinates corresponding to the index set $S$, given the positive coordinates $y_{S^c}$:

$$h_{\theta, \Sigma}(y) = \log \int_{t \leq 0} \exp\Big( -\|P_S^{\frac{1}{2}}(t - \theta_S) + (P_S)^{-1/2} P_{SS^c}(y_{S^c} - \theta_{S^c})\|^2/2 \Big). \tag{22}$$

The *log-likelihood ratio* is the difference between $f_{\theta, \Sigma}$ and $f_{\theta^*, \Sigma^*}$, which we denote by

$$\gamma_{\theta, \Sigma}(y) := f_{\theta, \Sigma}(y) - f_{\theta^*, \Sigma^*}(y)$$

Over a dataset $\{(x_i, y_i)\}_{i \in [n]}$, the average log-likelihood ratio is given by

$$\bar{\gamma}_{W, \Sigma} := \frac{1}{n} \sum_i \gamma_{W x_i, \Sigma}(y_i).$$

**Remark C.1.** *For ease of analysis, we will interchange between the precision matrix $P$ in $\gamma_{\theta, P}$ and the covariance matrix $\Sigma$ in $\gamma_{\theta, \Sigma}$, and it should be understood that $P = \Sigma^{-1}$. The same applies to the functions $g_{\theta, \Sigma}$ and $h_{\theta, \Sigma}$. Finally, the matrix $P^*$ refers to the ground truth precision matrix $(= \Sigma^{*-1})$.*

Analogous to Appendix B, we start by showing that the log-likelihood ratio is bounded by the noise in the sample. The proofs of results in this Section are in Subsection C.1.

**Lemma C.2.** *Assume $P^* := \Sigma^{*-1}$ satisfies Assumption 4.4.*

*For all $y = \phi(\theta^* + \eta)$ such that $S$ denotes the zero-coordinates of $y$, and $\eta$ such that $\|P_S^{*\frac{1}{2}}\eta_S\|, \|P_{S^c}^{*\frac{1}{2}}\eta_{S^c}\| \leq B$, if the max eigenvalue $\lambda_{\max}(P)$ satisfies*

$$\frac{\lambda_{\max}(P)}{\lambda_{\min}(P^*)} \leq C,$$

*then for all $\theta \in \mathbb{R}^d$, we have*

$$\gamma_{\theta,P} \leq \frac{d}{2}\log(C) + 3B^2.$$

For the ease of stating the next Lemma, we assume that across the samples of $y$ in the training data, at least one coordinate has sufficiently many positive samples. The proof of our theorem separately handles cases violating this assumption.

**Assumption C.3.** *Let $\delta \in (0,1)$ be a parameter corresponding to the failure probability of our algorithm. Then, there exists a coordinate $j \in [d]$, such that for at least $n' = O\left(\log\frac{1}{\delta}\right)$ samples $y_{i_1}, \ldots, y_{i_{n'}}$ in the dataset, the $j$-th coordinate is positive.*

This is a very weak assumption: if it is violated, then $W = 0_{d \times k}, \Sigma = 0$ will achieve a TV distance smaller than $\frac{2\varepsilon^2}{d}$.

Appendix B assumed that the variance in $y$ was 1. Since Section 4.2 considers an unknown $\Sigma^*$, we need the following Lemma to show that the MLE will select a precision matrix $P$, whose eigenvalues are reasonably bounded wrt $\Sigma^{*-1}$.

**Lemma C.4.** *Under Assumption 4.4, C.3, consider $P \in \mathbb{R}_+^{d \times d}$ such that $\frac{\lambda_{\max}(P)}{\lambda_{\min}(P)} \leq \kappa$ and*

$$\frac{\lambda_{\max}(P)}{\lambda_{\max}(P^*)} \geq O\left(\frac{\kappa^3 d^2 n^2}{k^2} + \frac{B^2 n \kappa}{k}\right).$$

*Then, for all $W \in \mathbb{R}^{d \times k}$, and for all $y_i = \phi(W^* x_i + \eta_i)$ with $\|P_{S^c}^{*\frac{1}{2}}\eta_{S^c}\|, \|P_S^{*\frac{1}{2}}\eta_S\| \leq B$, we have*

$$\bar{\gamma}_{W,P} := \frac{1}{n}\sum_{i \in [n]} \gamma_{Wx_i,P}(y_i) < 0.$$

Lemma C.2 and Lemma C.4 show that the MLE will only select precision matrices $P$ that have max eigenvalues in a certain range of the true precision matric $P^*$.

Now, for matrices in the above eigenvalue range, we first construct a geometric net over the max eigenvalue $\rho$ of the precision matrix, and then cover the matrices whose max eigenvalue is smaller than $\rho$.

**Lemma C.5** ($\Sigma$ cover). *For $B > 1$, and $0 < L < U$, let $A > \max\left\{\sqrt{\log\frac{1}{\varepsilon}}, B^2 U\kappa, \frac{d}{2}\log\left(\frac{\kappa U}{L}\right), 1\right\}$.*

*Let $P^* := \Sigma^{*-1}$ be the precision matrix of $\eta$. Let $\Omega \subset \mathbb{R}_+^{d \times d}$ denote the set of positive definite matrices $P \in \mathbb{R}_+^{d \times d}$ with condition number $\kappa$ and whose maximum eigenvalue lies in $[L\lambda_{\min}(P^*), U \cdot \lambda_{\max}(P^*)]$.*

*Then, there exists a partition of $\Omega$ of size*

$$\left(\text{poly}\left(A, \frac{1}{\varepsilon}\right)\right)^{d^2}$$

*such that for all $\theta \in \mathbb{R}^d$ and all $y = \phi(\theta^* + \eta) \in \mathbb{R}^d$ with $\|P_S^{*\frac{1}{2}}\eta_S\|, \|P_{S^c}^{*\frac{1}{2}}\eta_{S^c}\| \leq B$, and each cell $I$ in the partition, one of the following holds:*

- *for all $P \in I$, $\gamma_{\theta,P}(y) < -A$, or*
- *for all $P, P' \in I$, we have $|\gamma_{\theta,P}(y) - \gamma_{\theta,P'}(y)| \leq \epsilon$.*

Analogous to Appendix B, we now construct a partition over $W$ for a fixed precision matrix $P$, such that each cell in the partition has very small log-likelihood (in which case the MLE will not choose it) or the log-likelihood changes slowly.

**Lemma C.6** ($W$-net). *Let $\eta_{S^c}, \eta_S$ be such that*

$$\|P_{S^c}^{*\frac{1}{2}} \eta_{S^c}\| \le B_1, \|P_S^{*\frac{1}{2}} \eta_S\| \le B_2,$$

*for $B_1, B_2 \ge 0$.*

*Let $A > \max\{B_1^2, B_2^2, \mathrm{poly}(C, \kappa)\}$. Let $P^* = \Sigma^{*-1}$ be the precision matrix of $\eta$. For a fixed matrix $P \in \mathbb{R}^{d \times d}$ whose condition number satisfies Assumption 4.4 and whose eigenvalues satisfy $\lambda_{\max}(P) \in [e^{-\frac{2A}{d}} \lambda_{\min}(P^*), C\lambda_{\max}(P^*)]$, there exists a partition $\mathcal{I}$ of $\mathbb{R}^d$ with size*

$$\left(\mathrm{poly}\left(A, \frac{1}{\varepsilon}\right)\right)^{3d}$$

*such that for each interval $I \in \mathcal{I}$, we have one of the following:*

- *for all $\theta \in I$, $\gamma_{\theta, P}(y) < -A$, or*
- *for all $\theta, \theta' \in I$, $|\gamma_{\theta, P}(y) - \gamma_{\theta', P}(y)| \le \epsilon$.*

Using the above lemmas, we can show that the MLE will only pick out $\widehat{W}, \widehat{P}$ such that they have small TV on the dataset of $\{x_i\}$.

**Lemma C.7.** *Let $x_1, \ldots, x_n$ be fixed, and $y_i = \phi(W^* x_i + \eta_i)$ for $\eta_i \sim \mathcal{N}(0, \Sigma^*)$, and $W^* \in \mathbb{R}^{d \times k}$ with $\Sigma^* \in \mathbb{R}^{d \times d}$ satisfying Assumption 4.4 and Assumption C.3. For a sufficiently large constant $C > 0$,*

$$n = C \cdot \frac{(d^2 + kd)}{\varepsilon^2} \log \frac{kd\kappa}{\varepsilon}$$

*samples suffice to guarantee that with high probability, the MLE $\widehat{W}, \widehat{\Sigma}$ satisfies*

$$\widetilde{d}\left((\widehat{W}, \widehat{\Sigma}), (W^*, \Sigma^*)\right) \le \varepsilon.$$

**Lemma C.8.** *Let $\{x_i\}_{i=1}^n$ be i.i.d. random variables such that $x_i \sim \mathcal{D}_x$.*

*Let $P^* := \Sigma^{*-1}$. Let $\lambda_{\min}^*, \lambda_{\max}^*$ be the minimum and maximum eigenvalues of $P^*$. For $0 < L < U$, let $\Omega$ denote the following set of precision matrices*

$$\Omega := \left\{ P \in \mathbb{R}_+^{d \times d} : \frac{\lambda_{\max}(P)}{\lambda_{\min}(P)} \le \kappa \text{ and } \lambda_{\max}(P) \in [L \cdot \lambda_{\min}^*, U \cdot \lambda_{\max}^*] \right\}.$$

*Then, for a sufficiently large constant $C > 0$, and for*

$$n = C \cdot \left(\frac{kd + d^2}{\varepsilon^2}\right) \log\left(\frac{kd\kappa}{\varepsilon} \log\left(\frac{U}{L}\right)\right),$$

*we have:*

$$\Pr_{x_i \sim \mathcal{D}_x} \left[ \sup_{W \in \mathbb{R}^{d \times k}, P \in \Omega} \left| \widetilde{d}((W, P), (W^*, P^*)) - d((W, P), (W^*, P^*)) \right| > \varepsilon \right] \le e^{-\Omega(n\varepsilon^2)}.$$

**Theorem 4.5.** *Let $\mathbb{R}_\kappa^{d \times d}$ denote the set of positive definite matrices with condition number $\kappa$. Given $n$ samples $\{(x_i, y_i)\}_{i=1}^n$ satisfying Assumption 4.4, where $x_i \sim \mathcal{D}_x$ i.i.d., and $y_i$ is generated according to (7), let $\widehat{W}, \widehat{\Sigma} := \arg\max_{W \in \mathbb{R}^{d \times k}, \Sigma \in \mathbb{R}_\kappa^{d \times d}} \frac{1}{n} \sum_i \log p_{W, \Sigma}(y_i \mid x_i)$. Then, for a sufficiently large constant $C > 0$,*

$$n = C \cdot \left(\frac{kd + d^2}{\varepsilon^2}\right) \log\left(\frac{\kappa kd}{\varepsilon \delta}\right)$$

*samples suffice to ensure that with probability $1 - \delta$, we have*

$$d_{TV}\left((\widehat{W}, \widehat{\Sigma}), (W^*, \Sigma^*)\right) \le \varepsilon.$$

*Proof of Theorem 4.5.* First, we consider the cases violating Assumption C.3.

As $n \propto \frac{d^2}{\varepsilon^2} \log \frac{1}{\delta}$, if assumption C.3 is violated, then it implies that each coordinate is non-zero in atmost a $\varepsilon^2/d^2$ fraction of the samples, and a union bound implies that the probability of seeing a non-zero vector is atmost $\varepsilon^2/d$. Hence, with high probability over the draws of the data, returning the all-zeros vector always will achieve a TV distance smaller than $\frac{2\varepsilon^2}{d}$.

Let $\widehat{P}, P^* = \widehat{\Sigma}^{-1}, \Sigma^{*-1}$. Now, if Assumption C.3 holds, Lemma C.7 guarantees that the MLE has small TV on the $x_i$ observed in the dataset:

$$\widetilde{d}((\widehat{W}, \widehat{P}), (W^*, P^*)) \leq \varepsilon.$$

The above result is over the finite $x_i$ observed in our dataset. To generalize it over $x \sim \mathcal{D}_x$, we use Lemma C.8, which gives

$$d((\widehat{W}, \widehat{P}), (W^*, P^*)) - \widetilde{d}((\widehat{W}, \widehat{P}), (W^*, P^*)) \leq \varepsilon.$$

Rescaling $\varepsilon$ gives the conclusion of the Theorem. $\qquad\square$

## C.1 Proofs of Appendix C.

**Lemma C.2.** *Assume $P^* := \Sigma^{*-1}$ satisfies Assumption 4.4.*

*For all $y = \phi(\theta^* + \eta)$ such that $S$ denotes the zero-coordinates of $y$, and $\eta$ such that $\|P_S^{*\frac{1}{2}} \eta_S\|, \|P_{S^c}^{*\frac{1}{2}} \eta_{S^c}\| \leq B$, if the max eigenvalue $\lambda_{\max}(P)$ satisfies*

$$\frac{\lambda_{\max}(P)}{\lambda_{\min}(P^*)} \leq C,$$

*then for all $\theta \in \mathbb{R}^d$, we have*

$$\gamma_{\theta,P} \leq \frac{d}{2} \log(C) + 3B^2.$$

*Proof.* We have

$$\gamma_{\theta,\Sigma} \leq \frac{1}{2} \log \frac{|\Sigma^*|}{|\Sigma|} + g_{\theta,\Sigma} - g_{\theta^*,\Sigma^*} + h_{\theta,\Sigma} - h_{\theta^*,\Sigma^*}. \tag{23}$$

From Lemma C.9, C.10, we have

$$g_{\theta,\Sigma} - g_{\theta^*,\Sigma^*} + h_{\theta,\Sigma} - h_{\theta^*,\Sigma^*} \leq g_{\theta,\Sigma} + \frac{1}{2} \log \frac{|P_S^*|}{|P_S|} + 3B^2.$$

Substituting in Eqn (23), we get

$$\gamma_{\theta,\Sigma} \leq g_{\theta,\Sigma} + \frac{1}{2} \log \frac{|\Sigma^*|}{|\Sigma|} + \frac{1}{2} \log \frac{|P_S^*|}{|P_S|} + 3B^2.$$

As $(P_S^*)^{-1} = \Sigma_S^* - \Sigma_{SS^c}^* \Sigma_{S^c}^{*-1} \Sigma_{S^c S}^*$, by the matrix determinant rule, we have

$$\log|\Sigma^*| + \log|P_S^*| = \log|\Sigma_{S^c}^*|.$$

This gives

$$\gamma_{\theta,\Sigma} \leq g_{\theta,\Sigma} + \frac{1}{2} \log \frac{|\Sigma_{S^c}^*|}{|\Sigma_{S^c}|} + 3B^2.$$

This gives

$$\gamma_{\theta,\Sigma} \leq g_{\theta,\Sigma} + \frac{d}{2} \log \frac{\lambda_{\max}(\Sigma^*)}{\lambda_{\min}(\Sigma)} + 3B^2,$$

$$= g_{\theta,\Sigma} + \frac{d}{2} \log \frac{\lambda_{\max}(P)}{\lambda_{\min}(P^*)} + 3B^2. \tag{24}$$

As the matrix $\Sigma_{S^c}^{-1}$ is positive definite, we trivially get

$$g_{\theta_i,\Sigma}(y) = -(y_{i,S^c} - \theta_{i,S^c})^T(\Sigma_{S^c})^{-1}(y_{i,S^c} - \theta_{i,S^c})/2 \leq 0.$$

Substituting in Eqn (24), we get

$$\gamma_{\theta,\Sigma} \leq \frac{d}{2}\log\frac{\lambda_{\max}(P)}{\lambda_{\min}(P^*)} + 3B^2.$$

As the Lemma assumes

$$\lambda_{\max}(P) \leq C\lambda_{\min}(P^*),$$

we get

$$\gamma_{\theta,\Sigma} \leq \frac{d}{2}\log(C) + 3B^2.$$

$\square$

**Lemma C.9.** *Consider the function $g$ defined in Eq* (21). *For the ground truth parameters $\theta^*, \Sigma^*$, the function $g_{\theta^*,\Sigma^*}$ satisfies*

$$-g_{\theta^*,\Sigma^*} \leq \frac{1}{2}\|\eta_{S^c}\|_{\Sigma_{S^c}}^2,$$

*which is, with probability $1 - e^{-\Omega(d)}$,*

$$-g_{\theta^*,\Sigma^*} \leq O(d).$$

*Proof.* As $y_{S^c}$ are the positive valued coordinates in $y$, we have

$$y_{S^c} - \theta_{S^c}^* = \eta_{S^c},$$

which gives

$$g_{\theta^*,\Sigma^*}(y) = -(y_{S^c} - \theta_{S^c}^*)^T(\Sigma_{S^c}^*)^{-1}(y_{S^c} - \theta_{S^c}^*)/2,$$
$$= -\|\eta_{S^c}\|_{\Sigma_{S^c}}^2/2.$$

As $\eta_{S^c}$ is Gaussian with covariance $\Sigma_{S^c}^*$, the expected norm is $\frac{|S^c|}{2}$, which implies that with probability $1 - e^{-\Omega(|S^c|)}$, we have

$$-g_{\theta^*,\Sigma^*}(y) \leq O(|S^c|).$$

$\square$

**Lemma C.10.** *Consider $y$ generated according to Eqn* (20) *by*

$$y = \phi(\theta^* + \eta), \quad \eta \sim \mathcal{N}(0, \Sigma^*).$$

*For all $\theta \in \mathbb{R}^d, \Sigma \in \mathbb{R}_+^{d\times d}$, and the function $h_{\theta,\Sigma}$ defined in Eqn* (22), *the difference $h_{\theta,\Sigma}(y) - h_{\theta^*,\Sigma^*}(y)$ satisfies*

$$h_{\theta,\Sigma}(y) - h_{\theta^*,\Sigma^*}(y) \leq \frac{1}{2}\log\frac{|P_S^*|}{|P_S|} + \|P_{S^c}^{*\frac{1}{2}}\eta_{S^c}\|^2 + 2\|P_S^{*\frac{1}{2}}\eta_S\|^2 - \|\eta_{S^c}\|_{\Sigma_{S^c}}^2 + O(|S|), \quad (25)$$

*where $P^* = \Sigma^{*-1}$ is the precision matrix of $\eta$.*

*Proof.* For $\theta \in \mathbb{R}^d, \Sigma \in \mathbb{R}_+^{d\times d}$, and $P = \Sigma^{-1}$, we have

$$h_{\theta,\Sigma}(y) = \log\int_{t\leq 0}\exp\left(-\|P_S^{\frac{1}{2}}(t - \theta_S) + (P_S)^{-1/2}P_{SS^c}(y_{S^c} - \theta_{S^c})\|^2/2\right),$$

$$\leq \log\int_{t\in\mathbb{R}^{|S|}}\exp\left(-\|P_S^{\frac{1}{2}}(t - \theta_S) + (P_S)^{-1/2}P_{SS^c}(y_{S^c} - \theta_{S^c})\|^2/2\right),$$

$$\leq \frac{|S|}{2}\log(2\pi) - \frac{1}{2}\log|P_S|, \quad (26)$$

where the last step follows from the integral of a Gaussian pdf. This gives a sufficient upper bound on $h_{\theta,\Sigma}(y)$, and now we will focus on lower bounding $h_{\theta^*,\Sigma^*}(y)$.

For the coordinates of $y$ in $S^c$, we have $y_{S^c} - \theta^*_{S^c} = \eta_{S^c}$. Substituting in Eqn (22), we get

$$h_{\theta^*,\Sigma^*}(y) = \log \int_{t \leq 0} \exp\left(-\|P_S^{*\frac{1}{2}}(t - \theta^*_S) + (P_S^*)^{-1/2} P^*_{SS^c} \eta_{S^c}\|^2/2\right),$$

$$= \log \int_{t \leq 0} \exp\left(-\|P_S^{*\frac{1}{2}}(t - \theta^*_S) + (P_S^*)^{-1/2} P^*_{SS^c} \eta_{S^c}\|^2/2\right).$$

Using $\|a + b\|^2 \leq 2a^2 + 2b^2$, we get

$$h_{\theta^*,\Sigma^*}(y) \geq -\|(P_S^*)^{-1/2} P^*_{SS^c} \eta_{S^c}\|^2 + \log \int_{t \leq 0} \exp\left(-\|P_S^{*\frac{1}{2}}(t - \theta^*_S)\|^2\right).$$

Set $u := P_S^{*\frac{1}{2}}(t - \theta^*_S)$, and by the change of variables formula, we get:

$$h_{\theta^*,\Sigma^*}(y) \geq -\|(P_S^*)^{-1/2} P^*_{SS^c} \eta_{S^c}\|^2 + \log \int_{P_S^{*-\frac{1}{2}} u + \theta^*_S \leq 0} \left|(P_S^*)^{-1/2}\right| \cdot \exp\left(-\|u\|^2\right),$$

$$= -\|(P_S^*)^{-1/2} P^*_{SS^c} \eta_{S^c}\|^2 + \frac{1}{2}\log\left|P_S^{*-1}\right| + \log \int_{P_S^{*-\frac{1}{2}} u + \theta^*_S \leq 0} \exp\left(-\|u\|^2\right).$$

For $i \in S$, we have $\theta^*_i + \eta_i \leq 0$. This gives

$$P_S^{*-\frac{1}{2}} u \leq \eta_S \Rightarrow P_S^{*-\frac{1}{2}} u + \theta^*_S \leq 0 \Rightarrow \log \int_{P_S^{*-\frac{1}{2}} u + \theta^*_S \leq 0} \exp\left(-\|u\|^2\right) \geq \log \int_{P_S^{*-\frac{1}{2}} u \leq \eta_S} \exp\left(-\|u\|^2\right),$$

using which we get

$$h_{\theta^*,\Sigma^*}(y) \geq -\|(P_S)^{*-1/2} P^*_{SS^c} \eta_{S^c}\|^2 - \frac{1}{2}\log|P_S^*| + \log \int_{P_S^{*-\frac{1}{2}} u \leq \eta_S} \exp\left(-\|u\|^2\right).$$

By another change of variables via $v := P_S^{*-\frac{1}{2}} u - \eta_S$, we get

$$h_{\theta^*,\Sigma^*}(y) \geq -\|(P_S^*)^{-1/2} P^*_{SS^c} \eta_{S^c}\|^2 - \frac{1}{2}\log|P_S^*| + \log \int_{v \leq 0} \left|P_S^{*\frac{1}{2}}\right| \exp\left(-\|P_S^{*\frac{1}{2}}(v + \eta_S)\|^2\right),$$

$$\geq -\|(P_S^*)^{-1/2} P^*_{SS^c} \eta_{S^c}\|^2 - \frac{1}{2}\log|P_S^*| - 2\|P_S^{*\frac{1}{2}} \eta_S\|^2 + \log \int_{v \leq 0} \left|P_S^{*\frac{1}{2}}\right| \exp\left(-2\|P_S^{*\frac{1}{2}} v\|^2\right),$$

$$= -\|(P_S^*)^{-1/2} P^*_{SS^c} \eta_{S^c}\|^2 - \frac{1}{2}\log|P_S^*| - 2\|P_S^{*\frac{1}{2}} \eta_S\|^2 + O(|S|).$$

As $(\Sigma^*_{S^c})^{-1} = P^*_{S^c} - P^*_{S^cS}(P_S^*)^{-1} P^*_{SS^c}$, we have

$$-\|(P_S)^{*-1/2} P^*_{SS^c} \eta_{S^c}\|^2 = \|\eta_{S^c}\|^2_{\Sigma^*_{S^c}} - \|P_{S^c}^{*1/2} \eta_{S^c}\|^2,$$

which gives

$$h_{\theta^*,\Sigma^*}(y) \geq \|\eta_{S^c}\|^2_{\Sigma^*_{S^c}} - \|P_{S^c}^{*\frac{1}{2}} \eta_{S^c}\|^2 - \frac{1}{2}\log|P_S^*| - 2\|P_S^{*\frac{1}{2}} \eta_S\|^2 + O(|S|). \tag{27}$$

From Eqn (26) − Eqn (27), we get

$$h_{\theta,\Sigma}(y) - h_{\theta^*,\Sigma^*}(y) \leq \|P_{S^c}^{*\frac{1}{2}} \eta_{S^c}\|^2 + 2\|P_S^{*\frac{1}{2}} \eta_S\|^2 - \|\eta_{S^c}\|^2_{\Sigma^*_{S^c}} + \frac{1}{2}\log\frac{|P_S^*|}{|P_S|} + O(|S|). \tag{28}$$

$\square$

**Lemma C.4.** *Under Assumption 4.4, C.3, consider $P \in \mathbb{R}^{d \times d}_+$ such that $\frac{\lambda_{\max}(P)}{\lambda_{\min}(P)} \leq \kappa$ and*

$$\frac{\lambda_{\max}(P)}{\lambda_{\max}(P^*)} \geq O\left(\frac{\kappa^3 d^2 n^2}{k^2} + \frac{B^2 n \kappa}{k}\right).$$

*Then, for all $W \in \mathbb{R}^{d \times k}$, and for all $y_i = \phi(W^* x_i + \eta_i)$ with $\|P_{S^c}^{*\frac{1}{2}} \eta_{S^c}\|, \|P_S^{*\frac{1}{2}} \eta_S\| \leq B$, we have*

$$\bar{\gamma}_{W,P} := \frac{1}{n} \sum_{i \in [n]} \gamma_{W x_i, P}(y_i) < 0.$$

*Proof of Lemma C.4.* For each $W \in \mathbb{R}^{d \times k}$, let

$$\theta_i := W x_i.$$

From Eqn (24) in Lemma C.2, for each $i \in [n]$, we have,

$$\gamma_{\theta_i, \Sigma} \leq g_{\theta_i, \Sigma} + \frac{d}{2} \log \frac{\lambda_{\max}(P)}{\lambda_{\min}(P^*)} + 3B^2,$$

$$\leq g_{\theta_i, \Sigma} + \frac{d}{2} \log \frac{\kappa \lambda_{\max}(P)}{\lambda_{\max}(P^*)} + 3B^2.$$

Now consider

$$g_{\theta_i, \Sigma}(y) = -\frac{1}{2} (y_{i, S^c} - \theta_{i, S^c})^T (\Sigma_{S^c})^{-1} (y_{i, S^c} - \theta_{i, S^c}),$$

$$\leq -\frac{1}{2} \|y - \theta\|^2 \lambda_{\min}(\Sigma_{S^c}^{-1}),$$

$$\leq -\frac{1}{2} \|y - \theta\|^2 \lambda_{\min}(\Sigma^{-1}) = -\frac{1}{2} \|y - \theta\|^2 \lambda_{\min}(P),$$

$$\leq -\frac{1}{2} \|y - \theta\|^2 \frac{\lambda_{\max}(P)}{\kappa},$$

where the second inequality comes from the eigenvalue interlacing Theorem, and the last line follows from the condition number assumption on $\Sigma, P$.

By Assumption C.3, there exist at least $\varepsilon^2 n$ samples for a coordinate $j$ such that $(y_i)_j > 0$. Averaging $g_{\theta_i, \Sigma}$, by Lemma A.1, we get that with high probability,

$$\sum_i \|y_i - \theta_i\|^2 \geq \frac{\sigma_j^{*2} k}{2},$$

which gives

$$\frac{1}{n} \sum_i g_{\theta_i, \Sigma}(y_i) \leq -\frac{\sigma_j^{*2} k \lambda_{\max}(P)}{4 n \kappa},$$

$$\leq -\frac{k \lambda_{\max}(P)}{4 n \kappa \lambda_{\max}(P^*)}.$$

This gives

$$\bar{\gamma}_{W, \Sigma} \leq -\frac{\lambda_{\max}(P) k}{4 n \kappa \lambda_{\max}(P^*)} + \frac{d}{2} \log \left( \kappa \cdot \frac{\lambda_{\max}(P)}{\lambda_{\max}(P^*)} \right) + 3B^2,$$

$$\leq -\frac{\lambda_{\max}(P) k}{4 n \kappa \lambda_{\max}(P^*)} + d \sqrt{\kappa \cdot \frac{\lambda_{\max}(P)}{\lambda_{\max}(P^*)}} + 3B^2.$$

Completing the squares, we get

$$\bar{\gamma}_{W, \Sigma} \leq -\left( \sqrt{\frac{\lambda_{\max}(P) k}{4 n \kappa \lambda_{\max}(P^*)}} - \kappa d \sqrt{\frac{n}{k}} \right)^2 + \frac{\kappa^2 d^2 n}{k} + 3B^2.$$

For

$$\frac{\lambda_{\max}(P)}{\lambda_{\max}(P^*)} \geq O\left( \frac{\kappa^3 d^2 n^2}{k^2} + \frac{B^2 n \kappa}{k} \right),$$

the above inequality satisfies

$$\bar{\gamma}_{W, \Sigma} \leq 0.$$

$\square$

**Lemma C.11.** *Assume $P^* := \Sigma^{*-1}$ satisfies Assumption 4.4 with condition number $\kappa$.*

*For all $y = \phi(\theta^* + \eta)$ such that $S$ denotes the zero-coordinates of $y$, and $\eta$ such that $\|P_S^{*\frac{1}{2}}\eta_S\|, \|P_{S^c}^{*\frac{1}{2}}\eta_{S^c}\| \leq B$, consider precision matrices $P$ whose max eigenvalue $\lambda_{\max}(P)$ satisfies*

$$\frac{\lambda_{\max}(P)}{\lambda_{\min}(P^*)} \leq C.$$

*Let $A \geq 4\max\{\frac{d}{2}\log C, 3B^2\}$. Then for $V := (P^{-1})_{S^c}$ and $R_P$ defined as*

$$R_P := 2B\sqrt{C} + \sqrt{\frac{3}{2}A}, \tag{29}$$

*we have*

$$\|\theta_{S^c} - \theta_{S^c}^*\|_V \geq R_P \implies \gamma_{\theta,P} \leq -A.$$

*Proof of Lemma C.11.* Consider Eqn (24) in Lemma C.2. We have

$$\gamma_{\theta,P} \leq g_{\theta,P} + \frac{d}{2}\log\frac{\lambda_{\max}(P)}{\lambda_{\min}(P^*)} + 3B^2,$$

$$\leq g_{\theta,P} + \frac{d}{2}\log C + 3B^2,$$

where the last inequality follows from $\frac{\lambda_{\max}(P)}{\lambda_{\min}(P^*)} \leq C$ in the statement of the Lemma.

By the definition of $g_{\theta,P}$, we have

$$g_{\theta,P} := -\frac{1}{2}(y_{S^c} - \theta_{S^c})^T(P_{S^c} - P_{S^c S}P_S^{-1}P_{SS^c})(y_{S^c} - \theta_{S^c}).$$

We can rewrite the matrix $(P_{S^c} - P_{S^c S}P_S^{-1}P_{SS^c})$ as

$$(P_{S^c} - P_{S^c S}P_S^{-1}P_{SS^c}) = ((P^{-1})_{S^c})^{-1}.$$

By setting

$$V := (P^{-1})_{S^c},$$

we can rewrite $g_{\theta,P}$ as

$$g_{\theta,P} := -\frac{1}{2}\|y_{S^c} - \theta_{S^c}\|_V^2 = -\frac{1}{2}(y_{S^c} - \theta_{S^c})^T V^{-1}(y_{S^c} - \theta_{S^c}).$$

Now, as $y_{S^c} = \eta_{S^c} + \theta_{S^c}^*$, we have

$$g_{\theta,P} = -\frac{1}{2}\|\eta_{S^c} + \theta_{S^c}^* - \theta_{S^c}\|_V^2,$$

$$= -\frac{1}{2}\|\theta_{S^c}^* - \theta_{S^c}\|_V^2 + \|\eta_{S^c}\|_V\|\theta_{S^c}^* - \theta_S\| - \frac{1}{2}\|\eta_{S^c}\|_V^2.$$

Ignoring the $\|\eta_{S^c}\|_V^2$ term, we get

$$g_{\theta,P} \leq -\frac{1}{2}\|\theta_{S^c}^* - \theta_{S^c}\|_V^2 + \|\eta_{S^c}\|_V\|\theta_{S^c}^* - \theta_S\|_V.$$

By the Cauchy-Schwartz inequality and the eigenvalue interlacing theorem, we have

$$\|\eta_{S^c}\|_V \leq \lambda_{\max}^{\frac{1}{2}}(V^{-1}) \cdot \|\eta_{S^c}\|_2 = \frac{\|\eta_{S^c}\|_2}{\lambda_{\min}^{\frac{1}{2}}(V)} = \frac{\|\eta_{S^c}\|_2}{\lambda_{\min}^{\frac{1}{2}}(P_{S^c}^{-1})} \leq \frac{\|\eta_{S^c}\|_2}{\lambda_{\min}^{\frac{1}{2}}(P^{-1})} = \lambda_{\max}^{\frac{1}{2}}(P) \cdot \|\eta_{S^c}\|_2$$

By the statement of the Lemma, we have $\|P_{S^c}^{*\frac{1}{2}}\eta_{S^c}\| \le B \implies \|\eta_{S^c}\|_2 \le \frac{B}{\lambda_{\min}^{\frac{1}{2}}(P_{S^c}^*)}$. Substituting in the above inequality, we get

$$\|\eta_{S^c}\|_V \le \frac{\lambda_{\max}^{\frac{1}{2}}(P) \cdot \|\eta_{S^c}\|_2}{\lambda_{\min}^{\frac{1}{2}}(P_{S^c}^*)} \le \sqrt{C}B.$$

Substituting in the function $g_{\theta,P}$, we get

$$g_{\theta,P} \le -\frac{1}{2}\|\theta_{S^c}^* - \theta_{S^c}\|_V^2 + B\sqrt{C}\|\theta_{S^c}^* - \theta_S\|_V.$$

Hence, for $\theta$ satisfying

$$\|\theta_{S^c} - \theta_{S^c}^*\|_V \ge R_P := 2B\sqrt{C} + 2\sqrt{A},$$

we get

$$\gamma_{\theta,P} \le -A.$$

$\square$

In order to cover our precision matrices, we will consider a subset of matrices whose entries are quantized by an interval size $\beta$:

**Definition C.12** (Quantized Precision Matrices). *For $\kappa > 0$, define $\Omega \subset \mathbb{R}^{d \times d}$ as the set of positive definite matrices with condition number $\kappa$.*

*For $\rho > 0$, define the set $\Omega_\rho \subset \Omega$ as*

$$\Omega_\rho := \left\{ P \in \Omega : \lambda_{\max}(P) \in \left[\frac{\rho}{2}, \rho\right] \right\}$$

*For a quantization size $\beta > 0$, define $\widetilde{\Omega}_{\rho,\beta} \subset \Omega_\rho$ as:*

$$\widetilde{\Omega}_{\rho,\beta} := \{ P \in \Omega_\rho : P_{ij} \in \{-\rho, -\rho(1-\beta), -\rho(1-2\beta), \cdots, \rho(1-2\beta), \rho(1-\beta), \rho\}.\}$$

**Lemma C.5** ($\Sigma$ cover). *For $B > 1$, and $0 < L < U$, let $A > \max\left\{ \sqrt{\log \frac{1}{\varepsilon}}, B^2 U\kappa, \frac{d}{2}\log\left(\frac{\kappa U}{L}\right), 1 \right\}$.*

*Let $P^* := \Sigma^{*-1}$ be the precision matrix of $\eta$. Let $\Omega \subset \mathbb{R}_+^{d \times d}$ denote the set of positive definite matrices $P \in \mathbb{R}_+^{d \times d}$ with condition number $\kappa$ and whose maximum eigenvalue lies in $[L\lambda_{\min}(P^*), U \cdot \lambda_{\max}(P^*)]$.*

*Then, there exists a partition of $\Omega$ of size*

$$\left( \text{poly}\left(A, \frac{1}{\varepsilon}\right) \right)^{d^2}$$

*such that for all $\theta \in \mathbb{R}^d$ and all $y = \phi(\theta^* + \eta) \in \mathbb{R}^d$ with $\|P_S^{*\frac{1}{2}}\eta_S\|, \|P_{S^c}^{*\frac{1}{2}}\eta_{S^c}\| \le B$, and each cell $I$ in the partition, one of the following holds:*

- *for all $P \in I$, $\gamma_{\theta,P}(y) < -A$, or*
- *for all $P, P' \in I$, we have $|\gamma_{\theta,P}(y) - \gamma_{\theta,P'}(y)| \le \epsilon$.*

*Proof.* In order to construct the net over the precision matrices, we will consider geometrically spaced values of $\rho \in [L \cdot \lambda_{\min}(P^*), U \cdot \lambda_{\max}(P^*)]$, and for each $\rho$, we will construct a net over matrices that have max eigenvalue $\le \rho$.

Now consider $\rho > 0$ that lies in the following discrete set:

$$\left\{ \lambda_{\min}(P^*)2^j, j \in \lceil \log_2(\kappa \tfrac{U}{L}) \rceil \right\}$$

This set is a geometric partition over the possible max eigenvalues that the MLE can return.

For the current $\rho$, let $\Omega_\rho$ follow Definition C.12. Now consider $P \in \Omega_\rho$.

**Constructing the interval for which $\gamma_{\theta,P} < -A$.** By Lemma C.11, for $V = (P^{-1})_{S^c}$, and $R_P = O\left(B\sqrt{\frac{\rho}{\lambda_{\min}(P^*)}} + \sqrt{A}\right) = O(\sqrt{A})$, we have

$$\|\theta_{S^c} - \theta^*_{S^c}\|_V \geq R_P \implies \gamma_{\theta,P} < -A.$$

For any $\theta$, notice that the set of matrices $P$ satisfying $\|\theta_{S^c} - \theta^*_{S^c}\|_V \geq R_P$ is connected (as its complement is compact). This forms the set $I$ for which $\gamma_{\theta,P} < -A$.

**Constructing intervals for which $|\gamma_{\theta,P} - \gamma_{\theta,P'}| \leq \varepsilon$.** We will now construct a partition over those $P$ which satisfy $\|\theta_{S^c} - \theta^*_{S^c}\|_V < R_P$, and show that the log-likelihood changes by atmost $\varepsilon$ for each cell in this partition.

If $P \in \Omega_\rho$, then each of its elements $P_{ij} \in [-\rho, \rho]$. For a parameter $\beta > 0$ that we will specify later, consider the partition $\widetilde{\Omega}_{\rho,\beta}$ of $\Omega_\rho$, following Definition C.12. Clearly, the size of $\widetilde{\Omega}_{\rho,\beta}$ can be upper bounded by

$$\left|\widetilde{\Omega}_{\rho,\beta}\right| \leq \left(\frac{2}{\beta}\right)^{d^2}.$$

We will now analyze the effect of rounding down $P \in \Omega_\rho$ to its nearest element in $\widetilde{\Omega}_{\rho,\beta}$.

By Claim C.13, for $\gamma = 2\kappa\beta d^2$, we have

$$(1-\gamma)\|t - \theta\|_\Sigma^2 \leq \|t - \theta\|_{\Sigma'}^2 \leq (1+\gamma)\|t - \theta\|_\Sigma^2, \tag{30}$$

Consider the log-likelihood at $\theta, P'$:

$$f_{\theta,P'}(y) = \frac{1}{2}\log|P'| + \log\int_{t:t_S \leq 0, t_{S^c} = y_{S^c}} \exp\left(-\|t - \theta\|_{\Sigma'}^2\right).$$

We will use the LHS of Eqn (30) to show that

$$f_{\theta,P'}(y) - \frac{1}{2}\log|P'| \leq f_{\theta,P}(y) - \frac{1}{2}\log|P| + \varepsilon,$$

and deal with the $\log|P'|$ term later. The lower bound for the log-likelihood at $P'$ can be obtained via analogous proof using the RHS of Eqn (30).

By the LHS of Eqn (30), we get

$$f_{\theta,P'}(y) - \frac{1}{2}\log|P'| \leq \log\int_{t:t_S \leq 0, t_{S^c} = y_{S^c}} \exp\left(-(1-\gamma)\|t - \theta\|_\Sigma^2\right).$$

Rearranging the terms, we get

$$\begin{aligned} f_{\theta,P'}(y) - \frac{1}{2}\log|P'| \leq & -\frac{(1-\gamma)}{2}\|y_{S^c} - \theta_{S^c}\|_{\Sigma_{S^c}}^2 \\ & + \log\int_{t \leq 0} \exp\left(-\frac{(1-\gamma)}{2}\|P_S^{\frac{1}{2}}(t-\theta)_S + P_S^{-\frac{1}{2}}P_{SS^c}(y_{S^c} - \theta_{S^c})\|^2\right) \end{aligned}$$

The non-integral term corresponds to $g_{\theta,P}$ in Eqn (21), while the integral term corresponds to $h_{\theta,P}$ in Eqn (22).

**Handling the non-integral term.** As we are only considering $\theta$ such that $\|y_{S^c} - \theta_{S^c}\|_{\Sigma_{S^c}} \leq R_P$, we have that for

$$\beta = O\left(\frac{\varepsilon}{R_P^2 d^2 \kappa}\right) = O\left(\frac{\varepsilon}{\text{poly}(A)}\right),$$

the non-integral term corresponds to $g_{\theta,P} + \varepsilon$, which gives

$$f_{\theta,P'}(y) - \frac{1}{2}\log|P'| \leq g_{\theta,P} + \varepsilon + \log\int_{t \leq 0} \exp\left(-\frac{(1-\gamma)}{2}\|P_S^{\frac{1}{2}}(t-\theta)_S + P_S^{-\frac{1}{2}}P_{SS^c}(y_{S^c} - \theta_{S^c})\|^2\right) \tag{31}$$

**Handling the integral.**   Now we consider the integral term. Define the integral

$$I_1 = \log \int_{t \leq 0} \exp\left(-\frac{(1-\gamma)}{2}\|P_S^{\frac{1}{2}}(t-\mu)\|^2\right)$$

for $\gamma = 2d^2\kappa\beta$ and $\mu = \theta_S - P_S^{-1}P_{SS^c}(y_{S^c} - \theta_{S^c})$.

Define the analogous integral that does not have the $(1-\gamma)$ term in the exponential:

$$I_2 = \log \int_{t \leq 0} \exp\left(-\frac{1}{2}\|P_S^{\frac{1}{2}}(t-\mu)\|^2\right)$$

Clearly, $I_1 \geq I_2$.

We only need to consider $\mu$ such that $\|\mu\|_\infty \leq O(\sqrt{A}\rho)$: otherwise the likelihood will be smaller than $-A$.

By Lemma C.14, for $\gamma = O\left(\frac{\varepsilon}{\text{poly}(A)}\right)$, we have

$$I_1 \leq I_2 + \varepsilon.$$

**Handling the log-determinant term.**   Now consider the $\log|P|$ term. As we are decreasing each element by atmost $\beta\rho$, none of the eigenvalues can increase. Moreover, as

$$\text{Tr}(P') - \text{Tr}(P) \geq -d\beta\rho,$$

we can conclude that each eigenvalue decreases by at most $-d\beta\rho$. Also, as $\rho \leq \kappa\lambda_j(P) \ \forall \ j \in [d]$, we can conclude that each eigenvalue satisfies

$$\lambda_j(P') \geq \lambda_j(P)(1 - d\beta\kappa).$$

Hence, the log-determinant satisfies

$$\log|P'| \geq \log|P| + d\log(1 - \beta d\kappa) \geq \log|P| - \frac{d^2\beta\kappa}{1 - d\beta\kappa} \geq \log|P| - O(\varepsilon) \text{ for } \beta \leq \frac{\varepsilon}{\kappa d^2} \leq \frac{\varepsilon}{\kappa r^2 d^2}.$$

This finally gives

$$|\gamma_{\theta,P} - \gamma_{\theta,P'}| \leq O(\varepsilon).$$

**Bounding the size of the net**   As $\beta = O\left(\frac{\varepsilon}{\text{poly}(A)}\right)$, and the max radius is also $O(\text{poly}(A))$, we have a cover of size $\left(\frac{\text{poly}(A)}{\varepsilon}\right)$ per entry of the precision matrix (for a fixed $\Omega_\rho$).

Intersecting the $d^2$ nets means that for each $\Omega_\rho$, we have a net of size

$$\left(\text{poly}\left(A, \frac{1}{\varepsilon}\right)\right)^{d^2}.$$

As we are considering $\text{poly}(A)$ many $\Omega_\rho$s, the size of the net remains the same as the above.

$\square$

**Claim C.13.** *In the setting of Lemma C.5, if $P \in \Omega_\rho$ and $P' \in \widetilde{\Omega}_{\rho,\beta}$ is its nearest neighbor, then for $\gamma = 2\kappa\beta d^2$, we have*

$$(1-\gamma)\|t-\theta\|_\Sigma^2 \leq \|t-\theta\|_{\Sigma'}^2 \leq (1+\gamma)\|t-\theta\|_\Sigma^2, \tag{32}$$

*where $\Sigma := P^{-1}, \Sigma' := P'^{-1}$.*

*Proof.* Consider $P \in \Omega_\rho$ and $P' \in \widetilde{\Omega}_{\rho,\beta}$ such that $P = P' + \Delta$. Since $P'$ is the rounding down of $P$, we have $\Delta_{ij} \in [0, \beta\rho]$.

As $\text{Tr}(\Delta) \in [0, \rho\beta d]$, and $\|\Delta\|_F \leq \rho\beta d$, we have

$$\lambda_{\max}(\Delta) \leq \rho\beta d \text{ and } \lambda_{\min}(\Delta) \geq -\rho\beta d^2.$$

This implies that when considering untruncated Gaussians with precision matrices $P, P'$, we have that for all $t, \theta \in \mathbb{R}^d$,

$$\|t - \theta\|_\Sigma^2 - \rho\beta d^2\|t - \theta\|^2 \leq \|t - \theta\|_{\Sigma'}^2 \leq \|t - \theta\|_\Sigma^2 + \rho\beta d\|t - \theta\|^2.$$

Since $\lambda_{\min}(P) \geq \frac{\rho}{2\kappa}$, we have

$$\rho\|t - \theta\|^2 \leq 2\kappa\|t - \theta\|_\Sigma^2.$$

Substituting in the previous inequality, we get

$$\|t - \theta\|_\Sigma^2 - \rho\beta d^2\|t - \theta\|^2 \leq \|t - \theta\|_{\Sigma'}^2 \leq \|t - \theta\|_\Sigma^2 + \rho\beta d\|t - \theta\|^2,$$
$$\implies (1 - 2\kappa\beta d^2)\|t - \theta\|_\Sigma^2 \leq \|t - \theta\|_{\Sigma'}^2 \leq (1 + 2\kappa\beta d)\|t - \theta\|_\Sigma^2,$$

For the sake of symmetry, we will use the weaker bound of

$$(1 - 2\kappa\beta d^2)\|t - \theta\|_\Sigma^2 \leq \|t - \theta\|_{\Sigma'}^2 \leq (1 + 2\kappa\beta d^2)\|t - \theta\|_\Sigma^2,$$

Setting $\gamma = 2\kappa\beta d^2$ completes the proof. $\qquad\qquad\square$

**Lemma C.14.** *Consider a bounded mean vector $\mu$ with $\|\mu\|_\infty \leq \alpha$ and precision matrix $P$ with max eigenvalue $\rho$ and condition number $\kappa$.*

*For $\gamma = O\left(\min\left\{\frac{\varepsilon}{\alpha\rho^{1/2}d^{3/2}}, \frac{\varepsilon}{\alpha^2 d^3 \rho}\right\}\right)$, we have*

$$\log \int_{t \leq 0} \exp\left(-\frac{(1 - \gamma)}{2}\|P^{\frac{1}{2}}(t - \mu)\|^2\right) \leq \varepsilon + \log \int_{t \leq 0} \exp\left(-\frac{1}{2}\|P^{\frac{1}{2}}(t - \mu)\|^2\right).$$

*Proof of Lemma C.14.* Wlog, consider $\mu \geq 0$. The case where the entries are possibly negative follow a similar proof.

Define the integral on the LHS and RHS of the Lemma statement by $I_1$ and $I_2$ respectively.

By a change of variables, we set $t' = \sqrt{1 - \gamma}(t - \mu) + \mu$ in $I_1$, to get

$$I_1 = \log \frac{1}{\sqrt{1 - \gamma}} + \log \int_{t' \leq (1 - \sqrt{1-\gamma})\mu} \exp\left(-\frac{1}{2}\|P^{\frac{1}{2}}(t' - \mu)\|^2\right).$$

Since $\gamma < 1$, we have $(1 - \sqrt{1 - \gamma})\mu < \gamma\mu$. Substituting in $I_1$, and for $\gamma = O(\varepsilon)$, we get

$$I_1 \leq \log \frac{1}{\sqrt{1 - \gamma}} + \log \int_{t' \leq \gamma\mu} \exp\left(-\frac{1}{2}\|P^{\frac{1}{2}}(t' - \mu)\|^2\right),$$
$$\leq O(\varepsilon) + \log \int_{t' \leq \gamma\mu} \exp\left(-\frac{1}{2}\|P^{\frac{1}{2}}(t' - \mu)\|^2\right).$$

The integrating set in the above inequality can be split into two parts: one over the negative orthant (which is exactly to $e^{I_2}$) and another over the shell

$$C = \{t' \leq \gamma\mu\} \setminus \{t' \leq 0\}.$$

This gives

$$I_1 \leq O(\varepsilon) + \log\left(e^{I_2} + \int_{t' \in C} \exp\left(-\frac{1}{2}\|P^{\frac{1}{2}}(t' - \mu)\|^2\right)\right).$$

In the above inequality, let $e^{I_3}$ denote the integral over the shell $C$. We will now show that $I_3$ satisfies

$$e^{I_3} \leq \varepsilon e^{I_2}.$$

Let $f(x)$ denote the Gaussian density with mean $\mu$ and precision matrix $P$.

For a subset of co-ordinates $S \subseteq [d], S \neq \emptyset$, and $t \in \mathbb{R}^d$, let $x_+, x_- \in \mathbb{R}^d$ be such that

$$x_{+,S}(i) = \begin{cases} \gamma\mu_i & \text{if } i \in S, \\ t_i & \text{if } i \notin S, \end{cases} \quad , \quad x_{-,S}(i) = \begin{cases} -\frac{\gamma}{\varepsilon}\mu_i & \text{if } i \in S, \\ t_i & \text{if } i \notin S. \end{cases}$$

By the monotonicity of the Gaussian density, the integral over the shell $C$ can be upper bounded by breaking up into a sum of integrals over lower-dimensional strips, where for a fixed subset $S \in [d]$, the variables $t_{S^c}$ are integrated over $(-\infty, \gamma\mu_{S^c}]$, while the variables in $S$ are fixed to $\gamma\mu_S$.

This gives

$$
\begin{aligned}
e^{I_3} &\leq \sum_{S \subseteq [d]} \int_{t_{S^c} \leq \gamma\mu_{S^c}} f(x_{+,S}) \prod_{i \in S} \gamma\mu_i, \\
&\leq \sum_{S \subseteq [d]} \int_{t_{S^c} \leq \gamma\mu_{S^c}} f(x_{+,S})(\gamma\alpha)^{|S|}, \\
&\leq \sum_{k=1}^{d} \binom{d}{k}(\gamma\alpha)^k \max_{S \subseteq [d]: |S| = k} \int_{t_{S^c} \leq \gamma\mu_{S^c}} f(x_{+,S}).
\end{aligned}
$$

By Claim C.15, for any $S$, and $\gamma = O\left(\min\left\{\frac{\varepsilon}{\alpha\sqrt{\rho d}}, \frac{\varepsilon}{\alpha^2 d^2 \rho}\right\}\right)$ each summand satisfies

$$
f(x_{+,S}) \leq 2f(x_{-,S})
$$

Furthermore, for any $S \subseteq [d]$, we have

$$
\int_{t_{S^c} \leq \gamma\mu_{S^c}} f(x_{-,S})\left(\frac{\gamma}{\varepsilon}\alpha\right)^{|S|} \leq e^{I_2}.
$$

This gives

$$
e^{I_3} \leq \sum_{k=1}^{d} d^k 2\varepsilon^k e^{I_2} \leq 3\varepsilon d e^{I_2} \quad \text{if } \varepsilon d \leq \frac{1}{3}.
$$

Rescaling $\varepsilon \leftarrow \frac{\varepsilon}{3d}$ completes the proof. $\qquad \square$

**Claim C.15.** *Let $f$ be the Gaussian density with mean $\mu \in [0, \alpha]^d$ and precision matrix $P \in \mathbb{R}^{d \times d}$ with max eigenvalue $\rho$ and condition number $\kappa$.*

*Let $\gamma = O\left(\min\left\{\frac{\varepsilon}{\alpha\sqrt{\rho d}}, \frac{\varepsilon}{\alpha^2 d^2 \rho}\right\}\right)$. For any subset of co-ordinates $S \subseteq [d], S \neq \emptyset$, and $t \in \mathbb{R}^d$, let $x_+, x_- \in \mathbb{R}^d$ be such that*

$$
x_+(i) = \begin{cases} \gamma\mu_i & \text{if } i \in S, \\ t_i & \text{if } i \notin S, \end{cases} \quad , \quad x_-(i) = \begin{cases} -\frac{\gamma}{\varepsilon}\mu_i & \text{if } i \in S, \\ t_i & \text{if } i \notin S. \end{cases}
$$

*we have*

$$
f(x_+) \leq 2f(x_-)
$$

*Proof.* WLOG, let $S$ be a contiguous set such that we can separate the coordinates of $x_+$ and $x_-$ into disjoint sets. For the coordinates belonging to $S$, let $\mu_S$ denote the coordinates of $\mu$ belonging to $\mu$, and $\mu_{S^c}$ the coordinates not belonging to $S$ (similarly for $t_S$ and $t_{S^c}$).

Taking the logarithm on both sides of the claimed inequality, we want to show that

$$
-\frac{1}{2}\left\|P^{\frac{1}{2}}\begin{bmatrix} \gamma\mu_S - \mu_S \\ t_{S^c} - \mu_{S^c} \end{bmatrix}\right\|^2 \leq -\frac{1}{2}\left\|P^{\frac{1}{2}}\begin{bmatrix} -\frac{\gamma}{\varepsilon}\mu_S - \mu_S \\ t_{S^c} - \mu_{S^c} \end{bmatrix}\right\|^2 + \log 2
$$

Let $a$ and $b$ denote the vectors whose norms correspond to the log-densities in the claimed inequality, and let $\delta = a - b$.

This gives

$$
b = P^{\frac{1}{2}}\begin{bmatrix} -\mu_S(1 - \gamma) \\ t_{S^c} - \mu_{S^c} \end{bmatrix}, \qquad a = P^{\frac{1}{2}}\begin{bmatrix} -\mu_S(1 + \frac{\gamma}{\varepsilon}) \\ t_{S^c} - \mu_{S^c} \end{bmatrix}, \qquad \delta := a - b = P^{\frac{1}{2}}\begin{bmatrix} -\mu_S\gamma(\frac{1}{\varepsilon} + 1) \\ 0_{S^c} \end{bmatrix}
$$

We want to show that

$$-\frac{1}{2}\|b\|^2 \leq -\frac{1}{2}\|a\|^2 + \log 2,$$

$$\Leftrightarrow \langle \delta, b \rangle + \frac{1}{2}\|\delta\|^2 \leq \log 2. \tag{33}$$

As $\|P\| \leq \rho$ and $\|\mu\|_\infty \leq \alpha$, we have

$$\|\delta\|_2^2 \leq \rho(\alpha^2|S|)\gamma^2\left(1 + \frac{1}{\varepsilon}\right)^2.$$

For $\gamma = O\left(\frac{\varepsilon}{\alpha\sqrt{\rho d}}\right)$, we get

$$\|\delta\|_2^2 \leq \frac{1}{2}\log 2. \tag{34}$$

Similarly, consider the inner product $\langle \delta, b \rangle$ in Eqn (33). By the trace trick, we get

$$\langle \delta, b \rangle = \text{Tr}(\Delta P),$$

where

$$\Delta = \begin{bmatrix} -\mu_S(1-\gamma) \\ t_{S^c} - \mu_{S^c} \end{bmatrix}\begin{bmatrix} -\mu_S\gamma\left(\frac{1}{\varepsilon}+1\right) \\ 0_{S^c} \end{bmatrix}^T$$

Notice that the diagonal elements of $\Delta$ are all non-negative. This implies that all singular values are non-negative. The trace of $\Delta$ is

$$\text{Tr}(\Delta) = \|\mu_S\|_2^2(1-\gamma)\gamma\left(\frac{1}{\varepsilon}+1\right).$$

Hence, by Von Neumann's trace inequality, we get

$$\langle \delta, b \rangle \leq \text{Tr}(\Delta)\text{Tr}(P) \leq \|\mu_S\|_2^2(1-\gamma)\gamma\left(\frac{1}{\varepsilon}+1\right)\rho d.$$

For $\gamma = O\left(\frac{\varepsilon}{\alpha^2|S|\rho d}\right)$, this gives

$$\langle \delta, b \rangle \leq \frac{1}{2}\log 2. \tag{35}$$

Substituting Eqn (34) and Eqn (35) in Eqn (33) completes the proof. $\qquad\square$

**Lemma C.6** (W-net). *Let $\eta_{S^c}, \eta_S$ be such that*

$$\|P_{S^c}^{*\frac{1}{2}}\eta_{S^c}\| \leq B_1, \|P_S^{*\frac{1}{2}}\eta_S\| \leq B_2,$$

*for $B_1, B_2 \geq 0$.*

*Let $A > \max\{B_1^2, B_2^2, \text{poly}(C, \kappa)\}$. Let $P^* = \Sigma^{*-1}$ be the precision matrix of $\eta$. For a fixed matrix $P \in \mathbb{R}^{d \times d}$ whose condition number satisfies Assumption 4.4 and whose eigenvalues satisfy $\lambda_{\max}(P) \in [e^{-\frac{2A}{d}}\lambda_{\min}(P^*), C\lambda_{\max}(P^*)]$, there exists a partition $\mathcal{I}$ of $\mathbb{R}^d$ with size*

$$\left(\text{poly}\left(A, \frac{1}{\varepsilon}\right)\right)^{3d}$$

*such that for each interval $I \in \mathcal{I}$, we have one of the following:*

- *for all $\theta \in I$, $\gamma_{\theta,P}(y) < -A$, or*
- *for all $\theta, \theta' \in I$, $|\gamma_{\theta,P}(y) - \gamma_{\theta',P}(y)| \leq \epsilon$.*

*Proof of Lemma C.6.* Recall that the log-likelihood ratio $\gamma_\theta$ can be decomposed into the difference of two terms that depend on $\theta$:

$$\gamma_{\theta,P}(y) - \frac{1}{2}\log\frac{|P|}{|P'|} = g_{\theta,P}(y) - g_{\theta^*,P^*}(y) + h_{\theta,P}(y) - h_{\theta^*,P^*}(y).$$

Without loss of generality, consider the net for the first coordinate $\theta_1$. The final net will be the intersection of the per-coordinate nets.

We will construct three partitions: the first is $\mathcal{I}_{h,0}$ for $h$ when $y_1 = 0$, the second is is $\mathcal{I}_{h,1}$ for $h$ when $y_1 > 0$, and the last is $\mathcal{I}_g$ for $g$ when $y_1 > 0$. The final partition will be the intersection of these partitions.

**Case 1: Net over $h$, $y_1 = 0$.** As $y_1 = 0$, we have $1 \in S$. For $\theta \in \mathbb{R}^d$, we have

$$h_{\theta,P}(y) = \log\int_{t\leq 0}\exp\left(-\|P_S^{\frac{1}{2}}(t-\theta_S) + (P_S)^{-1/2}P_{SS^c}(y_{S^c}-\theta_{S^c})\|^2/2\right),$$

By Claim C.16, if $\theta_1 \geq \Theta(\frac{\sqrt{C\kappa A}}{p_1})$, then the log-likelihood is smaller than $-A$.

Now, consider $\theta_1 < O(\frac{\sqrt{C\kappa A}}{p_1})$. Let $\theta' = \theta + \alpha e_1$ for $\alpha > 0$. As $h$ is monotonically decreasing per coordinate, $\theta_1 < \theta_1' \implies h_{\theta,P} \geq h_{\theta',P}$. We would like to now upper bound $h_{\theta,P}$ in terms of $h_{\theta',P}$.

Let

$$\mu := \theta_S + (P_S)^{-1}P_{SS^c}(y_{S^c}-\theta_{S^c}),$$
$$\mu' := \theta_S' + (P_S)^{-1}P_{SS^c}(y_{S^c}-\theta_{S^c})$$

In the function $h_\theta$, break the integrating set into two domains: one where $t - \mu$ is small,

$$\Omega_1 = \left\{t \in \mathbb{R}^{|S|} : \|P_S^{\frac{1}{2}}(t-\mu)\| \leq r\right\},$$

and another where it is large:

$$\Omega_2 = \left\{t \in \mathbb{R}^{|S|} : \|P_S^{\frac{1}{2}}(t-\mu)\| > r\right\},$$

for some $r > 0$ that we will specify later.

Let $I_1$ and $I_2$ denote the integrals over $\Omega_1$ and $\Omega_2$ respectively.

$I_2$ corresponds to the tail of an unnormalized Gaussian distribution, and hence we have

$$h_{\theta,P}(y) = \log\left(I_2 + \int_{t\leq 0, t\in\Omega_1}\exp\left(-\|P_S^{\frac{1}{2}}(t-\mu)\|^2/2\right)\right),$$
$$\text{where } I_2 \leq (2\pi)^{\frac{|S|}{2}}\left|P_S^{-\frac{1}{2}}\right|e^{-r^2}.$$

We can simplify $I_2$ be comparing $|P|$ to $|P^*|$:

$$I_2 \leq (2\pi)^{\frac{|S|}{2}}\frac{\left|P_S^{-1/2}\right|}{\left|P_S^{*-1/2}\right|}\left|P_S^{*-1/2}\right|e^{-r^2} \leq (2\pi)^{\frac{|S|}{2}}\left(\frac{\lambda_{\max}^*}{\lambda_{\min}(P)}\right)^{|S|}\left|P_S^{*-\frac{1}{2}}\right|e^{-r^2},$$

$$\leq (2\pi)^{\frac{|S|}{2}}\left(\kappa e^{\frac{A}{d}}\right)^{|S|}\left|P_S^{*-\frac{1}{2}}\right|e^{-r^2}.$$

By Lemma C.10, we have

$$(2\pi)^{\frac{|S|}{2}}\left|P_S^{*-\frac{1}{2}}\right| \leq e^{h_{\theta^*,P^*}(y)+O(d+B_2^2+B_3^2)}$$

As we are only consider $\theta'$ such that $h_{\theta^*,P^*}(y) - A < h_{\theta',P}(y)$, for

$$r^2 = O(d\log\kappa + A + \log\frac{1}{\varepsilon}) = O(A),$$

we have

$$I_2 \le \varepsilon e^{h_{\theta',P}(y)}$$

Subsituting in $h_{\theta,P}(y)$, we get

$$h_{\theta,P}(y) \le \log\left(\varepsilon e^{h_{\theta',P}(y)} + \int_{t\le 0, t\in\Omega_1} \exp\left(-\|P_S^{\frac{1}{2}}(t-\mu)\|^2/2\right)\right)$$

Now consider the integral $I_1 = \int_{t\le 0, t\in\Omega_1} \exp\left(-\|P_S^{\frac{1}{2}}(t-\mu)\|^2/2\right)$.

By Claim C.17, as $\Omega_1$ is defined for $t$ bounded by $r$, and $\mu - \mu' = \alpha e_1$, we have

$$I_1 \le \exp\left(2\alpha p_1 r + \alpha^2 p_1^2\right) \cdot I_1',$$

$$\text{where } I_1' = \int_{t\le 0, t\in\Omega_1} \exp\left(-\|P_S^{\frac{1}{2}}(t-\mu')\|^2/2\right) \le e^{h_{\theta',P}(y)}.$$

Substituting in the expression for $h_{\theta,P}$, we get

$$h_{\theta,P}(y) \le \log\left(e^{h_{\theta',P}(y)} \cdot \left(\varepsilon + \exp\left(2\alpha p_1 r + \alpha^2 p_1^2\right)\right)\right)$$

As $\log(\varepsilon + e^x) \le \varepsilon + x$ for $x \ge 0$, we have

$$h_{\theta,P}(y) \le h_{\theta',P}(y) + \varepsilon + 2\alpha p_1 r + \alpha^2 p_1^2.$$

Setting $\alpha = O(\frac{\varepsilon}{p_1 r})$, we get

$$h_{\theta,P}(y) \le h_{\theta',P}(y) + 2\varepsilon.$$

This shows that $h_{\theta,P}$ changes by at most $\varepsilon$ for the considered net. We need to defined the other end point for the net. By a similar argument to the positive end point, if $\theta_1 = -O\left(\frac{\sqrt{C\kappa\log(\frac{1}{\varepsilon})}}{p_1}\right)$, the log-likelihood ratio changes by at most $\varepsilon$ until $\theta_1 = -\infty$.

As we are only trying to cover $\theta$ such that $|\theta_1| \le O(\frac{\sqrt{C\kappa A}}{p_1})$, this net has size

$$O\left(\frac{\sqrt{C\kappa A}}{p_1\alpha}\right) = O\left(\frac{\sqrt{C\kappa A}}{p_1}\frac{p_1 r}{\varepsilon}\right) = \frac{A}{\varepsilon}.$$

**Case 2: Net over $h$, $y > 0$.** A similar argument to Case 1 works here as well.

**Case 3: Net over $g$, $y_1 > 0$.** By Lemma C.11, if $|\theta - \theta_*|_{P_{S^c}} > R_1$ for

$$R_1 = O(\sqrt{A}),$$

then

$$g_{\theta,P} - g_{\theta^*,P} < -A.$$

Now consider $\theta$ such that

$$|\theta_1 - \theta_1^*| \le R,$$

and $\theta, \theta'$ such that $\theta - \theta' = \alpha e_1$.

The difference in $g_\theta - g_{\theta'}$ is

$$g_\theta(y) - g_{\theta'}(y) = \eta_{S^c}^T (\Sigma_{S^c})^{-1}(\theta_{S^c} - \theta'_{S^c}) - \frac{1}{2}\|\theta^*_{S^c} - \theta_{S^c}\|^2_{\Sigma_{S^c}} + \frac{1}{2}\|\theta^*_{S^c} - \theta'_{S^c}\|^2_{\Sigma_{S^c}},$$

The second and third terms in the RHS can bounded by observing that

$$|2\theta^* - \theta' - \theta| \leq 2R_1 = O(\sqrt{A}), |\theta' - \theta| \leq \alpha,$$

and hence we get

$$-\frac{1}{2}\|\theta^*_{S^c} - \theta_{S^c}\|^2_{\Sigma_{S^c}} + \frac{1}{2}\|\theta^*_{S^c} - \theta'_{S^c}\|^2_{\Sigma_{S^c}} \leq O(\sqrt{A})\alpha.$$

Now, for the first term in the RHS, we have

$$\|\eta_{S^c}\|_{P^*_{S^c}} \leq B_1 \implies \|\eta_{S^c}\| \leq \frac{B_1}{\lambda_{\min}^{\frac{1}{2}}(P^*)} \leq \frac{B_1\sqrt{\kappa}}{\lambda_{\max}^{\frac{1}{2}}(P^*)}.$$

This further implies that

$$\eta_{S^c}^T(\Sigma_{S^c})^{-1}(\theta_{S^c} - \theta'_{S^c}) \leq \frac{B_1\sqrt{\kappa}}{\lambda_{\max}^{\frac{1}{2}}(P^*)}\sqrt{p_1}\alpha = \text{poly}(A)\alpha.$$

Setting

$$\alpha = O\left(\frac{\epsilon}{\text{poly}(A)}\right),$$

we get

$$|g_\theta(y) - g_{\theta'}(y)| \leq \frac{\epsilon}{d}.$$

As we are covering a set of size $R_1$ using a grid size of $\alpha$, the size of this partition is

$$O(\frac{R_1}{\alpha}) = \frac{\text{poly}(A)}{\varepsilon}.$$

$\square$

**Claim C.16.** *In the setting of Lemma C.6, we have $\lambda_{\max}(P) \leq C\lambda_{\max}(P^*)$. Let $p_1$ denote the first diagonal element of $P$.*

*If $\theta_1 \geq \Theta(\frac{\sqrt{C\kappa A}}{p_1})$, then the function $h_{\theta,P}$ is such that*

$$h_{\theta,P} - h_{\theta^*,P} < -A.$$

*Proof of Claim C.16.* Recall that the function $h_{\theta,P}$ is defined as:

$$h_{\theta,P}(y) = \log \int_{t \leq 0} \exp\left(-\|P_S^{\frac{1}{2}}(t - \theta_S) + (P_S)^{-1/2}P_{SS^c}(y_{S^c} - \theta_{S^c})\|^2/2\right).$$

Consider the term $\|P_S^{\frac{1}{2}}(t - \theta_S) + (P_S)^{-1/2}P_{SS^c}(y_{S^c} - \theta_{S^c})\|$. By the triangle inequality, we have

$$\|P_S^{\frac{1}{2}}(t - \theta_S) + (P_S)^{-1/2}P_{SS^c}(y_{S^c} - \theta_{S^c})\|$$
$$\geq\|P_S^{\frac{1}{2}}(t - \theta_S) + (P_S)^{-1/2}P_{SS^c}(\theta'_{S^c} - \theta^*_{S^c})\| - \|P_S^{-\frac{1}{2}}P_{SS^c}(y_{S^c} - \theta^*_{S^c})\|,$$
$$\geq\|P_S^{\frac{1}{2}}(t - \theta_S) + (P_S)^{-1/2}P_{SS^c}(\theta'_{S^c} - \theta^*_{S^c})\| - \sqrt{C\kappa A},$$

where the last inequality follows as

$$\|P_S^{-\frac{1}{2}}P_{SS^c}(y_{S^c} - \theta_{S^c}^*)\| = \|P_S^{-\frac{1}{2}}P_{SS^c}(\eta_{S^c})\| \le \|P_{S^c}^{\frac{1}{2}}\|\|\eta_{S^c}\| \le \frac{\sqrt{C\lambda_{\max}^*}}{\sqrt{\lambda_{\min}^*}}B \le \sqrt{C\kappa A}.$$

Similarly, the function $g_{\theta,P}$ only considers $\theta$ such that $\|P_{S^c}^{\frac{1}{2}}(\theta_{S^c} - \theta_{S^c}^*)\| \le \sqrt{C\kappa A}$ ( otherwise, the log-likelihood ratio is smaller than $-A$ by virtue of $g_{\theta,P}$, irrespective of $h_{\theta,P}$). For these $\theta$, we have

$$\|P_S^{\frac{1}{2}}(t - \theta_S) + (P_S)^{-1/2}P_{SS^c}(\theta_{S^c}^* - \theta_{S^c})\|$$
$$\ge \|P_S^{\frac{1}{2}}(t - \theta_S)\| - \sqrt{C\kappa A},$$

which gives

$$\|P_S^{\frac{1}{2}}(t - \theta_S) + (P_S)^{-1/2}P_{SS^c}(y_{S^c} - \theta_{S^c})\| \ge \|P_S^{\frac{1}{2}}(t - \theta_S)\| - 2\sqrt{C\kappa A}.$$

Hence, if $\theta_1 \ge O(\frac{\sqrt{C\kappa A}}{p_1})$, then we have

$$\|P_S^{\frac{1}{2}}(t - \theta_S) + (P_S)^{-1/2}P_{SS^c}(y_{S^c} - \theta_{S^c})\| \ge \Omega(\sqrt{C\kappa A}) \ \forall \ t \le 0,$$

and hence the Gaussian integral is at most $(2\pi)^{|S|/2}\left|P_S^{-\frac{1}{2}}\right|e^{-\Omega(C\kappa A)}$.

By Lemma C.10, we have $h_{\theta^*,P^*} \ge -\frac{1}{2}\log|P_S^*| - O(A)$, which gives

$$h_{\theta,P}(y) - h_{\theta^*,P^*}(y) < \frac{1}{2}\log\frac{|P_S^*|}{|P_S|} - \Omega(C\kappa A)$$
$$< \frac{d}{2}\log\frac{\lambda_{\max}^*}{\lambda_{\min}(P)} - \Omega(C\kappa A) < O(A\log\kappa) - \Omega(C\kappa A) = -\Omega(C\kappa A).$$

This gives a contiguous interval over $\theta_1$ for which $\gamma_{\theta,P} < -A$. $\qquad\square$

**Claim C.17.** *In the setting of Lemma C.6, let $\mu, \mu'$ be such that $\mu - \mu' = \alpha e_1$*

*Then, for all $t$ such that*

$$\|P_S^{\frac{1}{2}}(t - \mu)\| \le r,$$

*and $p_1 := P_{11}$, we have*

$$\|P_S^{\frac{1}{2}}(t - \mu)\|^2 \ge -2\alpha p_1 r - \alpha^2 p_1^2 + \|P_S^{\frac{1}{2}}(t - \mu')\|^2.$$

*Proof of Claim C.17.* Consider the term $\|P_S^{\frac{1}{2}}(t - \mu)\|^2$.

Adding and subtracting $\|P_S^{\frac{1}{2}}(t - \mu')\|^2$, we get

$$\|P_S^{\frac{1}{2}}(t - \mu)\|^2 = \|P_S^{\frac{1}{2}}(t - \mu)\|^2 - \|P_S^{\frac{1}{2}}(t - \mu')\|^2 + \|P_S^{\frac{1}{2}}(t - \mu')\|^2,$$
$$= \langle P_S^{\frac{1}{2}}(2t - \mu' - \mu), P_S^{\frac{1}{2}}(\mu' - \mu)\rangle + \|P_S^{\frac{1}{2}}(t - \mu')\|^2,$$
$$= \langle P_S^{\frac{1}{2}}(2t - 2\mu - (\mu' - \mu)), P_S^{\frac{1}{2}}(\mu' - \mu)\rangle + \|P_S^{\frac{1}{2}}(t - \mu')\|^2,$$
$$= 2\langle P_S^{\frac{1}{2}}(t - \mu), P_S^{\frac{1}{2}}(\mu' - \mu)\rangle - \|P_S^{\frac{1}{2}}(\mu - \mu')\|^2 + \|P_S^{\frac{1}{2}}(t - \mu')\|^2.$$

As $\mu - \mu' = \alpha e_1$, we have

$$\|P_S^{\frac{1}{2}}(\mu - \mu')\|^2 = \alpha^2 p_1^2.$$

By the Cauchy-Schwartz inequality, and since $\mu - \mu' = \alpha e_1$, the inner product can be lower bounded as

$$2\langle P_S^{\frac{1}{2}}(t - \mu), P_S^{\frac{1}{2}}(\mu' - \mu)\rangle \ge -2\|P_S^{\frac{1}{2}}(t - \mu)\|\|P_S^{\frac{1}{2}}(\mu' - \mu)\| \ge -2\alpha p_1 r.$$

Substituting, we get

$$\|P_S^{\frac{1}{2}}(t - \mu)\|^2 \ge -2\alpha p_1 r - \alpha^2 p_1^2 + \|P_S^{\frac{1}{2}}(t - \mu')\|^2.$$

This completes the proof.

$\qquad\square$

**Lemma C.18.** *Following Definition C.12 let $\Omega_\rho$ be the set of precision matrices with condition number $\kappa$ satisfying $\lambda_{\max}(P) \in [\frac{\rho}{2}, \rho]$, and let $\widetilde{\Omega}_{\rho,\beta}$ be the quantized net with quantization level $\beta$.*

*For any $P \in \Omega_\rho$, let $\widetilde{P} \in \widetilde{\Omega}_{\rho,\beta}$ be its element-wise rounding down. Then, for any $\mu \in \mathbb{R}^d$, we have*

$$d_{TV}(\mathcal{N}(\mu; P), \mathcal{N}(\mu; \widetilde{P})) \leq O(d^2 \beta \kappa). \tag{36}$$

*Proof of Lemma C.18.* Let $\Sigma = P^{-1}$ and $\widetilde{\Sigma} = \widetilde{P}^{-1}$.

By Theorem 1.1 in [16], the TV between two Gaussians with the same mean is

$$d_{TV}(\mathcal{N}(\mu; \Sigma), \mathcal{N}(\mu; \widetilde{\Sigma})) = \Theta\left(\min\left\{1, \sqrt{\sum_i \xi_i^2}\right\}\right),$$

where $\xi_i$ are the eigenvalues of $\widetilde{\Sigma}^{-1}\Sigma - I_d$.

We can convert the bound on the eigenvalues to the Frobenius norm of $\widetilde{\Sigma}^{-1}\Sigma - I_d$:

$$\sqrt{\sum_i \xi_i^2} \leq \|\widetilde{\Sigma}^{-1}\Sigma - I_d\|_F.$$

Recall that $\widetilde{P}$ is the rounding down per entry of $P$. Hence,

$$\widetilde{\Sigma}^{-1}\Sigma - I_d = (\Sigma^{-1} - [\nu_{ij}])\Sigma - I_d, \text{ where } 0 \leq \nu_{ij} < \beta\rho,$$
$$= -\nu\Sigma.$$

Taking the Frobenius norm, we get

$$\|\widetilde{\Sigma}^{-1}\Sigma - I_d\|_F = \|\nu\Sigma\|_F,$$
$$\leq (d\beta\rho)(d\rho_{\max}(\Sigma)) = (d\beta\rho)\left(\frac{d}{\rho_{\min}(P)}\right),$$
$$\leq (d\beta\rho)\left(\frac{d\kappa}{\rho_{\max}(P)}\right) \leq 2d^2\beta\kappa.$$

where the first inequality follows as each element of $\nu$ is at most $\beta\rho$, the second inequality follows as $P$ has condition number $\kappa$, and the third follows as $P \in \Omega_\rho \implies \rho_{\max}(P) \geq \frac{\rho}{2}$.

This completes the proof. $\qquad\square$

**Lemma C.7.** *Let $x_1, \ldots, x_n$ be fixed, and $y_i = \phi(W^* x_i + \eta_i)$ for $\eta_i \sim \mathcal{N}(0, \Sigma^*)$, and $W^* \in \mathbb{R}^{d \times k}$ with $\Sigma^* \in \mathbb{R}^{d \times d}$ satisfying Assumption 4.4 and Assumption C.3. For a sufficiently large constant $C > 0$,*

$$n = C \cdot \frac{(d^2 + kd)}{\varepsilon^2} \log \frac{kd\kappa}{\varepsilon}$$

*samples suffice to guarantee that with high probability, the MLE $\widehat{W}, \widehat{\Sigma}$ satisfies*

$$\widetilde{d}\left((\widehat{W}, \widehat{\Sigma}), (W^*, \Sigma^*)\right) \leq \varepsilon.$$

*Proof of Lemma C.7.* For any $W \in \mathbb{R}^{d \times k}$, $\Sigma \in \mathbb{R}^{d \times d}$ and a sample $(x_i, y_i)$, let $p_{i,W,\Sigma}(y|x_i)$ be the conditional distribution of $y = \phi(Wx + \eta)$, and let $\gamma_{i,W,\Sigma}$ be the log-likelihood ratio between $(W, \Sigma)$ and $(W^*, \Sigma^*)$ on this sample:

$$\gamma_{i,W,\Sigma}(y) := \log \frac{p_{i,W,\Sigma}(y \mid x_i)}{p_{i,W^*,\Sigma^*}(y \mid x_i)}.$$

Then

$$\mathbb{E}_y[\gamma_{i,W,\Sigma}(y)] = -KL(p_{i,W^*,\Sigma^*}(y \mid x_i) \| p_{i,W,\Sigma}(y \mid x_i)).$$

**Concentration.** From Lemma B.1, we see that if $d_{TV}((W^*, \Sigma^*), (W, \Sigma)) \geq \varepsilon$, then for $n \geq O(\frac{1}{\varepsilon^2} \log \frac{1}{\delta})$,

$$\overline{\gamma}_{W,\Sigma} := \frac{1}{n} \sum_{i=1}^{n} \gamma_{i,W,\Sigma}(y_i) < -\frac{\varepsilon^2}{2}, \tag{37}$$

with probability $1 - \delta$.

Of course, whenever $\overline{\gamma}_{W,\Sigma} < 0$, the likelihood under $W^*, \Sigma^*$ is larger than the likelihood under $W, \Sigma$. Thus, for each *fixed* $W, \Sigma$ with $d_{TV}((W^*, \Sigma^*), (W, \Sigma)) \geq \varepsilon$, maximizing likelihood would prefer $W^*, \Sigma^*$ to $W, \Sigma$ with probability $1 - \delta$ if $n \geq O(\frac{1}{\varepsilon^2} \log \frac{1}{\delta})$.

Nothing above is specific to our ReLU-based distribution. But to extend to the MLE over all $W, \Sigma$, we need to build a net using properties of our distribution.

**Building a net.** First, for a given sample $y_i$, let $S_i$ be the set of coordinates of $y_i$ that are zero, and $S_i^c$ be its complement. Then, with high probability,

$$\|P_{S_i}^{*\frac{1}{2}}(\eta_i)_{S_i}\|, \|P_{S_i^c}^{*\frac{1}{2}}(\eta_i)_{S_i}\| \leq B = O(\sqrt{\kappa d \log n}) \; \forall \, i \in [n],$$

where $P^* = \Sigma^{*-1}$, and $P_S^*, P_{S^c}^*$ are the block matrices in $P^*$.

Supposing the above event happens, we will construct a net over the precision matrices $P = \Sigma^{-1}$. Note that as we are only considering matrices with bounded condition number, this is a bijective mapping.

**Net over $\Sigma^{-1}$.** By Lemma C.4, any precision matrix $P = \Sigma^{-1}$ satisfying Assumption 4.4 and whose max eigenvalue satisfies

$$\lambda_{\max}(P) \geq U \cdot \lambda_{\max}(P^*),$$

for $U = O\left(\frac{\kappa^3 d^2 n^2}{k^2} + \frac{\kappa^2 dn \log n}{k}\right)$, will have $\bar{\gamma}_{W,P^{-1}} < 0$, irrespective of $W$.

Similarly, by Lemma C.2, any precision matrix satisfying Assumption 4.4 and whose max eigenvalue satisfies

$$\lambda_{\max}(P) \leq L \cdot \lambda_{\min}(P^*)$$

for $L = e^{-O(\kappa \log n)}$ has $\gamma_{i,W,P^{-1}} \leq 0$ for all $i \in [n]$, and hence its average $\bar{\gamma}_{W,P^{-1}}$ is also $< 0$.

This shows that for all precision matrices $P$ whose max eigenvalue is extremely small / large when compared to the min / max eigenvalues of $P^*$, has

$$\bar{\gamma}_{W,P^{-1}} < 0,$$

irrespective of $W$, and the MLE, which has non-negative $\bar{\gamma}$, will never pick these $P$.

Let $A = \text{poly}(n, d, \kappa, \frac{1}{\varepsilon})$ be large enough such that $A > n(d \log(U\kappa) + B^2)$, and such that it meets the requirements of Lemma C.5 and Lemma C.6.

Then, by Lemma C.5 with $U = \text{poly}(d, \kappa, n)$ and $\log(\frac{1}{L}) = \text{poly}(\kappa, n)$, there exists a partition $\mathcal{P}$ of precision matrices whose max-eigenvalue lies in $[L \cdot \lambda_{\max}(P^*), U \cdot \lambda_{\max}(P^*)]$ into $\left(\text{poly}\left(d, \kappa, n, \frac{1}{\varepsilon}\right)\right)^{d^2}$ cells, such that for each cell $I \in \mathcal{P}$, and $P, P' \in I$, the following holds for all $i \in [n]$ and $W \in \mathbb{R}^{d \times k}$:

$$|\gamma_{i,W,P}(y_i) - \gamma_{i,W,P'}(y_i)| \leq \frac{\varepsilon^2}{16} \tag{38}$$

or $\gamma_{i,W,P}(y_i) < -A$.

Using Lemma C.18, we also have that for all $W$,

$$d_{TV}((W, P), (W, P')) \leq \frac{\varepsilon^2}{16}. \tag{39}$$

We can choose a net $N$ consisting of precision matrices from each cell in $\mathcal{P}$. This net has size

$$\log|N| \lesssim d^2 \log\left(\frac{d\kappa n}{\varepsilon}\right).$$

This gives a sufficient net over the precision matrices.

Now we will construct a net over $W$ for each precision matrix in the net.

$W$**-net.** Now, for each $\widetilde{P} \in N_P$, by Lemma C.6, for each $i \in [n]$, there exists a partition $\mathcal{P}_{\widetilde{P},i}$ of $\mathbb{R}^d$ into $\left(\mathrm{poly}\left(d, k, \kappa, n, \frac{1}{\varepsilon}\right)\right)^d$ cells such that for each cell $I \in \mathcal{P}_{\widetilde{P},i}$, and $W, W' \in I$, one of the following holds:

$$\left|\gamma_{i,W,\widetilde{P}}(y_i) - \gamma_{i,W',\widetilde{P}}(y_i)\right| \le \frac{\varepsilon^2}{16} \tag{40}$$

or $\gamma_{i,W,\widetilde{P}}(y_i) < -A$.

Let $W_j$ be the $j$-th row of $W$. The individual partitions $\mathcal{P}_{\widetilde{P},i}$ on $\langle x_i, W_j\rangle$ induce a partition $\mathcal{P}_{\widetilde{P},i,j}$ on $\mathbb{R}^k$, where $W_j, W'_j$ lie in the same cell of $\mathcal{P}_{\widetilde{P},i,j}$ if $\langle x_i, W_j\rangle$ and $\langle x_i, W'_j\rangle$ are in the same cell of $\mathcal{P}_{\widetilde{P},i}$ for all $i \in [n]$. Since $\mathcal{P}_{\widetilde{P},i,j}$ is defined by $n$ sets of $\left(\mathrm{poly}\left(d, k, \kappa, n, \frac{1}{\varepsilon}\right)\right)$ parallel hyperplanes in $\mathbb{R}^k$, the number of cells in $\mathcal{P}_{\widetilde{P},i,j}$ is

$$\left(\mathrm{poly}\left(d, k, \kappa, n, \frac{1}{\varepsilon}\right)\right)^k.$$

As there are $d$ rows in $W$, we can intersect $\mathcal{P}_{\widetilde{P},i,j}$ over $j \in [d]$, which induces $\mathrm{poly}(d, k, \kappa, n, \frac{1}{\varepsilon})^{kd}$ cells in $\mathbb{R}^k$. We choose a net $N_{\widetilde{P}}$ to contain, for each cell in $\bigcap_{j\in[d]} \mathcal{P}_{\widetilde{P},i,j}$, the $W$ in the cell maximizing $d_{TV}((W^*, P^*), (W, \widetilde{P}))$. This has size

$$\log\left|N_{\widetilde{P}}\right| \lesssim kd \log\left(\frac{\kappa dkn}{\varepsilon}\right).$$

**Proving MLE works.** By (37), for our $n \ge O\left(\frac{(kd+d^2)}{\varepsilon^2} \log \frac{kd\kappa}{\varepsilon}\right)$, we have with high probability that

$$\overline{\gamma}_{W,P} \le -\frac{\varepsilon^2}{2},$$

for all $P \in N$ and for all $W \in N_{\widetilde{P}}$ with $d_{TV}((W^*, P^*), (W, P)) \ge \varepsilon$. Suppose that both this happens, and

$$\|P_{S_i}^{*\frac{1}{2}}(\eta_i)_{S_i}\|, \|P_{S_i^c}^{*\frac{1}{2}}(\eta_i)_{S_i}\| \le B = O(\sqrt{\kappa d \log n}) \,\forall\, i \in [n].$$

We claim that the MLE $\widehat{W}, \widehat{\Sigma}$ must have $d_{TV}((W^*, \Sigma^*), (\widehat{W}, \widehat{\Sigma})) < \frac{17}{16}\varepsilon$.

Consider any $W \in \mathbb{R}^{d\times k}$ and $P \in \mathbb{R}^{d\times d}$ with $d_{TV}((W^*, P^*), (W, P)) \ge \frac{17}{16}\varepsilon$. Using our net on precision matrices, we can find $\widetilde{P} \in N$ such that

$$d_{TV}((W^*, P^*), (W, \widetilde{P})) \ge d_{TV}((W^*, P^*), (W, P)) - d_{TV}((W, P), (W, \widetilde{P})).$$

Recall that we are only currently considering $W, P$ such that $d_{TV}((W^*, P^*), (W, P)) \ge \frac{17}{16}\varepsilon$. By Eqn (39), we have $d_{TV}((W, P), (W, \widetilde{P})) \le \frac{\varepsilon^2}{16}$, which gives

$$d_{TV}((W^*, P^*), (W, \widetilde{P})) \ge \varepsilon\frac{17}{16} - \frac{\varepsilon^2}{16} \ge \varepsilon.$$

Now, for this $\widetilde{P}$, we can find a $\widetilde{W} \in N_{\widetilde{P}}$, and by our choice of $N_{\widetilde{P}}$, we know that

$$d_{TV}((W^*, P^*), (\widetilde{W}, \widetilde{P})) \ge d_{TV}((W^*, P^*), (W, \widetilde{P})) \ge \varepsilon,$$

and by (37), we have $\overline{\gamma}_{\widetilde{W},\widetilde{P}} \le -\frac{\varepsilon^2}{2}$.

Now we consider two cases. In the first case, there exists $i$ with $\gamma_{i,W,P}(y_i) < -A$. Then

$$\overline{\gamma}_{W,P} = \frac{1}{n}\sum_i \gamma_{i,W,P}(y_i) \le -\frac{A}{n} + B^2/2 < 0.$$

Otherwise, by Eqn (38) and Eqn (40), we have

$$\overline{\gamma}_{W,P} \le \overline{\gamma}_{\widetilde{W},\widetilde{P}} + \left|\overline{\gamma}_{\widetilde{W},\widetilde{P}} - \overline{\gamma}_{W,\widetilde{P}}\right| + \left|\overline{\gamma}_{W,\widetilde{P}} - \overline{\gamma}_{W,P}\right|,$$

$$\le -\frac{\varepsilon^2}{2} + \max_i\left|\overline{\gamma}_{i,\widetilde{W},\widetilde{P}} - \overline{\gamma}_{i,W,\widetilde{P}}\right| + \max_i\left|\overline{\gamma}_{i,W,\widetilde{P}} - \overline{\gamma}_{i,W,P}\right|,$$

$$\le -\frac{\varepsilon^2}{2} + \frac{\varepsilon^2}{16} + \frac{\varepsilon^2}{16} < 0.$$

In either case, $\overline{\gamma}_{W,P} < 0$ and the likelihood under $w^*$ exceeds that under $w$. Hence the MLE $\widehat{w}$ must have $d_{TV}(w^*, w) \le \frac{17}{16}\varepsilon$. Rescaling $\varepsilon$ gives the conclusion of the Lemma.

$\square$

**Lemma C.8.** *Let $\{x_i\}_{i=1}^n$ be i.i.d. random variables such that $x_i \sim \mathcal{D}_x$.*

*Let $P^* := \Sigma^{*-1}$. Let $\lambda_{\min}^*, \lambda_{\max}^*$ be the minimum and maximum eigenvalues of $P^*$. For $0 < L < U$, let $\Omega$ denote the following set of precision matrices*

$$\Omega := \left\{ P \in \mathbb{R}_+^{d \times d} : \frac{\lambda_{\max}(P)}{\lambda_{\min}(P)} \le \kappa \text{ and } \lambda_{\max}(P) \in [L \cdot \lambda_{\min}^*, U \cdot \lambda_{\max}^*] \right\}.$$

*Then, for a sufficiently large constant $C > 0$, and for*

$$n = C \cdot \left(\frac{kd + d^2}{\varepsilon^2}\right) \log\left(\frac{kd\kappa}{\varepsilon}\log\left(\frac{U}{L}\right)\right),$$

*we have:*

$$\Pr_{x_i \sim \mathcal{D}_x}\left[\sup_{W \in \mathbb{R}^{d \times k}, P \in \Omega}\left|\widetilde{d}((W,P),(W^*,P^*)) - d((W,P),(W^*,P^*))\right| > \varepsilon\right] \le e^{-\Omega(n\varepsilon^2)}.$$

*Proof of Lemma C.8.* For $P = \Sigma^{-1}$ and $P^* = \Sigma^{*-1}$, let

$$f(W,P) := d((W,\Sigma),(W^*,\Sigma^*))$$

and

$$f_n(W,P) := \widetilde{d}((W,\Sigma),(W^*,\Sigma^*)) = \frac{1}{n}\sum_i [d_{TV}(p_{W,P}(y|x_i), p_{W^*,P^*}(y|x_i))].$$

Since the function is bounded, for any fixed $W, P$, the Chernoff bound gives

$$\Pr[|f_n(W,P) - f(W,P)| > \alpha] \le e^{-2n\alpha^2}. \tag{41}$$

for any $\alpha > 0$. The challenge lies in constructing a net to be able to union bound over $\mathbb{R}^k$ without assuming any bound on $W$ or the covariate $x$. As before, we do so by constructing a "ghost" sample, symmetrizing, and constructing a net based on these samples.

**Ghost sample.** First, we construct a "ghost" dataset $D_x'$ consisting of $n$ fresh samples IID samples $\{x_i'\}_{i \in [n]}$ of $\mathcal{D}_x$. This gives another metric

$$f_n'(W,P) := \widetilde{d}'((W,\Sigma),(W^*,\Sigma^*)) = \frac{1}{n}\sum_i [d_{TV}(p_{W,P}(y|x_i'), p_{W^*,P^*}(y|x_i'))].$$

Similar to the proof in Lemma 4.3, it is sufficient to consider the difference between $f_n(W,P)$ and $f_n'(W,P)$ i.e.,

$$\Pr\left[\sup_{W,P}|f(W,P) - f_n(W,P)| > \varepsilon\right] \le 2\Pr\left[\sup_{W,P}|f_n(W,P) - f_n'(W,P)| > \varepsilon/2\right]. \tag{42}$$

**Symmetrization.** Since $D_x$ and $D'_x$ each have $n$ independent samples, we could instead draw the datasets by first sampling $2n$ elements $x_1, \ldots, x_{2n}$ from $\mathcal{D}_x$, then randomly partition this sample into two equal datasets. Let $s_i \in \{\pm 1\}$ so $s_i = 1$ if $z_i$ lies in $D'_x$ and $-1$ if it lies in $D_x$. Then

$$f_n(W, P) - f'_n(W, P) = \frac{1}{n} \sum_{i=1}^{2n} s_i \cdot d_{TV}(p_{W,P}(y|x_i), p_{W^*,P^*}(y|x_i)).$$

For a fixed $W, P$ and $x_1, \ldots, x_{2n}$, the random variables $(s_1, \ldots, s_{2n})$ are a permutation distribution, so negatively associated. Then the variables $s_i \cdot d_{TV}(p_{W,P}(y|x_i), p_{W^*,P^*}(y|x_i))$ are monotone functions of $s_i$, so also negatively associated. They are also bounded in $[-1, 1]$. Hence we can apply a Chernoff bound:

$$\Pr[|f_n(W, P) - f'_n(W, P)| > \varepsilon] < e^{-n\varepsilon^2/2}, \tag{43}$$

for any fixed $W, P$.

**Constructing a net.** We will first construct a net over the precision matrices $P$ (independent of $W$), and then for each element in the $P$-net, we will construct a net over $W$.

**Net over $\Sigma^{-1}$.** In the following, $\lambda_{\max}(P)$ denotes max eigenvalue of a matrix $P$, $\lambda_{\min}(P)$ denotes the min eigenvalue, and $\lambda_i(P)$ denotes the $i$-th eigenvalue, in decreasing order.

In order to construct the net over the precision matrices, we will consider geometrically spaced values of $\lambda \in [L \cdot \lambda_{\min}(P^*), U \cdot \lambda_{\max}(P^*)]$, and for each $\lambda$, we will construct a net over matrices that have max eigenvalue $\leq \lambda$.

Now consider $\lambda > 0$ that lies in the following discrete set:

$$\left\{ \lambda_{\min}(P^*) 2^j, j \in \lceil \log_2(\kappa \tfrac{U}{L}) \rceil \right\}$$

This set is a geometric partition over the possible max eigenvalues that the MLE can return.

Following definition C.12, let $\Omega_\lambda$ denote the subset of positive definite matrices in $\mathbb{R}^{d \times d}$ that have condition number $\kappa$ and max-eigenvalue in $\left[\frac{\lambda}{2}, \lambda\right]$. Similarly, following Definition C.12, let $\widetilde{\Omega}_{\lambda,\beta}$ denote the gridded version of $\Omega_\lambda$, where entries in the matrix are multiples of $\lambda\beta$.

For any $P \in \Omega_\lambda$, let $\widetilde{P} \in \widetilde{\Omega}_{\lambda,\beta}$ be the matrix obtained by rounding down every element in $P$.

By the Data Processing Inequality, for any $W \in \mathbb{R}^{d \times k}$, we have

$$d_{TV}\left(p_{W,P}(y|x), p_{W,\widetilde{P}}(y|x)\right) \leq d_{TV}(\mathcal{N}(Wx; P), \mathcal{N}(Wx; \widetilde{P})).$$

By Lemma C.18, we can upper bound the RHS of the above inequality by

$$d_{TV}(\mathcal{N}(Wx; P), \mathcal{N}(Wx; \widetilde{P})) \leq O(d^2 \beta \kappa).$$

Setting

$$\beta = O\left(\frac{\varepsilon}{d^2 \kappa}\right),$$

we have a partition of size $O\left((d^2\kappa/\varepsilon)^{d^2}\right)$ per $\lambda$ such that:

$$d_{TV}\left(p_{W,P}(y|x), p_{W,\widetilde{P}}(y|x)\right) \leq O(\varepsilon).$$

We will now construct a net over $W$, so as to show Eqn (43) for all $W, P$.

**$W$-net.** By repeated triangle inequalities, we have

$$|f_n(W, P) - f'_n(W, P)| \leq \left|f_n(W, P) - f_n(W, \widetilde{P})\right| + \left|f_n(W, \widetilde{P}) - f'_n(W, \widetilde{P})\right| + \left|f'_n(W, \widetilde{P}) - f'_n(W, P)\right|.$$

Using the cover over $P$, the first and last term on the RHS are $O(\varepsilon)$. This gives

$$|f_n(W, P) - f'_n(W, P)| \leq O(\varepsilon) + \left|f_n(W, \widetilde{P}) - f'_n(W, \widetilde{P})\right|. \tag{44}$$

For a fixed $\widetilde{W}, \widetilde{P}$, we will have (43) using a Chernoff bound. Since $\widetilde{P}$ is already finite, we will now construct a net over $W$ *for each* $\widetilde{P}$.

It is sufficient to bound $d_{TV}(p_{W,\widetilde{P}}(y|x_i), p_{W',\widetilde{P}}(y|x_i))$ as the triangle inequality implies that this is larger than the RHS above.

We want, for any $W, W'$ in a cell,

$$|d_{TV}(p_{W,\widetilde{P}}(y|x_i), p_{W^*,P^*}(y|x_i)) - d_{TV}(p_{W',\widetilde{P}}(y|x_i), p_{W^*,P^*}(y|x_i))| \leq O(\varepsilon).$$

for all $i \in [2n]$. It is also sufficient to bound $d_{TV}(p_{W,\widetilde{P}}(y|x_i), p_{W',\widetilde{P}}(y|x_i))$ as the triangle inequality implies that this is larger than the left hand side above.

Lemma C.19 implies that we can find $\mathcal{O}\left(\frac{d}{\epsilon^{3/2}}\sqrt{\log(\frac{2d}{\epsilon})}\right)$ per row of $W$ such that for any $W, W'$ in a cell, either

$$d_{TV}(p_{W,\widetilde{P}}(y|x_i), p_{W^*,P^*}(y|x_i)) \geq d_{TV}(p_{W,\widetilde{P}}(y_j|x_i), p_{W^*,P^*}(y_j|x_i)) \geq 1 - \varepsilon$$

which implies

$$d_{TV}(p_{W,\widetilde{P}}(y|x_i), p_{W',\widetilde{P}}(y|x_i)) \leq \epsilon$$

or

$$d_{TV}(p_{W,\widetilde{P}}(y|x_i), p_{W',\widetilde{P}}(y|x_i) \leq \sum_j d_{TV}(p_{W,\widetilde{P}}(y|x_i)_j|z_i, p_{W',\widetilde{P}}(y_j|x_i)) \leq \epsilon/d.$$

Therefore, for each $i$ regardless of the value of $W_j^* z_i$ there are at most $\mathcal{O}\left(\frac{d}{\epsilon^{3/2}}\sqrt{\log(\frac{2d}{\epsilon})}\right)$ partitioning hyperplanes.

We then take the intersection of all $2n$ partitions (for each data point $z_i$). The cells of this partition are defined by $2n$ sets of $\mathcal{O}\left(\frac{d}{\epsilon^{3/2}}\sqrt{\log(\frac{2d}{\epsilon})}\right)$ parallel hyperplanes. Since $z \in \mathbb{R}^k$, the number of cells is at most $\mathcal{O}\left(\left(\frac{nd}{\epsilon^{3/2}}\sqrt{\log\left(\frac{2d}{\epsilon}\right)}\right)^k\right)$.

Hence the total number of cells for $d$ rows is at most $\mathcal{O}\left(\left(\frac{nd}{\epsilon^{3/2}}\sqrt{\log\left(\frac{2d}{\epsilon}\right)}\right)^{dk}\right)$.

**Putting everything together.** Finally, for any $W \in \mathbb{R}^d$ let $\widetilde{W} \in N_{\widetilde{P}}$ be the representative of its cell. Recall that each representative $\widetilde{P}$ of $P$ induces a different cover $N_{\widetilde{P}}$ over $W$. Let $N$ be the net over the precision matrices $P$.

By definition of the cells,

$$\left|d_{TV}(p_{W,\widetilde{P}}(y|x_i), p_{W^*,P^*}(y|x_i)) - d_{TV}(p_{\widetilde{W},\widetilde{P}}(y|x_i), p_{W^*,\Sigma^*}(y|x_i))\right| < O(\varepsilon).$$

for all $i \in [2n]$. Thus

$$\left|\left(f_n'(W, \widetilde{P}) - f_n(W, P)\right) - \left(f_n'(\widetilde{W}, \widetilde{P}) - f_n(\widetilde{W}, P)\right)\right| \leq O(\varepsilon).$$

and so

$$\Pr[\sup_{W \in \mathbb{R}^{d \times k}, P \in \mathbb{R}^{d \times d}} |f_n'(W, P) - f_n(W, P)| > \varepsilon]$$

$$\leq \Pr\left[\max_{w \in N_P, P \in N} |f_n'(W, P) - f_n(W, P)| > \frac{\varepsilon}{4}\right]$$

$$\leq e^{\log|N| + \log|N_P| - (\frac{\varepsilon}{4})^2 n/2}$$

As there are $\log \kappa \frac{U}{L}$ partitions over $P$ (corresponding to the maximum possible eigenvalue of $P$), each with $(O(\frac{d^2 \kappa}{\varepsilon}))^{d^2}$ elements, we have

$$\log|N| \lesssim d^2 \log\left(\frac{d^2 \kappa}{\varepsilon}\right) + \log\log\left(\kappa \frac{U}{L}\right).$$

and each cover $N_P$ over $W$ has size

$$\log |N_P| = 2kd \log \left( \frac{d}{\epsilon^{3/2}} \sqrt{\log\left(\frac{2d}{\epsilon}\right)} \right).$$

This implies that

$$n = C \cdot \left( \frac{kd + d^2}{\varepsilon^2} \right) \log \left( \frac{kd\kappa}{\varepsilon} \log\left(\frac{U}{L}\right) \right),$$

suffices for

$$\Pr[\sup_{W,P} f'_n(W,P) - f_n(W,P) > \varepsilon] < e^{-\Omega(\varepsilon^2 n)}.$$

$\square$

**Lemma C.19.** *Let $y = \phi(\mu^* + \eta_{\sigma^*})$ where $\mu^*, \sigma^*$ are fixed, and $y_{\mu,\sigma} = \phi(\mu + \eta_\sigma)$. We partition the space $\mathbb{R}$ of $\mu$ s.t. for $\mu, \mu'$ in a cell, either*

$$d_{TV}(p_{\mu,\sigma}(y), p_{\mu',\sigma}(y)) \le \epsilon/2d.$$

*or*

$$d_{TV}(p_{\mu,\sigma}(y), p_{\mu^*,\sigma^*}(y)) \ge 1 - \epsilon \quad and \quad d_{TV}(p_{\mu',\sigma}(y), p_{\mu^*,\sigma^*}(y)) \ge 1 - \epsilon.$$

*Then the number of cells is at most $\mathcal{O}(\frac{d}{\epsilon^{3/2}} \sqrt{\log(\frac{2d}{\epsilon})})$.*

*Proof.* In one dimension

$$d_{TV}(p_{\mu,\sigma}(y), p_{\mu^*,\sigma^*}(y)) = d_{TV}(p_{c\mu,c\sigma}(y), p_{c\mu^*,c\sigma^*}(y))$$

where $c$ is a constant. So, we can assume WLOG that $\sigma^* = 1$. The number of cells in the grid only depends on $\sigma/\sigma^*$.

Now, we show that, regardless of the value of $\mu^*$ we only need to make a grid on a segment of length at most $3\max(\sigma, 1)\sqrt{\log(2d/\epsilon)}$. This is because for any $\mu$ outside the ranges specified below the $d_{TV}(p_{\mu,\sigma}(y), p_{\mu^*,\sigma^*}(y)) \ge 1 - \epsilon$.

- If $\mu^* \le -\sqrt{\log(2d/\epsilon)}$ and for any $\mu$ such that $\mu \ge \sigma\sqrt{\log(2d/\epsilon)}$, the $d_{TV}(p_{\mu^*,\sigma^*}(y), p_{\mu,\sigma}(y)) \ge$ the difference in the probabilities at $0$ which is bigger than $1 - \epsilon$.

- If $0 \ge \mu^* \ge -\sqrt{\log(2d/\epsilon)}$ and for any $\mu$ s.t. $\mu \ge \max(\sigma, 1)\sqrt{\log(2d/\epsilon)}$, the $d_{TV}(p_{\mu^*,\sigma^*}(y), p_{\mu,\sigma}(y))$ is the same as in the linear case and since, $\mu - \mu^* \ge \max(\sigma, 1)\sqrt{\log(2d/\epsilon)}$, the $d_{TV}(p_{\mu^*,\sigma^*}(y), p_{\mu,\sigma}(y)) \ge 1 - \epsilon$.

- If $0 \le \mu^* \le \sqrt{\log(2d/\epsilon)}$, for any $\mu$ s.t. $\mu - \mu^* \ge \max(\sigma, 1)\sqrt{\log(2d/\epsilon)}$, the $d_{TV}(p_{\mu^*,\sigma^*}(y), p_{\mu,\sigma}(y)) \ge 1 - \epsilon$.

- If $\mu^* \ge \sqrt{\log(2d/\epsilon)}$ then for any $\mu$ s.t. $\mu - \mu^* \ge \max(\sigma, 1)\sqrt{\log(2d/\epsilon)}$ using the same argument as above, we have, the $d_{TV}(p_{\mu^*,\sigma^*}(y), p_{\mu,\sigma}(y)) \ge 1 - \epsilon$. Moreover, this is also true for $\mu$ s.t. $-\sigma\sqrt{\log(2d/\epsilon)} \le \mu \le \mu^* - \max(\sigma, 1)\sqrt{\log(2d/\epsilon)}$. Therefore, in this case, we have an additional cell.

In addition to the above, for any $\mu, \mu' \le -\sigma\sqrt{\log(2d/\epsilon)}$, the $d_{TV}(p_{\mu,\sigma}(y), p_{\mu',\sigma}(y)) \le \epsilon/2d$ since both $y_{\mu,\sigma}, y_{\mu',\sigma}$ are only non-zero with probability at most $\epsilon/2d$. Therefore, for all the above cases, we only need to partition a segment of length at most $3\max(\sigma, 1)\sqrt{\log(2d/\epsilon)}$

Moreover, for $\sigma$ sufficiently small we can do better. We only need to partition a space of $\sigma\sqrt{\log(2d/\epsilon)}$. This is primarily because when $\sigma$ sufficiently small, for any $\mu$ in the linear case we have that $d_{TV}(p_{\mu,\sigma}(y), p_{\mu^*,\sigma^*}(y)) \ge 1 - \epsilon$.

It is easy to see that $d_{TV}(p_{\mu,\sigma}(y), p_{0,1}(y)) \geq d_{TV}(p_{0,\sigma}(y), p_{0,1}(y))$. The PDFs of $\mathcal{N}(0, \sigma)$ and $\mathcal{N}(0,1)$ intersect at $x = \pm\sigma\sqrt{\frac{\log(1/\sigma^2)}{1-\sigma^2}}$. To show that $d_{TV}(p_{0,\sigma}(y), p_{0,1}(y)) \geq 1 - \epsilon$, it is now sufficient to show that

$$1 - 2\Phi(-|x|/\sigma) \geq 1 - \epsilon \quad \text{and} \quad 1 - 2\phi(-|x|) \leq \epsilon,$$

where $\Phi(x)$ is the CDF of the standard normal distribution. By using classical bounds on $\Phi(x)$, we have that

$$\Phi(-|x|/\sigma) \leq \frac{\exp^{-x^2/2\sigma^2}}{|x|/\sigma} = \frac{\exp^{-\frac{\log(1/\sigma^2)}{2(1-\sigma^2)}}}{|x|/\sigma}$$

which is $\leq \epsilon/2$ if $\sigma^2 \leq \epsilon/2$. And,

$$\Phi(-|x|) \geq \frac{|x|\exp^{-x^2/2}}{x^2 + 1}$$

which is $\geq (1 - \epsilon)/2$ if

When $\mu^* \leq -\sqrt{\log(2d/\epsilon)}$ the same argument as above shows that if $\mu > \sigma\sqrt{\log(2d/\epsilon)}$ then the $d_{TV}(p_{\mu^*,\sigma^*}(y), p_{\mu,\sigma}(y)) \geq$ the difference in the probabilities at $0$ which is bigger than $1 - \epsilon$. We consider the case where $\mu^* \geq -\sqrt{\log(2d/\epsilon)}$.

Since, when $\sigma$ is small, for any $\mu$ in the linear case the TV distance is large it is sufficient to have $\mu$ large enough so that the intersection of the PDFs are positive and we are in the linear case.

The point of intersection assuming $\sigma^* = 1$ is given by

$$x = \frac{\mu_1 - \mu_2/\sigma^2 \pm \sqrt{(\mu_1 - \mu_2)^2/\sigma^2 + \left(\frac{1}{\sigma^2} - 1\right)\log(\sigma^2)}}{\frac{1}{\sigma^2} - 1}$$

which is positive whenever

$$\mu_2 \geq \sigma\sqrt{\log(1/\sigma^2 + \mu_1^2)} \geq \sigma\sqrt{\log(2d/\epsilon)}.$$

For the rest of the space, we partition $\mu \in \mathbb{R}^k$ s.t. for any $\mu, \mu'$ in a cell,
$$|\mu - \mu'| \leq \sigma\epsilon/2d.$$

This implies that for any $\mu, \mu'$ in a cell, either,
$$d_{TV}(p_{\mu,\sigma}(y), p_{\mu',\sigma}(y)) \leq \epsilon/2d$$

or
$$d_{TV}(p_{\mu,\sigma}(y), p_{\mu^*,\sigma^*}(y)) \geq 1 - \epsilon \text{ and } d_{TV}(p_{\mu',\sigma}(y), p_{\mu^*,\sigma^*}(y)) \geq 1 - \epsilon.$$

Then the number of cells is the max of $\mathcal{O}(d\sqrt{\log(2d/\epsilon)}/\epsilon)$ (when $\sigma$ small) or $\mathcal{O}(d\sqrt{\log(2d/\epsilon)}/\sigma\epsilon) \leq d\sqrt{\log(2d/\epsilon)}/\epsilon^{3/2}$ (when $\sigma$ large). $\qquad\square$

## D  Proof of composition of layers

*Proof.* We can use the triangle inequality to compose our single layer guarantees. Suppose, for layer $j$ and $j + 1$ we have

$$d_{TV}(X^j, \hat{X}^j) \leq \epsilon/2 \quad \text{and}$$
$$d_{TV}(\phi(\hat{W}_{MLE}^j X_j + \eta_j), \phi(W^j X_j + \eta_j)) \leq \epsilon/2$$

then,

$$d_{TV}(\phi(\hat{W}_{MLE}^j \hat{X}_j + \eta_j), \phi(W^j X_j + \eta_j))$$
$$\leq d_{TV}(\phi(\hat{W}_{MLE}^j \hat{X}_j + \eta_j), \phi(\hat{W}_{MLE}^j X_j + \eta_j))$$
$$+ d_{TV}(\phi(\hat{W}_{MLE}^j X_j + \eta_j), \phi(W^j X_j + \eta_j))$$
$$\leq \epsilon$$

where we use the fact that $d_{TV}(f(X), f(Y)) \leq d_{TV}(X, Y)$.

$\qquad\square$

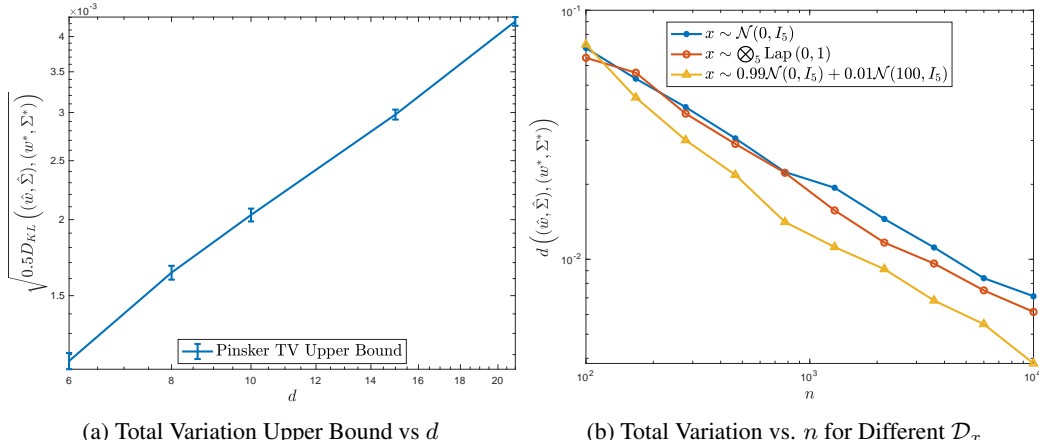

(a) Total Variation Upper Bound vs $d$  (b) Total Variation vs. $n$ for Different $\mathcal{D}_x$

Figure 4: (a) Plots Pinsker's upper bound on the TV distance as $d$ gets large. We set $\Sigma^* = I_d$ and $W^* = \mathbb{1}_{d \times 1}$, thus setting input dimension $k = 1$. $n = 5000$ samples are taken. As we might expect, the upper bound is increasing in $d$. each point is determined by 2000 samples. (b) Plot of TV vs. $n$ for additional distributions of $x$. All three distributions follow roughly the same trend, each point is determined by 2500 samples.

# E   Simulations

## E.1   Additional Simulations

In this section, we provide additional simulations to supplement some of the discussion in Section 5.

## E.2   Simulation Details

### E.2.1   Figure 2

In these experiment, we set $d = 1$, and plot the results for various values of the number of samples $n$ in Figure 2a and various values of the input dimension $k$ in Figure 2b. For each plot, we fix the true $\sigma^* = 1$ and the $w^* = \mathbb{1}_{k \times 1}$. In each case the MLE is solved via gradient descent with backtracking line search, and we check a first order condition $\|\nabla_{w,\sigma} \log p_{w,\sigma}((y \mid x))\|^2 < \delta = 10^{-3}$ as the exit condition. We verify that increasing or decreasing $\delta$ by one order of magnitude makes no difference to the Figure.

The expected total variation distance for the two distributions is calculated as follows. We sample $x$ according to the true distribution (in this case either Laplace or Normal). Then we compute $d_{TV}(p_{\hat{w},\hat{\sigma}}(y \mid x), p_{w,\sigma}(y \mid x))$ via the MATLAB integral function which uses vectorized adaptive quadrature. We repeat this a total of 100 times and take the average to compute our expected total variation. We then repeat the entire process 2000 times, each time optimizing to find an MLE, and then compute its average total variation distance. Lines indicate the average of these experiments, and the error bars, (not easily visible due to their size) indicates one standard error.

### E.2.2   Figure 3

In these experiments we fix $d = 3$ to retain reasonable complexity for computing the TV distance, and take input dimension $k = 1$ with deterministic $x$ in order to compare with [35]. In Figure 3a we fix $\Sigma^* = I_d$ and take let $W^* = b\mathbb{1}_{d \times 1}$, where $b$ will vary across our experiments. We set $n = 10000$. In Figure 3b we set $n = 5000$ and adjust $\Sigma$ such that one diagonal entry is $\kappa^{1/2}$, and the other is $\kappa^{-1/2}$, making the total condition number $\kappa$.

In both of these experiments, we restrict the MLE computation to be over diagonal $\Sigma$ only. This is not because computation of the MLE is too difficult, but rather because computing the TV distance is greatly simplified in this case. The algorithm of Wu et al. is hence modified to use the knowledge that the output must be diagonal. This is simply done, because the procedure of Wu et al. essentially first

estimates the diagonal entries of the matrix as if it were diagonal and then computes the correlations. Removing this second phase allows us to achieve our goal.

Since $x$ is deterministic, we do not need to consider randomness in computing the expected TV distance, though other challenges remain. Since our distribution is degenerate, we must be very careful in computing the TV distance in higher dimensions. Specifically, in the diagonal case, the TV may be written as:

$$d_{TV}\left((\widehat{W}, \widehat{\Sigma}), (W^*, \Sigma^*)\right) = \frac{1}{2} \int_{\mathbb{R}_{\geq 0}^d} \left|p_{\widehat{W}, \widehat{\Sigma}}(y \mid x) - p_{W^*, \Sigma^*}(y \mid x)\right| dy$$

$$= \frac{1}{2} \int_{\mathbb{R}_{\geq 0}^3} \left|\prod_{i=1}^d p_{\widehat{W}, \widehat{\Sigma}}(y_i \mid x) - \prod_{i=1}^d p_{W^*, \Sigma^*}(y_i \mid x)\right| dy,$$

where $y_i$ is the $i^{\text{th}}$ element of $y$. Though at first glace it seems that this is a single high-dimensional integral, the reality is that due to the truncation, the probability mass on the boundary of the non-negative orthant cone $\mathbb{R}_{\geq 0}^3$ has a complex structure that cannot be ignored. Instead we perform a series of integrals of continuous bounded functions, which are much more amenable to Monte-Carlo integration techniques:

$$d_{TV}\left((\widehat{W}, \widehat{\Sigma}), (W^*, \Sigma^*)\right) =$$

$$\frac{1}{2} \sum_{S' \in 2^{[d]}} \int_{\mathbb{R}_{\geq 0}^{|S'|}} \left|\prod_{i \in S'} p_{\widehat{W}, \widehat{\Sigma}}(y_i \mid x) \prod_{i \in (S')^c} \Phi\left(\widehat{\Sigma}_{ii}^{-1/2}\widehat{W}_i\right) - \prod_{i \in S'} p_{W^*, \Sigma^*}(y_i \mid x) \prod_{i \in (S')^c} \Phi\left(\Sigma_{ii}^{-1/2}W_i\right)\right| dy_{S'}.$$

$$(45)$$

Essentially, for each possible support of $y$, $S'$, we integrate over the absolute deviation in those coordinates.

### E.2.3   Figure 4

In Figure 4a, we plot an upper bounds for the TV distance of the MLE as the output dimension $d$ grows. We set the input dimension $k = 1$ with deterministic $x$ and fix the number of samples $n = 5000$. To estimate the KL divergence, we repeatedly sample $y$ according to the true distribution, and compute the empirical average log-likelihood ratio.

In Figure 4b we fix the output dimension $d = 1$ and input dimension $k = 5$, and compute the TV over a range of values of $n$. In addition to $x$ sampled i.i.d. from the Normal and Laplace distributions, we also plot a performance with a Normal mixture, where with probability $0.01$, the normal distribution has mean shifted by $100$. We observe, as our theory suggests, that in all cases, there is only very minor differences in the expected TV distance.

Note that in the case where $x$ is distributed according to a Normal mixture, we observe that the optimization may become very challenging, and in the plot above, we have omitted some of the instances where optimization failed due to lack of smoothness in the objective and numerical imprecision. Omitting these point may lead to a small systematic error in the figure, which may explain why it is lower than the other plots. In practice, for a fixed optimization budget, we may observe meaningful differences in TV for different distributions of $x$, since computing the MLE becomes more challenging for more complex heavy-tailed distributions.

## F   The Likelihood Function

In this section, we discuss the likelihood function, proving log-concavity, as well as discussing computational challenges.

### F.1   One Dimensional Case

In this section, we consider the case where the output dimension $d = 1$, with some $\sigma^*$ and some $W^* \in \mathbb{R}^{1,k}$ and describe how to compute the likelihood function. We defer the proof of the

log-concavity to the follwing section, which covers the more general case with $d \geq 1$ When we re-parameterize as $u = -W/\sigma$, $v = 1/\sigma$, the likelihood function is written as:

$$f_{u,v}(y) = -\frac{1}{2}\sum_{i \in S'}(vy_i - u \cdot x_i)^2 + |S'|\log(v) + \sum_{i \in S}\log\Phi(-u \cdot x_i) \tag{46}$$

where in this case we let $S = \{i \mid y_i = 0\}$, where $y_i$ is the $i^{\text{th}}$ sample in the set $\{y_i, x_i\}_{i=1}^n$. Note this is distinct from how we define $S$ and $S'$ in the multidimensional case, where it corresponds to the zero and non-zero coordinates of a single sample $y_i$. The case of $d > 0$ with uncorrelated $\eta$ follows a similar approach.

**Numerical concerns.** In (46), the term $\log\Phi(-u \cdot x_i)$ presents some numerical concerns when $u \cdot x_i \gg 0$ if we naively compute $\Phi(-u \cdot x_i)$ and then take the logarithm. Instead we compute it from the *mills ratio* $m(x)$ [27], defined to be the ratio of the standard normal pdf and the complementary cdf. The mills ratio is easily computed, with many well-known expansions, see for example, [13]. Then we can write:

$$\log\Phi(-x) = -\log m(x) - \frac{1}{2}\log(2\pi) - \frac{1}{2}x^2, \quad x > 0.$$

Since $m(x)$ changes relatively slowly in $x$ compared to $\Phi(-x)$, this greatly improves numerical stability.

## F.2 Multidimensional Case

In the multi-dimensional case, we will generally use the more standard natural parameters:

$$U := \frac{\Sigma^{-1}}{2},$$
$$v := -\Sigma^{-1}Wx.$$

Note that in the one-dimensional case, we could have also use the natural parameters, but due to the truncation structure, the parameters we used make the computation simpler, in a way that does not apply to the multidimensional case. Also note that here we are considering a fixed $x$ and writing $v$ as a vector. In full generality, we should take $V = -\Sigma^{-1}W$, however, this is a simple extension which we omit here for readability. It turns out that density is log-concave in these natural parameters:

**Lemma F.1.** *The log-likelihood function in Eqn* (8) *is concave in the natural parameter space.*

*Proof.* First, let's write the un-truncated density in terms of these parameters:

$$
\begin{aligned}
f_{W,\Sigma}(y|x) &= \exp\left(-\frac{1}{2}(y - Wx)^T\Sigma^{-1}(y - Wx) - \frac{1}{2}\log|2\pi\Sigma|\right), \tag{47}\\
&= \exp\left(-\frac{1}{2}y^T\Sigma^{-1}y + x^TW^T\Sigma^{-1}y - x^TW^T\Sigma^{-1}Wx - \frac{1}{2}\log|2\pi\Sigma|\right) \tag{48}\\
&= \exp\left(-\frac{1}{2}y^TUy - v^Ty - v^TU^{-1}v - \frac{1}{2}\log((2\pi)^n/|U|)\right) \tag{49}
\end{aligned}
$$

$$\tag{50}$$

Thus, the untrucated conditional density can be written as:

$$f_{U,v}(y) = \exp\left(-\frac{1}{2}y^TUy + y^Tv - A(U, v)\right),$$

where $A(U, v)$ is the cumulant function (note this is distinct from the related cumulant generating function). A well known result is that $A$ is jointly convex in its arguments, $U$ and $v$. Taking logs and using this fact, shows us that $f_{U,v}(y)$ is log-concave in $U, v$.

Our truncated density is simply:

$$f_{U,v}(y|x) = \int_{y_S \leq 0} p_{U,v}(y|x)dy_S,$$

For any log-concave density $f(x)$, integration over a convex subset of the coordinates preserves log-concavity ([9], Example 3.42-3.44). Thus the objective is log-concave. $\qquad\square$

Then the likelihood function at $U, v$ can be rewritten as

$$f_{U,v}(y) = \log \int_{t_S \leq 0, t_{S'} = y_{S'}} \exp\left(-t^T U t - t^T v - \frac{v^T U^{-1} v}{4} + \frac{1}{2}\log|2U|\right),$$

$$= -\frac{v^T U^{-1} v}{4} + \frac{1}{2}\log|2U| + \log \int_{t_S \leq 0, t_{S'} = y_{S'}} \exp\left(-t^T U t - t^T v\right),$$

Separating the terms corresponding to $S$ and $S'$, we get

$$f_{U,v}(y) = -\frac{v^T U^{-1} v}{4} + \frac{1}{2}\log|2U| - y_{S'}^T U_{S'} y_{S'} - y_{S'}^T v_{S'} + \log \int_{t_S \leq 0} \exp\left(-t_S^T U_S t_S - 2y_{S'}^T U_{S'S} t_S - v_S^T t_S\right).$$

The last term resembles the $\log \Phi$ term that appears in the univariate case. This resemblance can be made more clear as follows. Let $r_S = U_{SS'} y_S' + \frac{1}{2}v_S$ and $M^T M = U_S$.

$$= \log \int_{t_S \leq 0} \exp\left(-\left(t_S^T M^T M t_S + 2t_S^T r_S\right)\right)$$

$$= \log \int_{t_S \leq 0} \exp\left(-\left((Mt_S)^T M t_S + 2t_S^T r_S + r_S^T M^{-1} M^{-T} r_S - r_S^T M^{-1} M^{-T} r_S\right)\right)$$

$$= \log \int_{t_S \leq 0} \exp\left(-\|M t_S + M^{-T} r_S\|^2 + r_S^T M^{-1} M^{-T} r_S\right)$$

$$= r_S^T M^{-1} M^{-T} r_S + \log \int_{t_S \leq 0} \exp\left(-\|t_S + U_S^{-1} r_S\|_{U_S}^2\right)$$

$$= r_S^T U_s^{-1} r_S + \log \int_{t_S \leq 0} \frac{(2\pi)^{d/2}|U_S^{-1}/2|^{1/2}}{(2\pi)^{d/2}|U_S^{-1}/2|^{1/2}} \exp\left(-\|t_S + U_S^{-1} r_S\|_{U_S}^2\right)$$

$$= r_S^T U_s^{-1} r_S - \frac{1}{2}\log|2U_S| + \log \int_{t_S \leq 0} \frac{1}{(2\pi)^{d/2}|U_S^{-1}/2|^{1/2}} \exp\left(-\frac{1}{2}\|t_S + U_S^{-1} r_S\|_{2U_S}^2\right) + c$$

$$= r_S^T U_s^{-1} r_S - \frac{1}{2}\log|2U_S| + \log \Phi\left(0; \mu = -U_S^{-1} r_S, \Sigma = \frac{1}{2}U_S^{-1}\right) + c$$

Putting this together, $f_{U,v}$ can be written as:

$$f_{U,v}(y) = -\frac{v^T U^{-1} v}{4} + \frac{1}{2}\log|U| - y_{S'}^T U_{S'} y_{S'} - y_{S'}^T v_{S'} + r_S^T U_S^{-1} r_S - \frac{1}{2}\log|U_S| + |S'|\frac{\log(2)}{2}$$

$$+ \log \Phi\left(0; \mu = -U_S^{-1} r_S, \Sigma = \frac{1}{2}U_S^{-1}\right) + c \quad (51)$$

Thus, it appears that evaluating the likelihood for even a single sample involves the high-dimensional integral that is the rectangular cdf in equation (51).

### F.3 Computing Gradients

#### F.3.1 One Dimensional Case

In the one-dimensional case, the gradient with respect to $u$ is easily computed as:

$$\nabla_u f_{u,v}(y) = \sum_{i \in S'}(vy_i - u \cdot x_i)x_i - \sum_{i \in S}\frac{1}{m(u \cdot x_i)}x_i,$$

where we have previously defined $m(x)$ as the mills ratio. Furthermore, we have:

$$\nabla_v f_{u,v}(y) = |S'|\frac{1}{v} - \sum_{i \in S'} y_i(vy_i - u \cdot x_i)$$

### F.3.2 Multidimensional Case

First we consider the non-integral terms in the likelihood. Differentiating each term wrt $U$, we get

$$-\nabla_U \frac{v^T U^{-1} v}{4} = \frac{1}{4}(U^{-1} v v^T U^{-1}),$$

$$\nabla_U \frac{1}{2} \log|2U| = \frac{1}{2} U^{-1},$$

$$-\nabla_U y_{S'}^T U_{S'} y_{S'} = \begin{pmatrix} 0 & 0 \\ 0 & -y_{S'} y_{S'}^T \end{pmatrix}$$

Differentiating each term wrt $v$, we get

$$-\nabla_v \frac{v^T U^{-1} v}{4} = -\frac{1}{2} U^{-1} v,$$

$$-\nabla_v y_{S'}^T v_{S'} = \begin{pmatrix} 0 \\ -y_{S'} \end{pmatrix}$$

Now consider the integral term. Differentiating wrt $U$, we get

$$\nabla_U \log \int_{t_S \leq 0} \exp\left(-t_S^T U_S t_S - 2y_{S'}^T U_{S'S} t_S - v_S^T t_S\right)$$

$$= \frac{\int_{t_S \leq 0} \begin{pmatrix} -t_S t_S^T & -t_S y_{S'}^T \\ -y_{S'} t_S^T & 0 \end{pmatrix} \exp\left(-t_S^T U_S t_S - 2y_{S'}^T U_{S'S} t_S - v_S^T t_S\right)}{\int_{t_S \leq 0} \exp\left(-t_S^T U_S t_S - 2y_{S'}^T U_{S'S} t_S - v_S^T t_S\right)}$$

Let $M$ be a matrix such that

$$M^T M = U_S$$

Then via completion of squares in the exponential term, we get

$$\nabla_U \log \int_{t_S \leq 0} \exp\left(-t_S^T U_S t_S - 2y_{S'}^T U_{S'S} t_S - v_S^T t_S\right) \tag{52}$$

$$= \frac{\int_{t_S \leq 0} \begin{pmatrix} -t_S t_S^T & -t_S y_{S'}^T \\ -y_{S'} t_S^T & 0 \end{pmatrix} \exp\left(-\|M t_S + (M^{-1})^T \left(U_{SS'} y_{S'} + \frac{v_S}{2}\right)\|^2\right)}{\int_{t_S \leq 0} \exp\left(-\|M t_S + (M^{-1})^T \left(U_{SS'} y_{S'} + \frac{v_S}{2}\right)\|^2\right)} \tag{53}$$

Notice that this density is Gaussian, with mean and covariance:

$$\mathcal{N}\left(-M^{-1}(M^{-1})^T \left(U_{SS'} y_{S'} + \frac{v_S}{2}\right); \frac{U_S^{-1}}{2}\right).$$

And hence, the gradient can be estimated as

$$\nabla_U \log \int_{t_S \leq 0} \exp\left(-t_S^T U_S t_S - 2y_{S'}^T U_{S'S} t_S - v_S^T t_S\right) = \mathbb{E}_{t_S}\left[\begin{pmatrix} -t_S t_S^T & -t_S y_{S'}^T \\ -y_{S'} t_S^T & 0 \end{pmatrix}\right]$$

where $t_S$ is the truncation of

$$z \sim \mathcal{N}\left(-M^{-1}(M^{-1})^T \left(U_{SS'} y_{S'} + \frac{v_S}{2}\right); \frac{U_S^{-1}}{2}\right).$$

to the negative quadrant. A similar calculation gives the gradient for $v$ as

$$\nabla_v \log \int_{t_S \leq 0} \exp\left(-t_S^T U_S t_S - 2y_{S'}^T U_{S'S} t_S - v_S^T t_S\right) = \mathbb{E}_{t_S}\left[\begin{pmatrix} -t_S \\ 0 \end{pmatrix}\right]$$

