# A  Proofs of Linear Case

Throughout the appendix, for ease of notation, we overload the definition of the function $d_{TV}(\cdot, \cdot)$. When inputs are random variables, it represent the TV distance between the distributions of those random variables.

**Lemma 4.2.** *Let $\{(x_i, y_i)\}_{i=1}^n$ be i.i.d. random variables such that $y_i = x_i \cdot w^* + \mathcal{N}(0, 1)$. Then, for $n \geq \frac{k}{2}$, with probability $1 - e^{-\Omega(n)}$, the MLE $\widehat{w}$ satisfies*

$$\widetilde{d}(\widehat{w}, w^*) \leq \sqrt{\frac{k}{2n}}.$$

The proof of this lemma requires Lemma A.1, which characterizes the distribution of the residual error of the MLE.

**Lemma A.1.** *Given $y \in \mathbb{R}^n, X \in \mathbb{R}^{n \times k}$ satisfying $y = Xw^* + \eta$, where $\eta \sim \mathcal{N}(0, \sigma^2 I_n)$, the least square solution $\widehat{w}$ satisfies*

$$Xw^* - X\widehat{w} \sim \mathcal{N}(0, \sigma^2 X(X^T X)^{-1} X^T) \Rightarrow \mathbb{E}[\|X\widehat{w} - Xw^*\|^2] = \sigma^2 k.$$

*Proof.* The least squares solution is given by

$$
\begin{aligned}
\widehat{w} &= (X^T X)^{-1} X^T y, \\
&= (X^T X)^{-1} X^T (Xw^* + \eta), \\
&= w^* + (X^T X)^{-1} X^T \eta.
\end{aligned}
$$

Multiplying on the left by $X$, we have

$$X\widehat{w} = Xw^* + X(X^T X)^{-1} X^T \eta.$$

Since $\eta$ is i.i.d. Gaussian with variance $\sigma^2$, we have,

$$
\begin{aligned}
X(X^T X)^{-1} X^T \eta &\sim \mathcal{N}(0, \sigma^2 X(X^T X)^{-1} X^T X (X^T X)^{-1} X^T) \\
&\sim \mathcal{N}(0, \sigma^2 X(X^T X)^{-1} X^T)
\end{aligned}
$$

This implies

$$
\begin{aligned}
\mathbb{E}[\|X\widehat{w} - Xw^*\|^2] &= \sigma^2 \text{Tr}[X(X^T X)^{-1} X^T], \\
&= \sigma^2 \text{Tr}[(X^T X)^{-1} X^T X], \\

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

/2} \left|U_S^{-1}/2\right|^{1/2}}{(2\pi)^{d/2} \left|U_S^{-1}/2\right|^{1/2}} \exp\left(-\left\|t_S + U_S^{-1} r_S\right\|_{U_S}^2\right)$$

$$= r_S^T U_s^{-1} r_S - \frac{1}{2}\log|2U_S| + \log \int_{t_S \leq 0} \frac{1}{(2\pi)^{d/2} \left|U_S^{-1}/2\right|^{1/2}} \exp\left(-\frac{1}{2}\left\|t_S + U_S^{-1} r_S\right\|_{2U_S}^2\right) + c$$

$$= r_S^T U_s^{-1} r_S - \frac{1}{2}\log|2U_S| + \log \Phi\left(0; \mu = -U_S^{-1} r_S, \Sigma = \frac{1}{2} U_S^{-1}\right) + c$$

Putting this together, $f_{U,v}$ can be written as:

$$f_{U,v}(y) = -\frac{v^T U^{-1} v}{4} + \frac{1}{2}\log|U| - y_{S'}^T U_{S'} y_{S'} - y_{S'}^T v_{S'} + r_S^T U_S^{-1} r_S - \frac{1}{2}\log|U_S| + |S'|\frac{\log(2)}{2}$$

$$+ \log \Phi\left(0; \mu = -U_S^{-1} r_S, \Sigma = \frac{1}{2} U_S^{-1}\right) + c \quad (51)$$

Thus, it appears that evaluating the likelihood for even a single sample involves the high-dimensional integral that is the rectangular cdf in equation (51).

### F.3 Computing Gradients

#### F.3.1 One Dimensional Case

In the one-dimensional case, the gradient with respect to $u$ is easily computed as:

$$\nabla_u f_{u,v}(y) = \sum_{i \in S'} (vy_i - u \cdot x_i) x_i - \sum_{i \in S} \frac{1}{m(u \cdot x_i)} x_i,$$

where we have previously defined $m(x)$ as the mills ratio. Furthermore, we have:

$$\nabla_v f_{u,v}(y) = |S'|\frac{1}{v} - \sum_{i \in S'} y_i (vy_i - u \cdot x_i)$$

### F.3.2 Multidimensional Case

First we consider the non-integral terms in the likelihood. Differentiating each term wrt $U$, we get

$$-\nabla_U \frac{v^T U^{-1} v}{4} = \frac{1}{4}(U^{-1} v v^T U^{-1}),$$

$$\nabla_U \frac{1}{2} \log|2U| = \frac{1}{2} U^{-1},$$

$$-\nabla_U y_{S'}^T U_{S'} y_{S'} = \begin{pmatrix} 0 & 0 \\ 0 & -y_{S'} y_{S'}^T \end{pmatrix}$$

Differentiating each term wrt $v$, we get

$$-\nabla_v \frac{v^T U^{-1} v}{4} = -\frac{1}{2} U^{-1} v,$$

$$-\nabla_v y_{S'}^T v_{S'} = \begin{pmatrix} 0 \\ -y_{S'} \end{pmatrix}$$

Now consider the integral term. Differentiating wrt $U$, we get

$$\nabla_U \log \int_{t_S \leq 0} \exp\left(-t_S^T U_S t_S - 2y_{S'}^T U_{S'S} t_S - v_S^T t_S\right)$$

$$= \frac{\int_{t_S \leq 0} \begin{pmatrix} -t_S t_S^T & -t_S y_{S'}^T \\ -y_{S'} t_S^T & 0 \end{pmatrix} \exp\left(-t_S^T U_S t_S - 2y_{S'}^T U_{S'S} t_S - v_S^T t_S\right)}{\int_{t_S \leq 0} \exp\left(-t_S^T U_S t_S - 2y_{S'}^T U_{S'S} t_S - v_S^T t_S\right)}$$

Let $M$ be a matrix such that

$$M^T M = U_S$$

Then via completion of squares in the exponential term, we get

$$\nabla_U \log \int_{t_S \leq 0} \exp\left(-t_S^T U_S t_S - 2y_{S'}^T U_{S'S} t_S - v_S^T t_S\right) \tag{52}$$

$$= \frac{\int_{t_S \leq 0} \begin{pmatrix} -t_S t_S^T & -t_S y_{S'}^T \\ -y_{S'} t_S^T & 0 \end{pmatrix} \exp\left(-\|M t_S + (M^{-1})^T \left(U_{SS'} y_{S'} + \frac{v_S}{2}\right)\|^2\right)}{\int_{t_S \leq 0} \exp\left(-\|M t_S + (M^{-1})^T \left(U_{SS'} y_{S'} + \frac{v_S}{2}\right)\|^2\right)} \tag{53}$$

Notice that this density is Gaussian, with mean and covariance:

$$\mathcal{N}\left(-M^{-1}(M^{-1})^T \left(U_{SS'} y_{S'} + \frac{v_S}{2}\right); \frac{U_S^{-1}}{2}\right).$$

And hence, the gradient can be estimated as

$$\nabla_U \log \int_{t_S \leq 0} \exp\left(-t_S^T U_S t_S - 2y_{S'}^T U_{S'S} t_S - v_S^T t_S\right) = \mathbb{E}_{t_S}\left[\begin{pmatrix} -t_S t_S^T & -t_S y_{S'}^T \\ -y_{S'} t_S^T & 0 \end{pmatrix}\right]$$

where $t_S$ is the truncation of

$$z \sim \mathcal{N}\left(-M^{-1}(M^{-1})^T \left(U_{SS'} y_{S'} + \frac{v_S}{2}\right); \frac{U_S^{-1}}{2}\right).$$

to the negative quadrant. A similar calculation gives the gradient for $v$ as

$$\nabla_v \log \int_{t_S \leq 0} \exp\left(-t_S^T U_S t_S - 2y_{S'}^T U_{S'S} t_S - v_S^T t_S\right) = \mathbb{E}_{t_S}\left[\begin{pmatrix} -t_S \\ 0 \end{pmatrix}\right]$$