# OpenReview forum: "Learning a 1-layer conditional generative model in total variation"
_NeurIPS.cc/2023/Conference — NeurIPS 2023 poster_

### Official Review · Reviewer_NYne · 2023-07-05

**Soundness:** 3 good
**Presentation:** 3 good
**Contribution:** 3 good
**Rating:** 7
**Confidence:** 3

**Summary:**

The paper investigates the sample complexity of learning conditional generative models without assumptions on the input distribution. It applies the Maximum Likelihood Estimator (MLE) to linear regression and 1-layer networks with ReLU activation. The results show that the MLE achieves small total variation error with sample complexities of $O(k/\epsilon^2 log(1/\epsilon))$ for linear regression and O((kd + d^2) / \epsilon^2 log(kd\kappa/\epsilon)) for 1-layer networks. The paper also discusses the extension to multilayer networks,  given access to the internal activations. The results suggest that MLE is a promising approach for learning feed-forward generative models from limited samples, though the authors did mention that the computational aspects of the optimization problem are not thoroughly analyzed in the paper.


**Strengths:**

This paper provides a solid theoretical foundation for understanding the sample complexity of learning multi-layer ReLU networks using MLE method. The derived bounds do not make assumptions on the distribution of X or the condition number of W, and achieve a sample complexity polynomial in the system parameters. The developed algorithm and theories show considerable improvement over those in the existing literature. Though there are limitations that are also noted by the authors, I think the paper is innovative and provides interesting insights into efficient learning of conditional generative models.
Moreover, the paper is well-written and presents its concepts and results in a clear and concise manner.

**Weaknesses:**

The weakness and limitations of the paper are well noted and discussed by the authors.

1. To extend the theory to multilayer neural networks, it requires access to intermediate activations, which is impractical
2. It assumes that the learner has an understanding of the model architecture, which might not be the case in practice
3. It might be challenging to perform MLE on some of the models being considered, such as neural networks, and the computational aspects of the optimization problem are not thoroughly examined in this paper.



**Questions:**

What are the practical implications of the sample complexity bounds derived in the paper? How do these bounds impact the feasibility and scalability of learning generative models in real-world applications?

I understand that the paper is limited by space restrictions, but can you provide some insights into how to go about addressing the requirement for accessing intermediate activations in order to apply the proposed algorithm to solving deep neural nets? Is it going to be a deal breaker for the practical applicability of the proposed method?

**Limitations:**

No potential negative societal impact.

---

> ### Author Rebuttal · Authors · 2023-08-09
>
> Thank you for your positive feedback, we are delighted that you find
> our results to be a solid theoretical foundation and a considerable
> improvement over existing literature.
>
> Please find below our response to your questions:
>
> **Question 1:**
> "What are the practical implications of the sample
> complexity bounds derived in the paper? How do these bounds impact the
> feasibility and scalability of learning generative models in
> real-world applications?"
>
> An important part (perhaps the most important part) about our complexity bounds is that they do not depend on the input distribution $x$. This means that our result can be composed for learning multi-layer networks. but also, this means that learning a one layer ReLu network in TV, which is in itself a valid and potentially useful statistical framework is possible with a sample complexity independent of the input distribution $x$.
>
> In terms of learning actual generative models, this work suggests that learning models in TV is possible even when learning model parameters is not. This might help to explain recent work (https://crfm.stanford.edu/2023/03/13/alpaca.html, https://arxiv.org/pdf/2305.11206.pdf) where researchers are able to essentially clone the behaviour of other language models with a very small fraction of samples when fine-tuning another language model. This may be related to the *superficial alignment hypothesis*, which says that these models only require a superficial modification to align their behaviours. It is clear in this case that the weights are not being learned.
>
> **Question 2:**
> "I understand that the paper is limited by space
> restrictions, but can you provide some insights into how to go about
> addressing the requirement for accessing intermediate activations in
> order to apply the proposed algorithm to solving deep neural nets? Is
> it going to be a deal breaker for the practical applicability of the
> proposed method?"
>
> This is a very interesting question and one we are actively
> considering. We think that the approach in Allen-Zhu and Li [1] can
> actually be used in our paper as well. Allen-Zhu and Li assume that
> the activations are sufficiently sparse across layers, which means they
> can use results from sparse-coding to recover these activations given
> only images at these layers.
>
> The main difficulty in directly combining the results in [1] into our
> algorithm as a subroutine is that they are only able to
> *approximately* recover the activations -- this would mean there is a
> distributional mismatch between the true activations and the estimated
> activations. If this distribution mismatch is in TV distance, then the
> results in [1] can be straightforwardly incorporated into ours, where
> the error in the activations will be an additive error in our results.
> Unfortunately, the error in [1] is wrt the $\ell_2$-norms of the
> activations -- this means that the true and estimated activations
> would perhaps only be close in Wasserstein distance. Accommodating
> this distribution mismatch is an open problem we are exploring, and
> one that would make our method more practical.
>
> References:
>
> [1] Allen-Zhu, Zeyuan, and Yuanzhi Li. "Forward Super-Resolution: How
> Can GANs Learn Hierarchical Generative Models for Real-World
> Distributions." The Eleventh International Conference on Learning
> Representations. 2022.

---

> > ### Comment · Reviewer_NYne · 2023-08-16
> >
> > Thank you very much for the response

---

### Official Review · Reviewer_9ZPm · 2023-07-07

**Soundness:** 3 good
**Presentation:** 3 good
**Contribution:** 3 good
**Rating:** 7
**Confidence:** 3

**Summary:**

This paper studies conditional distribution learning: given iid samples (x,y) where x ~ D and y ~ p(y|w*,x), the goal is to find some estimate w such that the distributions p(y|w*,x) and p(y|w,x) are close in expectation over x ~ D, or equivalently the learned distribution of (x,y) (where x ~ D) is close to the true distribution.

Specifically, this work studies the 1-layer conditional generative model y = max(W* x + eta, 0) where W* is a matrix and eta is a multivariate mean-zero Gaussian with some unknown covariance Sigma*. The main result shows that for an arbitrary covariate distribution D, and arbitrary W*, so long as Sigma* has covariance at most kappa, the sample complexity of MLE (needed to learn up to total variation distance epsilon) is polynomial in the dimension parameters, log(kappa), and 1/epsilon.

An straightforward extension to multi-layer networks is also given (under the assumption that intermediate activations are known).

**Strengths:**

- This work introduces a (to my knowledge) novel perspective to the problem of learning generative models: distribution learning rather than parameter estimation. This more accurately addresses the problem that actually matters in practice for generative models, and avoids needing to worry about identifiability issues.
- Due to the new goal, no distributional assumptions are needed (aside from the tame bound on condition number of the noise).
- The paper is well-written, with a toy example of linear regression given for intuition. I did not have time to check the proofs, but the approach and proof sketch seem reasonable.

**Weaknesses:**

- As the authors acknowledge, the paper only addresses the statistical question. It's claimed that the MLE is concave; however, I could not find a proof of this fact in the paper. Moreover, looking at the log-likelihood function (8) I do not see why it should be concave.

**Questions:**

See above

**Limitations:**

Yes

---

> ### Author Rebuttal · Authors · 2023-08-08
>
> Thank you for your positive feedback, we are delighted that you find
> our problem novel and the results significant.
>
> Please find below our response to your concern regarding our claim
> that the MLE is concave.
>
> **Question 1:**
> "It's claimed that the MLE is concave, however, I could
> not find a proof"
>
> We apologize, due to an editing error we neglected to provide a proof of this in the
> supplementary. The proof actually follows as a straightforward result
> of the following fact. The objective in Eqn (8) is an integral  of
> two log-concave functions: (i) the indicator function over the negative
> orthant, and (ii) the Gaussian likelihood. As (i) and (ii) are
> log-concave functions, their integral in Eqn (8) is also log-concave.
>
> Log-concavity being closed under integration is a non-trivial fact -- for a concrete reference, please refer to Page 106 in Boyd and
> Vandenberghe (https://web.stanford.edu/~boyd/cvxbook/bv_cvxbook.pdf),
> in particular, Examples 3.42, 3.43, and 3.44.

---

> > ### Comment · Reviewer_9ZPm · 2023-08-14
> > **Confused**
> >
> > I'm confused: isn't $\Sigma$ also unknown? I get that the Gaussian likelihood is log-concave in $W$. But is it log-concave jointly in $W$ and $\Sigma$?

---

> > > ### Author Response · Authors · 2023-08-15
> > > **Follow-up**
> > >
> > > Thank you for following up! You are correct to suggest that the objective is not concave in the standard mean-covariance parameterization. There is, however, a simple, easily invertible transformation of these parameters under which the objective is concave. Let $U = \Sigma^{-1}$ and $v = \Sigma^{-1}Wx$ (the "natural" parameter space we discuss in Appendix E). To simplify things slightly, as we did in Appendix E.3.2, we have made the conditioning on $x$ implicit in our parameters.
> > > The un-truncated density is a multivariate normal, and is thus written as an  exponential family in this natural parameter space*:
> > > \begin{equation*}
> > >           p_{U,v} \left( y | x \right) = \exp \left(  -\frac{1}{2}y^TUy + y^Tv - A(U,v)\right),
> > > \end{equation*}
> > > where $A(U,v)$ is the cumulant function (note this is distinct from the related cumulant generating function).  A well known result [1] is that $A$ is jointly convex in $U$ and $v$. Taking logs and using this fact, shows us that $p_{U,v} \left( y | x \right)$ is log-concave in $U,v$.
> > >
> > > Our truncated density is simply:
> > >
> > > \begin{equation*}
> > >           f_{U,v} \left( y | x \right) = \int_{y_S \leq 0} p_{U,v} \left( y | x \right)dy_S,
> > > \end{equation*}
> > >
> > > As we mentioned in our original response, for any log-concave density $f(x)$, integration over a convex subset of the coordinates preserves log-concavity.  Thus the objective is log-concave.
> > >
> > > We hope this clears up your confusion.
> > >
> > > *A few more steps going between the standard and normal parameters:
> > > \begin{eqnarray}
> > >     p \left( y | x \right) &=& \exp\left(-\frac{1}{2}\left( y - Wx \right)^T \Sigma^{-1}(y - Wx) - \frac{1}{2}\log\mid 2\pi \Sigma \mid \right),\\\\
> > >     &=& \exp\left(-\frac{1}{2} y^T\Sigma^{-1}y + x^TW^T\Sigma^{-1}y - \frac{1}{2} x^TW^T\Sigma^{-1}Wx - \frac{1}{2}\log\mid 2\pi \Sigma \mid \right) \\\\
> > >     &=& \exp\left(-\frac{1}{2} y^TUy + v^Ty - \frac{1}{2} v^T U^{-1}v - \frac{1}{2} \log\left({\left(2 \pi\right)^n} / \mid U \mid \right)\right) \\\\
> > > \end{eqnarray}
> > >
> > > By definition [1] the part of the density in the exponential that does not depend on $y$ is the cumulant function.
> > >
> > > **References:**
> > >
> > > [1] Jordan, M. Stat 260 *The Exponential Family: Basics* (https://people.eecs.berkeley.edu/~jordan/courses/260-spring10/other-readings/chapter8.pdf)

---

> > > > ### Comment · Reviewer_9ZPm · 2023-08-15
> > > > **Thanks**
> > > >
> > > > Yes, that makes sense. Thanks for the detailed response! I maintain my rating.

---

### Official Review · Reviewer_zitB · 2023-07-07

**Soundness:** 3 good
**Presentation:** 3 good
**Contribution:** 1 poor
**Rating:** 3
**Confidence:** 3

**Summary:**

In this paper, the authors consider the problem of linear regression and ReLU applied to linear regression. The goal is to recover the weight vector such that the resulting distributions are close as opposed to recovering the weight vector under certain norms such as $\ell_2$ which has been thoroughly studied. For the linear regression, the authors show that the MLE estimator learns the distribution in total variation distance. For the ReLU regression, the authors show again the MLE estimator works when the covariance matrix of the noise is well-conditioned. The resulting algorithm is sample-efficient but not time-efficient.

**Strengths:**

The problem considered is fundamental and it has important connections to learning practically important problems.

**Weaknesses:**

I think the scientific novelty and contribution of this paper is very limited. The proposed results were well-known in the literature in my opinion. As an example, the distributional learning guarantee of linear regression in TV distance was analyzed in [arXiv:2107.10450]. Similarly, one-layer ReLU networks have been analyzed in the works of [Diakonikolas et al, Klivans et al, and Arora et al]. Moreover, the condition number assumption and the high running time of the proposed algorithms are quite restrictive in my opinion.

**Questions:**

None.

**Limitations:**

None,

---

> ### Author Rebuttal · Authors · 2023-08-09
>
> We are disappointed that you've found no positives in our paper. After
> reading the other reviews and this rebuttal, please can you let us
> know if you have any questions that can help us improve our paper?
>
> Please find below our responses to your concerns.
>
> **Weakness 1:** "The proposed results were well-known in the literature in
> my opinion. As an example, the distributional learning guarantee of
> linear regression in TV distance was analyzed in [arXiv:2107.10450]."
>
> 2107.10450 considers a Gaussian Bayesian network, where *all* the
> variables are Gaussian, and they can find parameters that are close to
> the true distribution over these Gaussian random variables.
>
> In our problem, the distribution of $x$ is not Gaussian, and should be
> thought of as a distribution over Word2Vec embeddings. The conditional
> distribution of $y|x$ is assumed to be Gaussian, and the final sample
> complexity is independent of any parameter other than the dimension of
> $x$.
>
> **Weakness 2:** "Similarly, one-layer ReLU networks have been analyzed in
> the works of [Diakonikolas et al, Klivans et al, and Arora et al]."
>
> These are some of the most prolific names in learning theory, and
> without pointing to specific papers, we have no concrete way of
> responding to your criticism.
>
> Nonetheless, in an attempt to guess your concern, the simplest response is that none of
> these results can been extended to multi-layer networks, and our
> assumptions and analysis allow us to consider multi-layer networks.
> As we make no assumptions on the distribution of $x$, this allows us to
> compose our one-layer guarantee recursively over multiple layers.
>
> The assumption that the conditional distribution per layer is a truncated Gaussian is actually used in practice, for example in StyleGAN [1], where the ``style'' vectors are random Gaussian noise variables with learned mean and covariance.
>
> References:
> [1] Karras, Tero, Samuli Laine, and Timo Aila. "A style-based generator architecture for generative adversarial networks." Proceedings of the IEEE/CVF conference on computer vision and pattern recognition. 2019.
>
> **Weakness 3**: "the condition number assumption and the high running time of the proposed algorithms are quite restrictive in my opinion"
>
> Our sample complexity has a logarithmic dependence on the condition number, which allows for an exponentially large condition number with only a polynomial increase in the sample complexity. We agree that the running time is a concern, but as the optimization problem involves maximizing a concave function over a convex set, we believe a better designed optimization algorithm should be able to achieve a rigorous guarantee. The goal of this work is to show that MLE can be used for *distributional learning* of generative models with a polynomial sample complexity.

---

> > ### Comment · Reviewer_zitB · 2023-08-18
> > **response**
> >
> > I have read the rebuttal of the authors. Unfortunately, I am not convinced with their response regarding my points raised. Indeed I am talking about a large body of works over the years by the three co-authors I have mentioned for learning neural networks under different assumptions including Gaussianity.
> >
> > I also believe the conditional number assumption and the exponential running time is quite restrictive.

---

> > > ### Author Response · Authors · 2023-08-20
> > >
> > > This is absurd. Do you have any *specific* papers that you think get comparable guarantees, namely, learning in TV under no distributional assumption on x?

---

> > > > ### Comment · Reviewer_zitB · 2023-08-21
> > > > **response**
> > > >
> > > > No, I don't have any specific reference currently. But I will maintain that the results lack novelty.

---

### Official Review · Reviewer_9DZK · 2023-07-07

**Soundness:** 3 good
**Presentation:** 3 good
**Contribution:** 2 fair
**Rating:** 6
**Confidence:** 2

**Summary:**

The article provides complexity bounds for learning the conditional distribution y|x. One of the main novelties claimed by the authors is that the control of the TD distance between the estimated distribution and the ground truth is more meaningful. Thus, they are able to provide bounds independently of the distribution of label x. The article is well written and the flow is quite pleasant.
See my comments below for more critical details

**Strengths:**

The document is clearly written and the main arguments are transparent and easy to follow (even if they remain fairly technical). The authors did a great job explaining the intuition before getting into formal details.

The distribution free result (wrt to label x) is quite remarkable.

**Weaknesses:**

My main concern is that the results presented might be difficult to compare to classical results on the subject. Below I describe some of my missunderstanding.

**Questions:**



- Usually x is the features and y the labels. The notation might be misleading.
As a matter of fact the product $x \cdot w^*$ is even weird when the label does not belong to a vectorial space.
- On the same flavor, it is difficult to compare with classical results, since the assumptions are usually in the feature space ie distribution assumption on the objet $x$ and the label is generated conditional on $x$ and some noise. It is unclear how the results fit on the same ground.
- The lines 32-35 need explicit references and maybe explicit classical bounds to discuss.

- The complexity presented is not explicitly compared with (the criticized) classical ones. Hence it makes the contributions harder to appreciates

- Theorem 4.1 (and probably) others must states the full assumptions eg $n \geq k/2$ which usually does not hold in high dimensional regime where the number of features is way larger than the sample size.

- As a main proof technique, the authors rely on the relation between TV distance and prediction error in the equation below line 159.
Mainly $d(\hat w, w^*) \approx |x\cdot \hat w - x \cdot w^*|$. And then we directly fall into the OLS analysis as in https://www.di.ens.fr/%7Efbach/ltfp_book.pdf Chapter 3. Can the authors comments on the main novelties after this. Usually the prediction error does not require much restriction on the design matrix (compared to estimation error control).

- In Lemma 4.3, can the authors specify what the probability Pr is?
Also, since the (empirical) distance is Lipschitz, can't we deduce this lemma directly from Talagrand inequality?

- In which practical settings do we know the ground-truth conditioning number?
- Also in Theorem 4.5, $\hat W$ and $\hat \Sigma$ might be non unique. Can the authors comment on that?

Some failure of MLE are quite known, for example in Chapter 24.1.3, https://www.cs.huji.ac.il/~shais/UnderstandingMachineLearning/understanding-machine-learning-theory-algorithms.pdf an explicit overfitting example is shown. Can the authors discuss how their proposition escape such a situation?

---

> ### Author Rebuttal · Authors · 2023-08-09
>
> Thank you for your detailed analysis of our paper. We will include
> your suggestions in future versions.
>
> **Question 1:**"Usually $x$ denotes features and $y$ the labels... $x
> \cdot w^*$ is even weird when the label does not belong to a vector
> space."
>
> We apologize for the confusion -- we used labels in the sense of class
> labels that would be passed to a text-conditional generative model. In
> the context of linear regression, $x$ would be feature vectors (such
> as Word2Vec embeddings) and $y$ would be the data generated. We will
> change this to reduce confusion.
>
> **Question 2:**"On the same flavor, it is difficult to compare with
> classical results, since the assumptions are usually in the feature
> space ie distribution assumption on the objet and the label is
> generated conditional on and some noise. It is unclear how the results
> fit on the same ground."
>
> While it would be nice for comparison to use the same assumptions as classical works as you suggest, the nature of this work is such that this is not possible.
>
> A central aspect of this work is to examine this learning framework from a perspective more relevant to modern problems --- specifically in learning generative models. In doing so, we consider new assumptions (1) no distributional assumptions over $x$ and new goals (2) learning distributions in TV.
>
> (1) The reason we assume an unknown arbitrary distribution over $x$ is
> that it allows us to compose our guarantee over multiple layers. For
> example, if the generative model maps Word2Vec embeddings to images,
> then at the first layer, the distribution of $x$ would be those of
> Word2Vec embeddings. Then for the second layer, we can treat $x$ as
> the output of the first layer. This allows us to compose our guarantee
> over multiple layers, and hence allows for an expressive distribution
> over the data.
>
> (2) Learning in TV is useful because in problems concerning generative models, we are generally concerned with the output of the model, rather than learning the parameters exactly. In our simulation Section 5, Fig. 3 (a), we provide a particular example where the classical objective fails, while our proposed objective succeeds.
>
> **Q3:**"lines 32-35 need explicit references and classical bounds".
>
> We will add this, thank you for the suggestion.
>
> **Q4:**"The complexity presented is not explicitly compared with (the criticized) classical ones. Hence it makes the contributions
> harder to appreciates"
>
> In Section 4.2, we focused on comparing our results to existing
> results (Wu et al [1], Allen-Zhu and Li [31]) on generative models. The results in Section 4.1 are to
> build intuition for our proof techniques, and what needs to change for
> the ReLU model in Section 4.2. We will add more citations to directly
> compare our results to existing work in Section 4.1.
> Ultimately, as mentioned with our answer to question 2, the differences in assumptions and objectives mean that an exact direct comparison is difficult.
>
> **Q5:**"Theorem 4.1 ... must states the full
> assumptions eg $ n \ge k/2$"
>
> Thank you for the suggestion, we shall include this. This was implicit
> in Theorem 4.1 as $\varepsilon < 1$ with a large enough constant $C$,
> but we will explicitly mention this. We will also mention that these
> bounds are not for the over-parameterized regime.
>
> **Q6:**"What are the novelties over OLS analysis."
>
> The analysis in the reference provided by the reviewer has a sample
> complexity that depends on the design matrix $\Phi$, in terms of
> trace$[(\frac{1}{n} \Phi^T \Phi)]$. This would introduce a
> dependency on the $\ell_2$-norm of $x$ in Eqn 6 in our paper, and typical assumptions to deal with this would require bounded moments for $x$. Our sample
> complexity has no dependence on this design matrix.
>
> We avoid this dependence on the $x$ distribution by adopting a similar analysis to Theorem 11.2 in Györfi et
> al (https://link.springer.com/book/10.1007/b97848), which is relatively simple because $d(\hat{w}, w^*)$ is bounded.
>
> For Theorem 4.5, we cannot fall into the OLS analysis for half the proof, and we use the Györfi approach twice.  The second time is more challenging because the variables we concentrate are KL divergences, which are unbounded.
>
> **Q7:**"In Lemma 4.3, can the authors specify what the
> probability Pr is? Also, since the (empirical) distance is Lipschitz,
> can't we deduce this lemma directly from Talagrand inequality?"
>
> The $\mathrm{Pr}(\cdot)$ refers to the probability over the finite data samples $x$
> drawn i.i.d. from some arbitrary distribution $D_x$.  That lemma is independent of the samples $y_i$, sorry for the confusion.
>
> The distance is actually not Lipschitz: it's Lipschitz in $\langle w, x \rangle$, but if $x$ is extremely large then a small change in $w$ can have a large change in $d(w, w^*)$.  So we don't see how to apply
> Talagrand's inequality without assumptions on the distribution
> of $x$.
>
> **Q8:**"In which practical settings do we know the ground-truth
> conditioning number?"
>
> This is a good point and an open question which we shall explicitly
> state. We only require an upper bound on the condition number, and as
> our sample complexity scales logarithmically on the condition number,
> hence we did not consider this to be a major limitation.
>
> **Q9:**"$\widehat{W}, \widehat{\Sigma}$ may not be unique".
>
> We will state this more explicitly in the paper, but we do not require
> uniqueness.  The reason for this is that as we are not trying to
> estimate the parameters themselves, and instead care about fitting the
> distribution of the data, any parameters that fit the observed
> distribution are acceptable.
>
> **Q10:**"How does the proposition escape known failures of the
> MLE."
>
> In the limitations section, we did mention that our proposition is succeptable to failues of MLE, such as requiring knowledge of the
> generative model's architecture, exacerbating bias, etc. The failure case mentioned by the reviewer would correspond to exacerbating bias, as having too few samples leads to a heavily overfit solution.

---

### Official Review · Reviewer_77Ng · 2023-07-26

**Soundness:** 3 good
**Presentation:** 3 good
**Contribution:** 2 fair
**Rating:** 5
**Confidence:** 4

**Summary:**

1. This paper shows how MLE can perform distribution learning in the setting of linear regression and in multi-layer ReLU networks.
2. This does not take a distribution on labels to be any specific form but rather unknows and tries to derive the sample complexity for a small total variational distance between the model’s conditional distribution and actual conditional distribution.
3. Improves the sample complexity of the previous work which suffers an exponential dependence on the ||W|| term.
4. Generalized how one layer Relu layer sample complexity can be extended to multilayer Relu networks.

**Strengths:**

1. This work is a nice extension of the previous work mentioned in the reference[27]. It relaxes the assumption of previous work which assumed a fixed distribution of the labels.
2. The quality and clarity of the paper is good. Gives good proof techniques using the ideas from the learning theory.
3. Nice result for the linear regression in Theorem-4.1 where the sample complexity is linear in k: dimensionality of the labels which is some finite value in most of the cases.
4. This paper can be significant where the label distribution is unknown but the data distribution is known.

**Weaknesses:**

1. The proof techniques assumed the distribution of the data is Gaussian which is rarely the case because the actual data distribution for generative models thought of some complex unknown distribution.
2. Even though theorem 4.2 proved the result nicely there is a dependence of the square of the dimensionality of data. So the curse of dimensionality remains and these bounds can be very loose.

**Questions:**

1. Theoretically the distribution of labels x being unknown is correct. But in most cases, labels follow some multinomial distribution with some parameters. Does this assumption make the bound better? It would be better if some analysis is being done with the distribution of x being multinomial.

2. The proofs rely on the distribution of y being Gaussian. What would happen if you take the distribution of y as unknown and rather x being some multinomial as it happens in many practical settings of time series and computer vision?

3. There is some confusion in the notation. In line 36 it is mentioned that x denotes the labels and y is the data generated using that label. In linear regression, authors tried to predict y  using x which is correct. In theorem 4.1. is that notation of y being the data and x being the labels being followed or is it reversed?

**Limitations:**

1. There is some limitation where we can have the conditional distribution of data to be Gaussian in those cases the proof techniques do not work.
2. Mostly in the literature of generative models the data distribution is taken to be some unknown complex distribution and labels are multinomial distribution with some appropriate parameters. These bounds don’t address that. So that is why there will be very limited use cases.

---

> ### Author Rebuttal · Authors · 2023-08-09
>
> We appreciate your thoughtful feedback and criticism. Please find
> below our response to your concerns. We wish to emphasize that we do
> not make assumptions about the distribution over the labels or the
> data, we only assume that the conditional distribution per-layer is
> Gaussian followed by a ReLU.
>
> **Weakness 1:** "The proof techniques assume the data is Gaussian".
>
> There's been a misunderstanding here: in Section 4.1 we assume that
> the *conditional distribution* of $y \mid x$ is Gaussian, and in
> Section 4.2 we assume that it is Gaussian followed by a ReLU.
>
> This assumption is not just a theoretical convenience. The additive Gaussian noise before the ReLU is used in state-of-the-art models like StyleGAN, where the learned Gaussian noises are called ``style'' variables: these give desirable stochasticity to the images, such as texture in hair, skin, etc.
>
> As we make no assumptions about the distribution of $x$, the resulting
> distribution of $y$ will be $p_Y(y) = \int_{x} q_X(x) p(y|x) dx$, and this
> distribution can be quite complicated based on the distribution of
> $x$. Furthermore, we can compose our theorem for one-layer networks
> multiple times. This allows us to give sample complexities for
> multi-layer ReLU networks. As this process of linear transformation
> followed by a non-linear activation is repeated $L$ times, the final
> distribution will be far more expressive than a simple Gaussian.
> For example, the distribution of images produces by StyleGAN has such a form.
>
>
> References:
>
> [1] Karras, Tero, Samuli Laine, and Timo Aila. "A style-based generator architecture for generative adversarial networks." Proceedings of the IEEE/CVF conference on computer vision and pattern recognition. 2019.
>
>
> **Weakness 2:**
> "There is a dependence on the square of the dimensionality of
> data... the bounds are loose".
>
> Unfortunately, learning a high-dimensional Gaussian with unknown covariance matrix in total variation takes $\tilde{\Omega}(d^2)$ samples; see [2].
> Our Theorem 4.5 would solve their lower bound instance, so the same lower bound applies and our bound is not loose beyond possibly log factors.
>
> References:
> [2] Ashtiani, Hassan, et al. "Nearly tight sample complexity bounds for learning mixtures of gaussians via sample compression schemes." Advances in Neural Information Processing Systems 31 (2018).
>
>
> **Question 1:**
> "If $x$ has a multinomial distribution, does this improve
> the bound."
>
> Perhaps assuming multinomial
> distributions on $x$ can help simplify the analysis and remove the
> condition number dependence, but it's not going to remove the $d^2$ dependence (which appears even if $x = 0$ always).
>
> The reason we assume an unknown arbitrary distribution
> over $x$ is that it allows us to compose our guarantee over multiple
> layers. For example, if the generative model maps a Word2Vec embedding
> to an images, then at the first layer, the distribution of $x$ would be
> that of a Word2Vec embedding. Then for the second layer, we can treat
> $x$ as the output of the first layer. This allows us to compose our
> guarantee over multiple layers, and hence allows for an expressive
> distribution over the data.
>
> **Question 2:**
> "What would happen if the distribution of $y$ is unknown
> but $x$ has a multinomial distribution".
>
> This would be a significantly harder case, as this would imply that we
> have no way of writing the likelihood of $y$. The assumption that
> the conditional distribution is a Gaussian allows us to write the
> log-likelihood of $y$ in terms of $x$, regardless of the distribution
> of $x$. It's an interesting question what one could do with fewer assumptions on
> $y|x$, but it would need a pretty different approach.
>
> **Question 3:**
> "Line 36 says $x$ denotes the labels and $y$ is the data
> generated using that label ... does Theorem 4.1 use the same notation
> or is it reversed"
>
> We apologize for the confusion -- in line 36, we used labels in the
> sense of the class conditional labels that would be passed to a
> text-conditional generative model. In the context of linear
> regression, $x$ would be feature vectors (such as Word2Vec embeddings)
> and $y$ would be the data generated. We will change this to reduce
> confusion.

---

> > ### Comment · Reviewer_77Ng · 2023-08-19
> > **Response**
> >
> > I have read the rebuttal of the authors. About the misunderstanding in Weakness-1 actually by the distribution of the data I mean the conditional distribution of data depending upon labels. Sorry for the typo and thankful to the authors for the correction. The specific StyleGAN example that they have referred to is correct. The additive learned noise is Gaussian but in the style component, there is another added term of the latent variable(W space concatenated with another learned matrix A) which follows some unknown distribution. So I am not sure how the resulting style variable in each layer follows the Gaussianity assumption. Anyway, the other answers and comments are satisfactory. I will maintain my rating.
> >
> > Thanks.

---

> > > ### Author Response · Authors · 2023-08-20
> > >
> > > Thank you for following up!
> > >
> > > Regarding the question about StyleGAN: indeed, as the reviewer has suggested, the style component has an additive term of the W space variable (which has an unknown distribution) multiplied by a matrix $A$. Note that this linear transformation by $A$ followed by additive Gaussian noise implies that, if we conditioned on the W space variable, the output of each generator layer is Gaussian before applying the non-linear activation.
> > >
> > > In our work, the *variable $x$ has an unknown distribution*, and $y|x$ is Gaussian. Hence, in our notation, we can take $x$ to be the W space variable (which has an unknown distribution), and  the matrix $W$ in our work corresponds to the matrix $A$ in the StyleGAN architecture.
> > >
> > > We hope this resolves your confusion, thank you for your response!

---

### Author Rebuttal · Authors · 2023-08-09

**General response to reviewers**

Thank you for your thoughtful reviews, we appreciate the
time and effort that you put into the review process. We are delighted that the reviewers
found our paper to have: a solid theoretical foundation [NYne], a
novel perspective [9ZPm], considerable improvements over existing
literature [NYne], clarity with good proof sketches[77Ng, 9DZK, 9ZPm,
NYne], and distribution-free results that are significant and
remarkable [77Ng, 9DZK, 9ZPm, NYne].

We have addressed individual reviewer concerns as separate replies to the respective reviewers.

---

### Decision · Program_Chairs · 2023-09-21

**Decision:**

Accept (poster)

**Comment:**

The paper shows that maximum likelihood estimation can be used to learn one-layer conditional generative models in total variation distance regardless of the label x's distribution. The paper is well written and provides an elegant solution. I strongly encourage authors to incorporate reviewer comments in the final version.